# Climate change will reduce North American inland wetland areas and disrupt their seasonal regimes

Donghui Xu [1], Gautam Bisht [1], Zeli Tan [1], Eva Sinha[1], Alan V. Di Vittorio [2], Tian Zhou[1], Valeriy Y. Ivanov [3] & L. Ruby Leung [1]

Climate change can alter wetland extent and function, but such impacts are perplexing. Here, changes in wetland characteristics over North America from 25° to 53° North are projected under two climate scenarios using a state-of-the-science Earth system model. At the continental scale, annual wetland area decreases by ~10% (6%-14%) under the high emission scenario, but spatio-temporal changes vary, reaching up to ±50%. As the dominant driver of these changes shifts from precipitation to temperature in the higher emission scenario, wetlands undergo substantial drying during summer season when biotic processes peak. The projected disruptions to wetland seasonality cycles imply further impacts on biodiversity in major wetland habitats of upper Mississippi, Southeast Canada, and the Everglades. Furthermore, wetlands are projected to significantly shrink in cold regions due to the increased infiltration as warmer temperature reduces soil ice. The large dependence of the projections on climate change scenarios underscores the importance of emission mitigation to sustaining wetland ecosystems in the future.

Inland wetlands are important freshwater resources and one of the most productive ecosystems on Earth[1]. Besides the importance for biodiversity[2], inland wetlands play a critical role in global water, energy, and carbon cycles[3]. Specifically, inland wetlands are the largest natural source of methane ($CH_4$), the second most important atmospheric greenhouse gases[4-7], and can sequester soil organic carbon at a rate much higher than many other ecosystems[8]. Inland wetlands also function as a buffer zone that delays and mitigates the runoff/streamflow peaks[9,10]. They further act as a natural filter that reduces the sediments, nutrients, and pollutants entering groundwater and downstream waterways, thereby improving water quality[11]. Wetlands also significantly impact land-atmosphere interactions because of the enhanced evaporation from the open water or saturated soil[12]. As a result, any changes of inland wetlands may cause cascading consequences to biogeochemical and hydrological cycles at different spatial scales[13].

Climate change can impact the spatiotemporal distribution of inland wetlands[14-17], but the direction and magnitude of the changes remain uncertain[18]. Periodically to permanently inundated by water, wetland regime is strongly controlled by surface water dynamics (hereafter used interchangeably with wetland) that are influenced by variations over small spatial scales. However, some wetlands may not be necessarily inundated by water, as saturated soils are sufficient to create wetland ecosystems. Previous studies using satellite observations at the sub-kilometer scale[14,15,19-21] found that annual precipitation is the dominant factor controlling large-scale surface water dynamics[20,22]. However, temperature can also affect wetlands through its influence on several processes including evapotranspiration, snowmelt, infiltration, soil thawing and freezing, and precipitation[23]. When considered individually, these processes may have very dissimilar outcomes. In addition, the surface water dynamics also depend on the regime of groundwaters which are controlled by climate,

[1]Atmospheric, Climate, & Earth Sciences Division, Pacific Northwest National Laboratory, Richland, WA, USA. [2]Earth and Environmental Sciences Area, Lawrence Berkeley National Laboratory, Berkeley, CA, USA. [3]Department of Civil and Environmental Engineering, University of Michigan, Ann Arbor, MI, USA. ✉e-mail: donghui.xu@pnnl.gov; gautam.bisht@pnnl.gov

vegetation dynamics, and human extraction. Given the complexity and nonlinearity of the climate system and related hydrological responses[24,25], data-driven approaches could be useful for deriving the relationships between wetland dynamics and their hydroclimatic drivers[26]. However, statistical relationships derived from data-driven methods trained using historical observations only may not be transferrable to the future warmer climate as wetlands are sensitive to nonstationary climate trajectories[27,28]. Besides, the accuracy of such approaches depends on the quantity and quality of training data, which are still limited for surface water dynamics over large areas. Although numerous studies derived surface water dynamics from satellite datasets at regional or national scales[20,29–32], such data are limited at the integrating continental and global scales[14,15], and the spatial coverage of monthly wetland dynamics is very poor (Supplementary Fig. 1).

Earth system models (ESMs) are alternative tools for understanding the large-scale wetland changes induced by external forcings and uncovering the driving mechanisms[33]. ESMs are physically based models that couple atmosphere, land, ocean, land ice, sea ice, and river processes at large scales. Typically applied at spatial resolutions of ~100 km or coarser, ESMs parameterize smaller-scale processes that are not explicitly resolved by the models. Compared to data-driven methods, ESMs do not require a large amount of data for training and can well capture the wetland evolution under changing conditions if the related processes are appropriately parameterized. However, the current ESMs usually represent the wetland hydrology in oversimplified ways[33]. Additionally, they are highly uncertain in representing the two inundation processes that form inland wetlands: fluvial and pluvial processes. Fluvial inundation occurs when river flow accumulated from upstream exceeds the channel capacity and generates overbank flow flooding into the neighboring floodplain wetlands[34–37]. Pluvial inundation typically occurs in low-lying areas, when excess water from precipitation, overland flow, and groundwater discharge cannot infiltrate into the soil or drain away with surface flow. Both fluvial and pluvial inundation mechanisms of formation are needed to explicitly model wetland dynamics as wetlands interact with rivers, runoff generation process, surface-subsurface interaction, and evaporation process. While most ESMs can reasonably capture the dynamics of fluvial inundation[34,36], the pluvial inundation process is generally inferred using a diagnostic scheme. For example, a wetland diagnostic scheme uses the simulated groundwater depth to estimate the wetland areas, but the inferred wetland areas do not impact other processes in the model[4,33,38–41]. This widely used diagnostic schemes ignore the soil freeze-thaw cycle, which is critical for the wetlands dynamics in cold regions[39]. Accurately representing soil freeze-thaw process is necessary for understanding the interactions between wetland dynamics and groundwater under climate change conditions, as over half of the global wetlands are located in the northern high latitudes (e.g., north of 50° N)[42]. Furthermore, parameter calibration is commonly needed to constrain ESMs' uncertainty, which is computationally intensive for applications at large scales. As a result, the parameters of ESMs related to fluvial and pluvial inundation are usually not well constrained. The process simplification and inherent parametric uncertainty could result in significant biases in wetland simulations, especially when regional understanding of their regime is sought.

In this study, we aim to understand the drivers and future trajectories of inland wetland area changes over North America from 25°N to 53°N) using a state-of-the-science ESM, Energy Exascale Earth System Model (E3SM), based on the most up-to-date climate change projections. We implemented a modified infiltration scheme to improve modeling of the pluvial inundation process (see Methods) in E3SM[43]. Simulations were performed using the coupled land and river components of E3SM, and calibrated against an upscaled global surface water dynamics dataset from Global Land Analysis & Discovery (GLAD, see details in the Methods)[15]. Another global satellite dataset[14] is available for use in model calibration as well, and it has been used to benchmark GLAD[15]. The Shared Socioeconomic Pathways (SSP)[44] scenarios for lower emissions (SSP126) and higher emission (SSP585) were used along with five global climate models from the Coupled Model Intercomparison Project Phase 6 (CMIP6) to provide a multi-model ensemble capturing a range of warming trajectories[45,46]. Compared to previous studies of large-scale wetland projections, this work improves the representation of wetland dynamics in an ESM to increase the confidence in wetland projections by (1) relying on a process-based inundation process to identify wetland dynamics; (2) running the simulation at a relatively higher spatial resolution; and (3) calibrating model parameters against satellite dataset. We hypothesized that climate change will significantly impact wetland area (i.e., defined as the sum of fluvial and pluvial inundated areas excluding rivers, lakes, and reservoirs) and wetland habitats (i.e., defined as the inundated area for at least 1 month during the growing season with surface temperature above 5 °C). Additional definition details can be found in Methods. We further uncover the driving mechanisms for the wetland changes from different emissions scenarios.

## Results

### Wetland characteristics during the historical period

The E3SM model with refined wetland hydrology and reduced parametric uncertainty closely captures the upscaled surface water extent retrieved from satellite data over 1999–2020. At an annual time scale, the E3SM simulations demonstrate good performance when benchmarked against upscaled satellite observations at the model spatial resolution of ~12.5 km × 12.5 km (Fig. 1a, b). The model also adequately captures the observed surface water seasonality (Supplementary Fig. 2), despite the weaker performance during winter. The latter can be attributed to data gaps in the GLAD dataset (Supplementary Fig. 3a) and challenges in model representation of snow melting processes[47]. Importantly, the calibrated model captures the positive trend of annual wetland changes from 1999 to 2020 (Fig. 1c), which was also reported in a previous study[20]. The performance of capturing interannual variability is demonstrated by the high evaluation metrics in Supplementary Fig. 4. The simulated surface water dynamics closely follow the benchmark for selected zoomed-in regions (e.g., with an averaged correlation coefficient of 0.83), though substantial underestimation can be found around the Great Salt Lake (Supplementary Fig. 5). The model performance is further validated using the Global Inundation Extent from Multi-Satellite[19] during an independent period (1993–2007), showing consistency with our simulated spatiotemporal variation and negative annual trend of wetland area (Supplementary Fig. 6). The high fidelity of the refined E3SM in simulating the inland wetland area changes is largely due to the improved soil water infiltration scheme (Supplementary Fig. 7) that resolves the difference of infiltration between saturated soils and unsaturated soils within a grid cell (see more details in Methods). In contrast, the surface water extent is significantly underestimated without improvements in representing pluvial inundation process, except in snow-dominated regions during cold season (Supplementary Fig. 8) when there is likely significant overestimation of surface water extent.

Pluvial inundation is the dominant wetland generation mechanism over North America. Aggregated from model grid cells to basin scale (see Methods), the pluvial process accounts for more than 70% of the annual surface water area over 2282 of the total 2478 selected basins (Fig. 1d). Fluvial inundation is important for surface water dynamics in only 37 basins, where fluvial process explain more than 70% of the annual surface water area. When averaged at the continental scale, inundated areas due to pluvial mechanism represent ~90% of surface water extent. Pluvial inundation is not only important for the averaged wetland areas, but also explains a large fraction of their temporal variation (Fig. 1e), while fluvial inundation is mostly

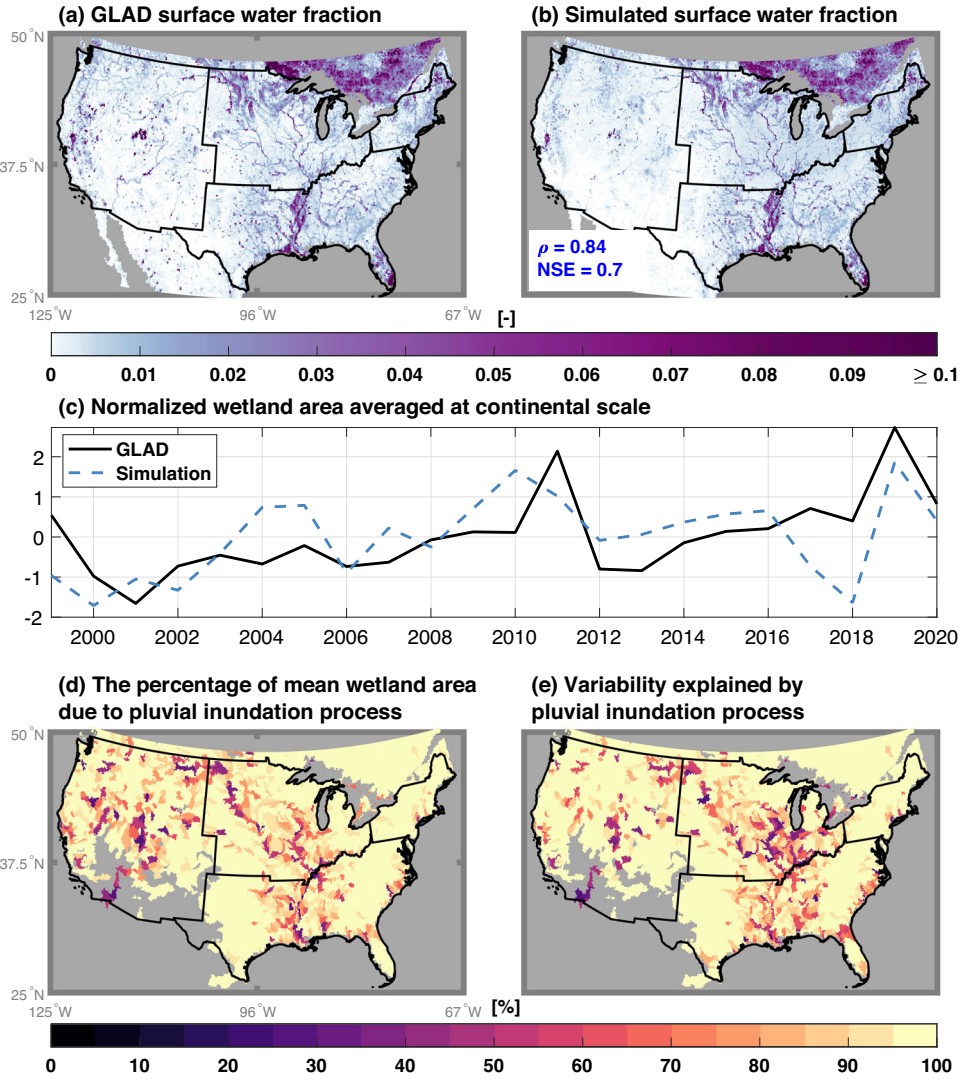

**Fig. 1 | Comparison of simulated surface water (unit: fraction of a grid cell) with satellite observation from Global Land Analysis & Discovery (GLAD) averaged over 1999–2020. a** Upscaled GLAD surface water with permanent water bodies removed; **b** Simulated surface water. The inset text in (**b**) shows the evaluation metrics comparing the simulation with the upscaled GLAD surface water fraction, where $\rho$ and NSE are the Pearson correlation and Nash−Sutcliffe model efficiency coefficients, respectively. **c** Shows the normalized wetland area (subtracting the mean and divided by the standard deviation) comparison in Y-Axis at continental scale between the simulation and GLAD. X-Axis represents the year. **d, e** Iillustrate contribution of pluvial inundation process to the mean and temporal variability of surface water during 1971–2000, respectively. In (**d**), the contribution is obtained by dividing the pluvial inundated area by the mean wetland area. In subplot (**d**), the coefficient of determination ($R^2$) between the annual time series of pluvial inundation and wetland area is used to determine the contribution of pluvial inundation process to the annual wetland area variability. The gray color in (**a**), (**b**), (**d**), and (**e**) denotes no available data or areas with negligible wetland in the historical period (i.e., less than 0.05%).

significant along major rivers (e.g., Mississippi river, Colorado river, etc.). However, a time-invariant wetland area is routinely removed from satellite-observed surface water to derive fluvial inundation in previous studies[36,48], assuming the variability of pluvial inundation is negligible in explaining surface water dynamics. Such assumption can bias benchmark datasets towards higher importance of fluvial inundation mechanism. Overall, the inclusion of pluvial inundation in land processes of ESMs emerges as crucial for accurate understanding wetland area dynamics.

**Future changes in wetland area**
At the continental scale, wetland area decreases in all seasons in the future with the exception of winter (Fig. 2). These projected seasonal changes result in a decreasing annual averaged wetland area under both lower and higher emission scenarios (Fig. 2a). Specifically, the median annual wetland area in 2071–2100 decreases by 5.2%

(4.2–7.0%) and 10.6% (5.9–13.5%) based on our multi-model ensemble projections under SSP126 and SSP585, respectively, relative to the historical period (1971–2000). The 25th percentile of annual wetland area decreases by 5–20%, suggesting that climatologically drier wetland environments become even drier in the future. Further, wetland seasonality is significantly altered in both SSP126 and SSP585, with increased wetland area during winter (Fig. 2b) and decreased wetland area in other seasons (Fig. 2c–e). Such differences of seasonal wetland changes are driven by the changes in water supply to wetlands (i.e., rainfall and snowmelts) which increases more in winter than other seasons in the warmer future (Supplementary Fig. 9). In SSP126, the increase in wetland areas during winter offsets their decrease in the other three seasons, resulting in smaller changes in wetland area at the annual time scale. However, the SSP585 projections yield a significantly larger reduction of wetland area in summer and fall, with wetlands at the 25th percentile and median sizes decreasing by up to -25% by the

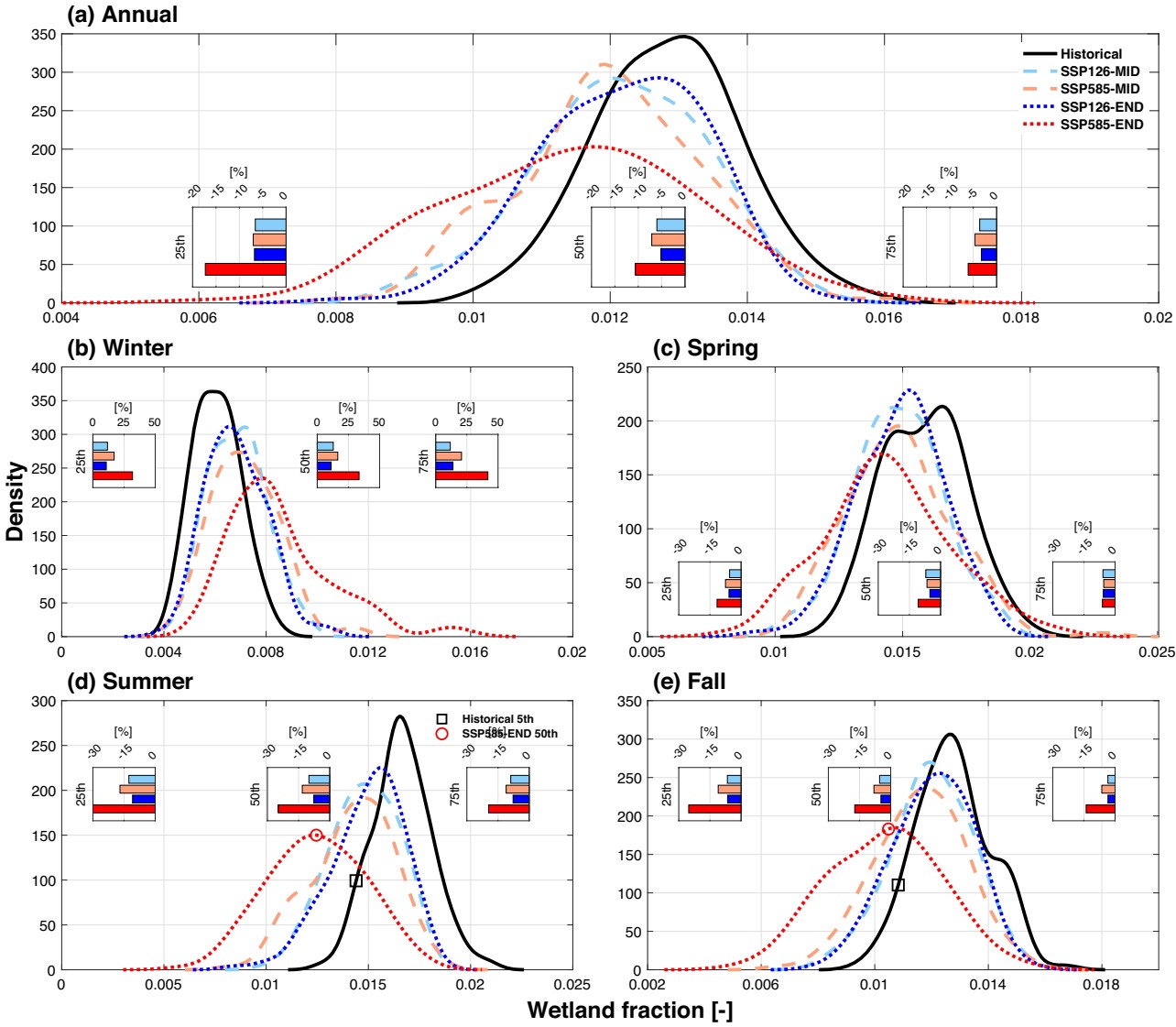

**Fig. 2 | Projection of wetland area at continental scales under different scenarios.** Probability distribution of continent-averaged wetland area (including both pluvial and fluvial inundation) for (**a**) annual, (**b**) winter, (**c**) spring, (**d**) summer, and (**e**) fall. Historical, MID, and END denote the historical period (1971–2000), mid-century (2041–2070), and end-of-century (2071–2100), respectively. Each probability distribution function is constructed from the muti-model ensemble. The inset plots show the relative change of wetland area for wetland area at the 25th, 50th, and 75th percentile between each future scenario (FUT) and the historical period (HIS): $\frac{FUT-HIS}{HIS} \times 100\%$. Wetland area is represented as a fraction of total simulation domain area on the x-axis. In (**d**) and (**e**), the 5th percentile wetland fraction in the historical period and the 50th percentile wetland fraction at the end of the century under SSP585 are indicated by black square and red circle for comparison.

end of this century (Fig. 2d, e). Consequently, the SSP585 projections exhibit a significant reduction of wetland area at the annual time scale. The projected changes in wetland areas remain relatively stable from mid-century to end-century under SSP126, suggesting that wetlands can be conserved if the global warming level is constrained. However, under the high emission scenario, the projected reduction in wetland areas intensifies by the end of the century. For example, the median wetland areas in future summer and fall seasons are even smaller than the 5th percentile of the historical period (Fig. 2d, e), implying that drier wetland environments will be much more common during growing seasons under the SSP585 scenario.

Regionally, the shift of wetland area seasonality varies (Fig. 3). The northeastern and western US are projected to have a higher wetland area earlier in the year which is consistent with previous studies that attribute that to earlier onset of snowmelt induced by the warmer climate[49]. However, the midwestern and southern US will experience a delayed wetland area seasonality, potentially associated with spring-time soil wetness changes[49]. The direction of seasonal shift in wetland

dynamics is consistent between the two SSP scenarios, but the shift is more pronounced in the higher emission scenario. Such seasonality changes can present challenges for water management and agriculture and have consequences for ecosystem diversity based on plant adaptive potential[50].

Changes in wetland dynamics are more pronounced at the basin scale, with a strong spatial pattern featuring divergent trends (Fig. 4). For example, the multi-model ensemble simulations project a pattern of change that is spatially consistent for both SSP126 and SSP585, with an increase in wetland area in the western mountainous regions, Midwest, Northeast, and Florida and decreasing wetland area in Southwest, Southern Great Plains, the southern Appalachian Mountains, and southeastern Canada. The spatial contrast intensifies with the warming level, such that 15% and 35% of the continent show at least a 25% absolute change in wetland area by the end of the century under SSP126 and SSP585, respectively. Increasing wetland areas are mainly located in snow-dominated regions during winter (Fig. 2b), and attributed to earlier snowmelt and higher fraction of liquid

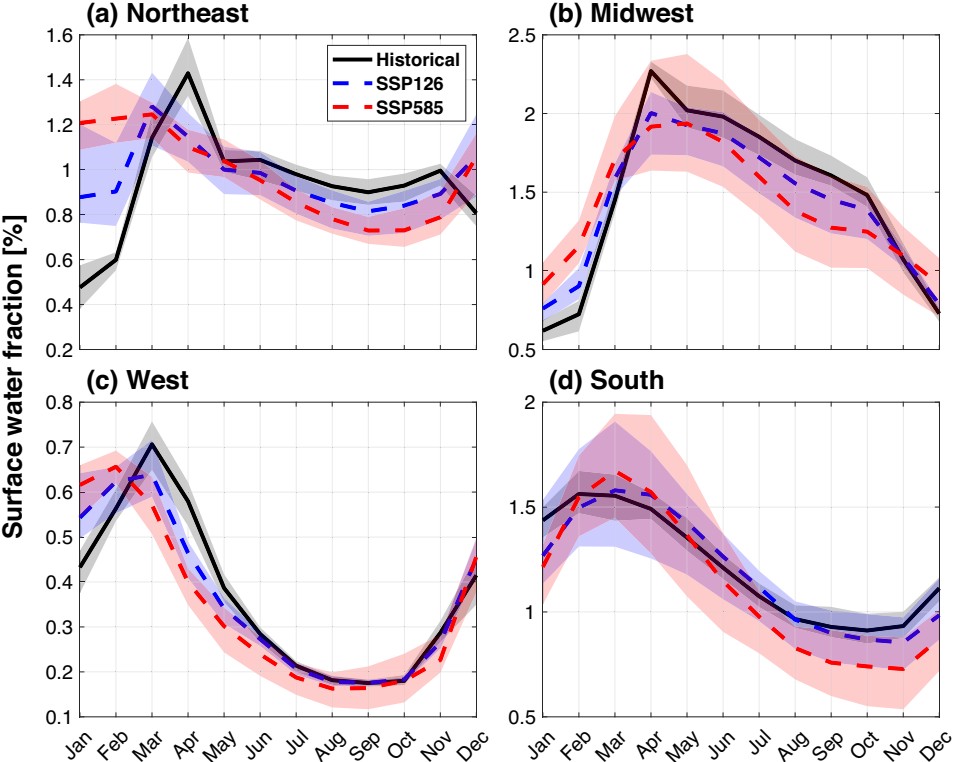

**Fig. 3 | Projection of surface water at regional scales under different scenarios.** Seasonality of surface water averaged over the (**a**) Northeast, (**b**) Midwest, (**c**) West, and (**d**) South of United States. The black line represents the multi-model mean for the historical period (1971–2000). The blue dashed line and red dashed line are the multi-model means for the end-century period (2071–2100) from SSP126 and SSP585, respectively. The shaded areas denote the corresponding 5%-95% of the multi-model ensemble. The subregion boundary can be found in Fig.1a.

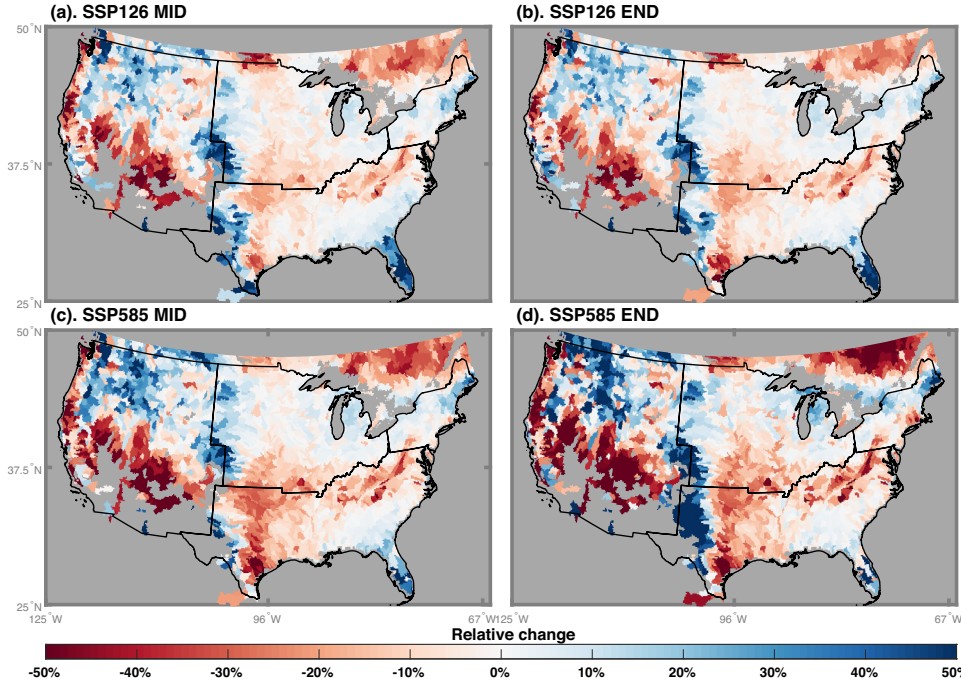

**Fig. 4 | Relative change of mean annual wetland area between the future (FUT) and historical (HIS) periods.** The relative change ($\frac{FUT - HIS}{HIS} \times 100\%$) is estimated as the mean of the equal-weighted multi-model ensemble. "MID" (**a**, **b**) represents the relative change between 2041–2070 and 1971–2000, while "END" (**c**, **d**) is the change between 2071–2100 and 1971–2000. The gray color denotes no available data or areas with negligible wetland in the historical period (i.e., less than 0.05%). Results are shown at the basin scale (See Methods and materials).

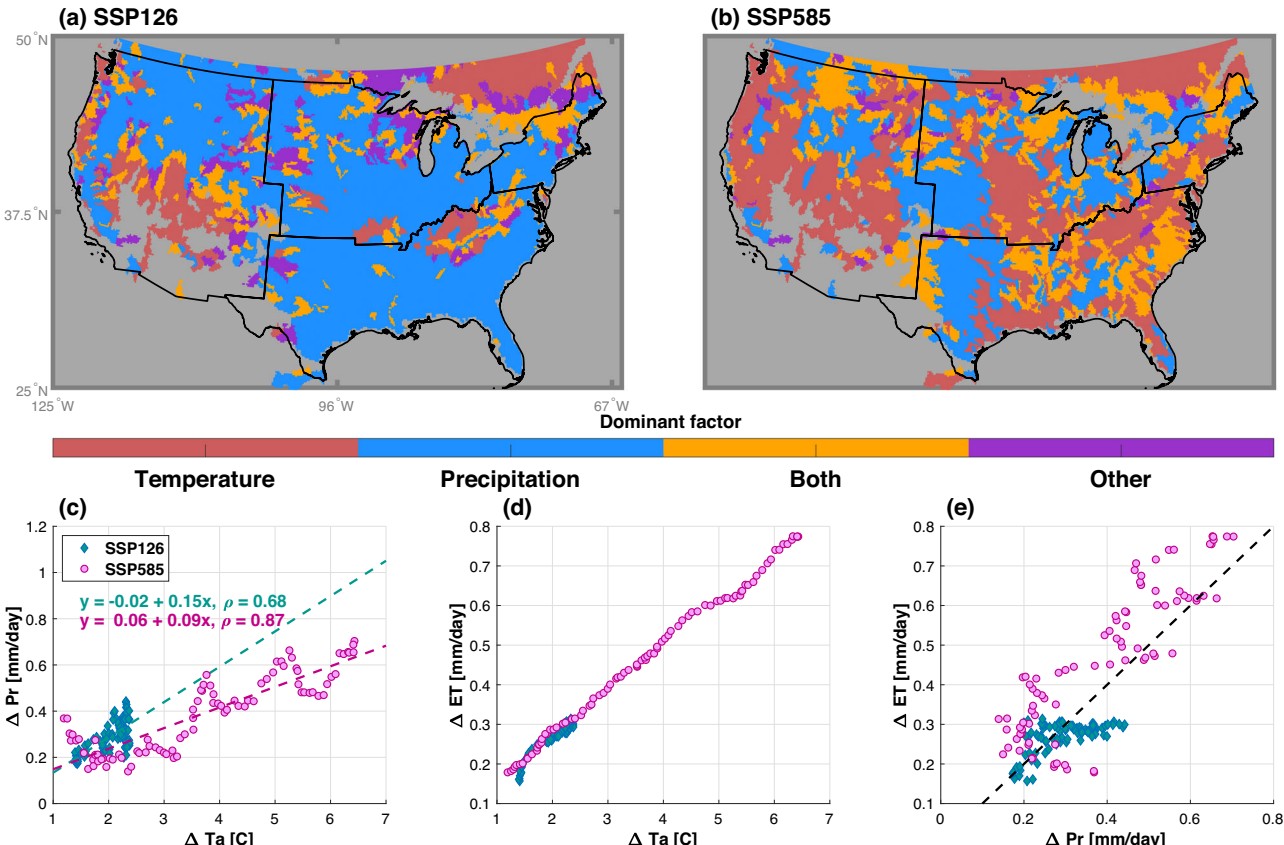

**Fig. 5 | Dominant drivers for the changes of wetland area under different scenarios.** (**a**) SSP126 and (**b**) SSP585. "Both" means both climatic factors are critical for wetland changes, while "Other" means neither temperature (Ta), precipitation (Pr), nor their combined effect explains wetland changes. The gray color denotes no available data or areas with negligible wetland in the historical period (i.e., less than 0.05%). **c**–**e** show scatter plots for the change of precipitation, change of temperature, and change of evapotranspiration (ET) for an exemplary region delineated by the white boundary in (**a**) and (**b**) over southeastern US. The inserted text in (**c**) are the fitted linear regression equation and the corresponding Pearson correlation between change of Ta and change of Pr. Dashed lines in (**c**) and (**d**) are linear regressions, while as the dashed line in (**e**) is a 1:1 line.

precipitation with warming enhance winter runoff[49]. However, most regions are projected to have less wetlands during growing seasons (Supplementary Fig. 10), except for Florida and Rocky Mountain regions, where wetland area is projected to increase throughout the year.

**Attributing the changes in wetland dynamics**

Changes in both temperature and precipitation can influence wetland area by perturbing the surface water balance. The projected changes in wetland area are strongly correlated with the concurrent precipitation and temperature changes, with an averaged correlation coefficient of 0.76 and 0.86, respectively, based on multiple linear regressions across all the basins under SSP126 and SSP585 (Supplementary Fig. 11a, b). Notably, wetland area is only sensitive to one climatic factor in certain regions, such as precipitation changes in the southeastern US under SSP126 (Supplementary Fig. 11c, e) and temperature changes in the coastal mountains of the western US under SSP585 (Supplementary Fig. 11d, f). As the changes in temperature and precipitation are highly correlated across the models (Supplementary Fig. 12), the changes in wetland area are statistically strongly correlated with both temperature and precipitation for some regions. However, through different impacts of temperature and precipitation on the generation mechanism of wetlands, either temperature or precipitation changes can dominate the wetland changes in different regions. For example, the decreasing wetland trend in southeastern Canada results from perched water table (i.e., water table underlain by soil ice) drop associated with increasing temperature rather than precipitation increases.

Note that the soil ice in southeastern Canada is not related to permafrost, but rather due to the freezing of soil moisture in the subsurface when the temperature falls below the freezing point. Therefore, a detailed attribution analysis is required to further uncover the mechanisms that drive the wetland changes (see details in Methods).

Overall, precipitation changes dominate wetland changes under SSP126, while warming is the dominant factor under SSP585. For the SSP126 scenario, 55% of all the basins are controlled by precipitation changes (Fig. 5a), while 72% are controlled by temperature changes for SSP585 (Fig. 5b). Previous studies identified precipitation to be the major driver of wetland variabilities based on satellite observations[20,22], consistent with the larger role of precipitation in the wetland dynamics under the lower emission scenario with global mean warming below 2 °C throughout this century. However, the dominant driver shifts from precipitation to temperature (or both) for the higher greenhouse gas emissions scenario, with global warming exceeding 4 °C by 2100 relative to 2000, highlighting the higher evaporation is increasingly negatively affecting wetland area. Higher temperatures likely reduce wetland area due to evapotranspiration (ET) that grows because of enhanced air vapor pressure deficit. This can also be due to an increase in infiltration losses in the colder season because of the increased soil hydraulic conductivity (e.g., ice in the soil thaws) and the higher fraction of liquid precipitation.

Broadly, wetland dynamics are controlled by the net precipitation (difference between precipitation and ET). The shift of the dominant wetland driver from precipitation to temperature is caused by the different sensitivities of precipitation and ET to temperature increases.

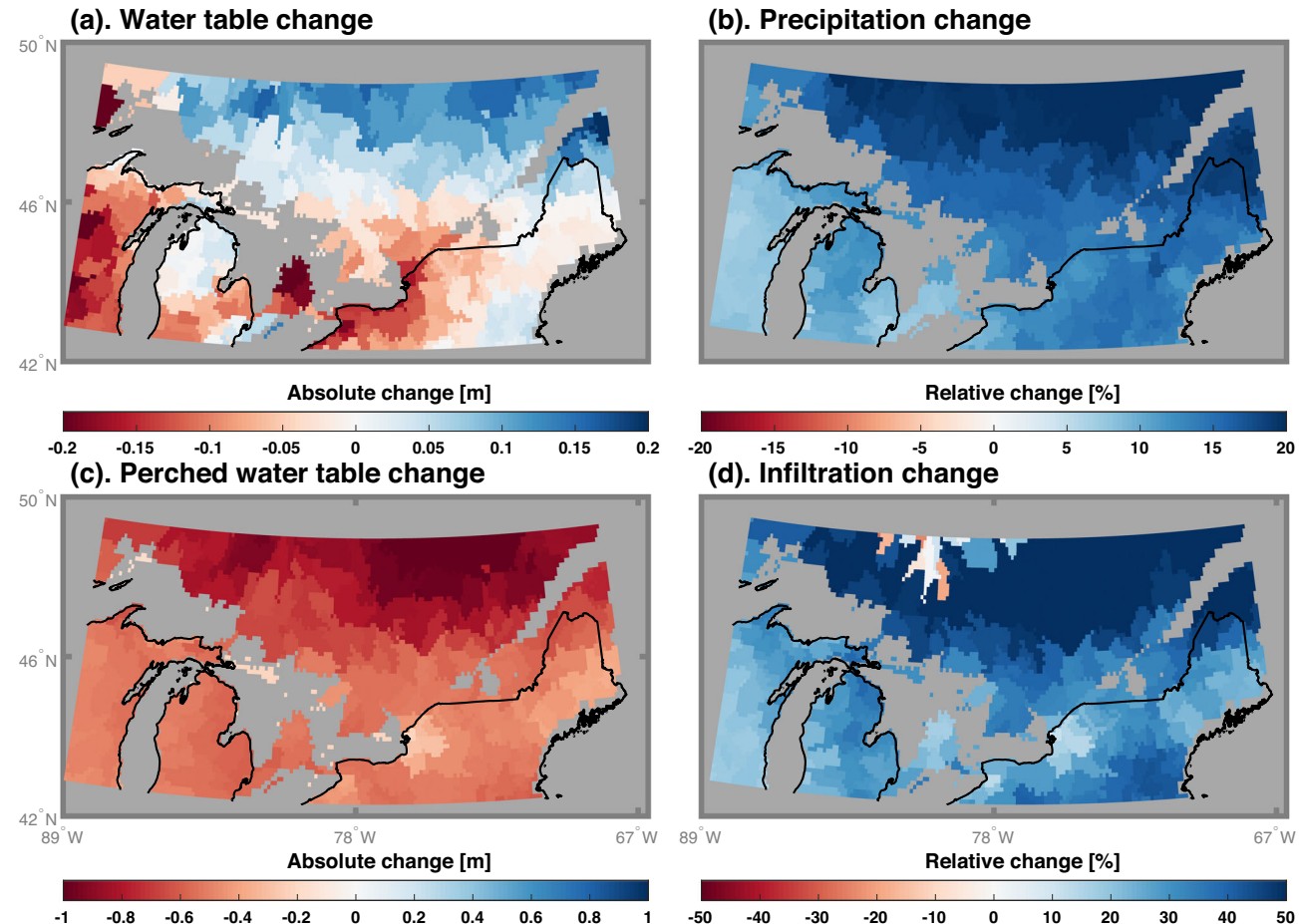

**Fig. 6 | Projected changes of groundwater dynamics over the southeastern Canada under SSP585.** Changes in (**a**) groundwater table elevation, (**b**) precipitation, (**c**) perched groundwater table elevation, and (**d**) infiltration amount between SSP585 end-century projection and historical period. All of the changes were estimated as the mean of the equal-weighted multi-model ensemble projections. Positive water table changes imply groundwater rise, while negative values mean water table deepening. The gray color denotes no available data.

Specifically, warming does not strongly constrain regional precipitation changes (Fig. 5c), which are also influenced by the changes in atmospheric circulation, but higher temperatures have a significant control on ET (Fig. 5d), as the ratio of surface latent to sensible heat fluxes increases with temperature[51,52]. Therefore, under SSP126 with limited warming, the increase of precipitation can be larger than the increase of ET (i.e., the blue diamonds are below the 1:1 line in Fig. 5e), dominating the wetland changes. However, under SSP585 with higher warming level, ET increases substantially and always surpasses the precipitation increase (i.e., the magenta circles are above the 1:1 line in Fig. 5e), which results in unidirectional drying trend in wetland dynamics. As a result, temperature increase becomes the dominant driver for wetland changes under the higher emission scenario. Despite the shift in the dominant driver from precipitation in SSP126 to temperature in SSP585, the spatial pattern of the wetland area change directions are similar between the two scenarios (Fig. 4). This is because both precipitation and ET increase with warmer temperature (Fig. 5c, d), albeit at different rates for precipitation under the two scenarios, and the patterns of precipitation change are similar between the two scenarios. With ET increasing monotonically with warming and precipitation change showing more variable behavior with increasing temperature, there are larger reductions per 1 °C warming in the SSP585 scenarios, as compared to SSP126 (Supplementary Fig. 13).

Unlike in most other regions, temperature significantly dominates the wetland dynamics in southeastern Canada and the southwestern US for both the lower and higher emissions scenarios. In cold regions (e.g., southeastern Canada), surface water accumulates in wetlands when the soil is frozen, thereby constraining infiltration (e.g., ice reduces hydraulic conductivity), or partially frozen, when the surface soil can be easily saturated since water percolation is inhibited (e.g., shallower perched water table). By ignoring these processes, the commonly used diagnostic wetland scheme projects expanding wetland areas over the cold regions[33] due to the rising groundwater level (Fig. 6a) caused by increased precipitation (Fig. 6b) and higher temperatures projected in the future[53]. However, our physically based wetland scheme projected shrinking wetlands since the higher temperatures thaw the soil ice, which increases infiltration from wetlands (Fig. 6a) and leads to deepening of the perched water table (Fig. 6c). Although lower hydraulic conductivity due to frozen soil is the major generation mechanism of wetlands in southeastern Canada, we note the surface water can stay in wetlands during warm periods due to soil saturation. The southwestern US is a hot and dry region, where wetland area dynamics are mainly controlled by evaporative losses. Increased evaporation with warming reduces wetland area in that region year-round under both scenarios. Although wetlands are not common for dry regions, they cover significant areas in northern high latitudes[42]. With a high sensitivity of the wetland water cycle to temperature changes in cold regions (e.g., annual temperature around or below 0 °C, Supplementary Fig. 14), global warming may induce a significant loss of wetlands in these regions – an unexpected result supported by physical considerations of thermal and water-related processes.

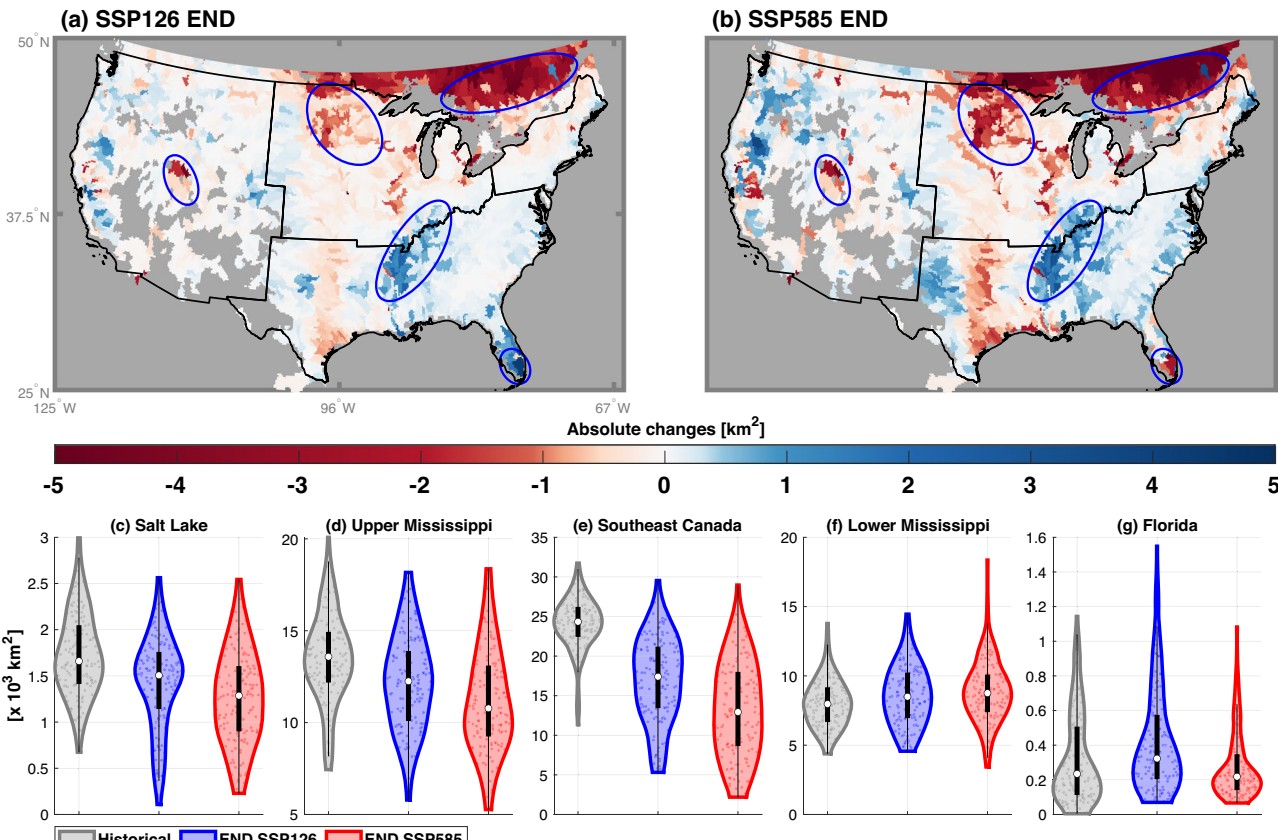

**Fig. 7 | Absolute change of wetland habitats between the historical (HIS) and future (FUT) periods.** In subplots (**a**) and (**b**), "END" represents the difference of multi-model ensemble simulation mean between 2071–2100 and 1971–2000. The gray color denotes no available data or areas with negligible wetland habitats in the historical period (i.e., less than 0.05%). Subplots (**c**–**g**) show the probability distribution of annual wetland habitat area [$\times 10^3 km^2$] for five major wetland regions circled in subplots (**a**) and (**b**). Simulations forced by all five climate forcings during the historical (1971–2000) and end-of-century (2071–2100) periods are used to construct the violin plots. The white circles represent the medians, the black lines denote the 25–75% percentile of the distribution, and the boundary of the violin plot represents the density of the scatters.

## Impacts of climate change on major wetland habitats

Wetland habitats will significantly shrink in 36% and 41% of the study basins ($p < 0.05$ in ANOVA test) under SSP126 and SSP585, respectively (Fig. 7a, b). The major wetland habitats (encircled in green in Fig. 7a) show larger impacts caused by the climate change. By the end of this century, wetland habitats near Salt Lake, upper Mississippi, and Southeast Canada will shrink by 27% (18%), 18% (10%), and 45% (30%) under the higher (lower) emissions scenario according to multi-model ensemble mean of the projections (Fig. 7c–e). Our projected reduced wetland habitats are consistent with previous studies assessing the sensitivity of wetlands to climate change in the Great Salt Lake region[54] and Prairie Pothole Region[55]. The lower Mississippi region is projected to have a significant increase in wetlands habitats by about 10% (Fig. 7f), which can be caused by the increased streamflow[56]. Notably, the Everglades wetland region in Florida exhibits different sensitivities under the two emission scenarios with a 30% increase in wetland habitats under SSP126 and a 14% decrease in wetland habitats under SSP585 (Fig. 7g). A similar nonlinear response of the Everglades wetland to different climate change scenarios was reported in a previous study[57]. Such uncertainty may be caused by the change of the dominant driving factor from precipitation control (under milder warming) to temperature control (under larger warming, Fig. 5), or changes in maximum precipitation dynamics (Supplementary Fig. 15), resulting in changes in available water. Furthermore, the projected wetland habitats exhibit greater temporal variability in the future compared to historical period (Fig. 5c–g). Consequently, the major wetland habitats can experience more significant reduction during drier years than normal years, especially under the higher emission scenario.

Importantly, we find that the change of functional wetland habitats will not always follow the change of inland wetland area under climate change, which is contradictory to the assumption used in previous studies[33]. For example, wetland area is projected to increase over upper Mississippi and decrease over lower Mississippi (Fig. 4), but the changes of wetland habitats exhibit an opposite direction (Fig. 7). The difference between the projected changes in wetland area (Fig. 4c, d) and wetland habitats (Fig. 7a, b) can be attributed to the changes in inundation seasonality and increase in precipitation variability with warming[58]. Specifically, the maximum inundation during the growing season can decrease as the peak inundation shift to winter months, while the annual averaged inundation increases compared to historical period. As the proper inundation during the growing season is needed for the function of wetland habitats but projected to decrease substantially in the future warming conditions (Supplementary Fig. 10), wetland habitats can be more sensitive to global warming than wetland area. It emphasizes the significance of considering inundation seasonality to better understand wetland evolution in the future.

## Discussion

Inland wetlands have been found to shrink significantly at global scales in the past several decades, driven mainly by human activities[16,21,59,60]. Global warming (i.e., climate driver) will further negatively impact

wetland ecosystems by altering their water balances[33,61–64], which is demonstrated in this study by using a physically based wetland generation scheme in a state-of-the-science ESM. Specifically, we project the changes of wetland characteristics for North America under different climate projections following two socioeconomic trajectories. Compared to historical period, the averaged wetland area at continental scale will decrease by ~5% in the low emissions scenario but will substantially decrease by ~10% in the high emissions scenario. In addition, the wetlands during drier years will be more disrupted than an average year, as wetland area of 25th percentile in the future period is projected to decrease by ~20% in the high emissions. The wetland change is more significant at the regional scale with a divergent trend, such as over 50% loss of the wetlands in southeastern Canada and ~50% increase of wetlands in some western US mountainous regions. The seasonality of wetland dynamics is projected to shift as well, with increased wetland area in the winter and decreased wetland area in spring, summer, and fall.

Wetland changes also exhibit regional features associated with the dominant driver of the change. Under SSP126 with milder warming, the wetland changes are mainly driven by precipitation changes, but with larger warming under SSP585, temperature (and associated monotonic increase in ET) becomes a dominant driver in many regions across North America. With the dominant driver shifting from precipitation to temperature in the higher emission scenario, wetland area will experience larger reductions per 1°C temperature increase in SSP585 than SSP126. Consequently, the wetlands of Florida are projected to expand under SSP126 but shrink under SSP585 due to climate factors.

We further found that wetlands from cold regions are only sensitive to temperature changes under both SSP126 and SSP585 (e.g., southeastern Canada). Because the surface water is sustained by perched water table in frozen soil, thawing of the frozen soil in warming scenarios lowers the perched water table and reduces the surface water area. This sensitivity and the potential permafrost thaw[39,65] imply that Northern Hemisphere high latitude regions may lose large wetland areas in a warmer climate. Our results dispute the conclusion from previous studies that wetland over cold regions will expand with warming because of larger inundated areas and permafrost thaw[4,33,66]. Other study also suggested a potential loss of wetland over northern high latitudes, but attributing to a different factor than our study, i.e., the increased evapotranspiration[67]. We argue that the difference arises because previous studies ignored the process that accounts for increasing infiltration when soil ice thaws, which is a significant sink term in the wetland water budget.

Functional wetland habitats are particularly vulnerable to the impacts of climate change, with the major wetlands of the Salt Lake, upper Mississippi, and southeastern Canada regions projected to shrink significantly under both emissions scenarios. This projected reduction in wetland habitats is caused by the early shift in inundation seasonality, such as peak inundation shifts from growing seasons to winter months. Wetland habitats reduction will lead to substantial loss of biodiversity and negative impacts on ecological processes since wetlands provide home to a variety of plant and animal species. According to our multi-model ensemble projections, the future wetland habitats can shrink to a much lower level during drier years than the historical period due to the significant water deficits (e.g., high evapotranspiration and low precipitation)[62]. This presents additional challenges to wetland ecosystems considering that droughts are likely to be more intense and frequent in the future[68]. For example, if certain plant or animal species are unable to survive during periods of drier wetland habitats, they may not regrow or return in the later wetter years. Losing biodiversity will further result in instability of wetland ecosystems. Additionally, under the future conditions, temperature becomes a more dominant driver of wetland dynamics and reduces wetland areas by increasing evaporation and infiltration, which in turn

lower the carbon sequestered by wetlands[69], further aggravating global warming. Although the wetland habitats are sensitive to any increases in temperature over certain regions, constraining global warming to a lower level is key to reducing vulnerability of wetlands to climate change.

Uncertainties in projecting wetland changes may come from different sources. First, the separation of wetland dynamics into pluvial inundation and fluvial inundation can be affected by the bias of inundation scheme of river component[48], which requires calibration. However, previous studies demonstrated that the macro inundation scheme in the river component of E3SM captures floodplain inundation magnitudes quite well[34–36].

Second, the climate forcings used to drive E3SM contain uncertainties, especially for precipitation projection[53], even though the forcings used in this study have been downscaled and bias-corrected[70]. The uncertainty of precipitation projection is more critical for wetland projection under SSP126, as precipitation is the dominant driver under this scenario. Using a muti-model ensemble allows us to provide uncertainty bounds in the projection[45,71]. While recognizing uncertainties in the future projections particularly related to precipitation projections, this study provides insights on the relative uncertainty between scenarios by highlighting the more dominant role of temperature vs. precipitation in driving wetland changes under the SSP585 vs. the SSP126 scenarios and the physical basis for such differences. This knowledge allows us to assign more confidence in the projections under the SSP585 scenario than the SSP126 scenario due to the differential uncertainty in their dominant driver.

Third, although our model can simulate both non-permanent and permanent inundation (Supplementary Fig. 16), the permanent wetlands (e.g., surface ponding water, swamp, etc.) may not be captured by the upscaled GLAD dataset. This is because we removed the permanent surface water bodies identified in GLAD dataset from its seasonal surface water to exclude rivers, lakes, and reservoirs. However, the permanent wetlands can be unintentionally removed as well, resulting in underestimated wetland dynamics in the upscaled GLAD dataset. This bias can propagate to the simulated wetland dynamics through parameter calibration. Currently, no method is available to separate permanent wetlands from lakes, rivers, and reservoirs in the permanent surface water bodies. For example, the total area of rivers, lakes, and reservoirs in the study domain estimated with Global Lakes and Wetlands Database[72] is about 500,000 [km²], which is much higher than the total permanent surface water detected by GLAD (i.e., 280,000 [km²]).

Fourth, the wetland projections may underestimate wetland extent because our model simulates wetlands that formed by inundation processes while wetlands may also form due to soil saturation or shallow groundwater level. However, we note the sensitivity of soil saturation wetlands can be implicitly inferred by our analysis because surface water has been found to be closely related groundwater dynamics[20,73]. In addition, our model may underestimate flooded forest wetlands or other wetland types that cannot be observed by satellite datasets, which are significantly impacted by cloud, dense forest, shadows, etc[15,74]. This will result in inevitable uncertainty in model parameters calibrated using satellite data. Lastly, our model only considers the responses of wetland to climate change in the future, as direct disturbances from human activities are not explicitly represented in our model. As a result, our estimated wetland changes do not account for additional wetland loss that may be caused by urbanization and agriculture expansion[75] if wetland conversion is not regulated in the future.

## Methods
### Wetland definitions
In this study, we define 'wetland area' as a region within a computational cell inundated due to either pluvial or fluvial processes,

excluding rivers, lakes, and reservoirs, while lakes can be classified as wetland in other study[76]. Therefore, wetland area represents the area of a grid cell that covered by both non-permanent and permanent inundated water. Such a definition of wetland area is consistent with the surface water dynamics observed by satellites with the rivers, lakes, and reservoirs removed.

The definition of "wetland habitat" is based on expert understanding that although wetland environments (e.g., emergent wetlands) do not need to be covered by water permanently, they have to be inundated for at least 1 month during the growing season to develop suitable biotic characteristics. Therefore, we define wetland habitat as the maximum wetland area during the growing season resolved on a monthly basis (growing season represents months with surface temperature higher than 5 °C). Such a definition of wetland habitat is consistent with emergent wetland, which is a transitional area between permanently wetland and dry environments. Note that not all wetland areas can become wetland habitats, i.e., a fraction of the computational cell becomes inundated only during the months outside of the growing season period. We note our definition of wetland habitat doesn't include wetlands that formed by soil saturation (e.g., no inundation occurs). In addition, the criteria for the wetland habitats, such as maximum wetland area for the months with surface temperature higher than 5 °C, may further introduce uncertainties to our analysis.

## Model description

E3SM, a state-of-the-science ESM, is used to simulate wetland dynamics and project their future changes. E3SM is a fully coupled ESM with the atmosphere, land, ocean, land ice, sea ice, and river components. E3SM version 2 is used here, and detailed model description and validation cases are provided in Ref. 43 In this study, the wetland area dynamics in E3SM are simulated as the sum of pluvial and fluvial inundation mechanisms in the land and river components briefly described below.

The E3SM land model (ELM) was developed based on the Community Land Model 4.5 (CLM4.5)[77]. Surface water storage component was introduced to simulate the pluvial inundation process and to store excess rainfall, runoff, and snowmelt (See Supporting Information). The simulated inundation in ELM is controlled by surface-subsurface interactions (i.e., infiltration and water excess). Due to the typically coarse resolution of ESM simulations, a sub-grid scale scheme is implemented to include topographic impacts on different processes (Supplementary Text 1). While a fraction of a grid cell can be covered by snow and/or water, the infiltration rate is assumed to be constant across the entire grid cell (Supplementary Fig. 7). The infiltration capacity ($q_{infl,max}$) is formulated as:

$$q_{infl,\max} = (1-f_{sat})\Theta_{ice}k_{sat},\qquad(1)$$

where $\Theta_{ice}$ represents the ice impedance factor to include the presence of ice in the soil, $k_{sat}$ is the saturation hydraulic conductivity, and $f_{sat}$ is the saturated area fraction, which is determined by the topographic characteristic and water table depth:

$$f_{sat} = f_{\max}\exp(-0.5\times f_{over}\times Z_{wt}),\qquad(2)$$

where $f_{\max}$ is the maximum saturated fraction, $Z_{wt}$ is water table depth $[m]$, and $f_{over}$ is a decay factor $[m^{-1}]$ determining how water table depth controls area saturation fraction. In the default ELM configuration, $f_{over}$ is set to be 0.5 $[m^{-1}]$ for all grid cells. The underlying assumption is there is no infiltration for the saturated area fraction.

The original infiltration scheme can result in unrealistic small surface water inundation when the soil temperature is above freezing and infiltration in the surface water storage is overestimated (Supplementary Fig. 7). The overestimation of infiltration in surface water

storage occurs because a uniform infiltration rate is applied over the whole grid cell despite different surface conditions (snow, surface water, floodplain, open soil) being present. However, infiltration below the surface water should be much smaller than for soil that is not covered by water[78]. Therefore, a sub-grid infiltration scheme has been developed to improve the realism of modeling wetland inundation in ELM. To constrain the infiltration from surface water storage, the saturation fraction is assumed to overlap with the surface water fraction rather than being uniformly distributed in different fractions (Supplementary Fig. 7). Specifically, the infiltration capacity of the surface water fraction ($q_{infl,\max}^{h2osfc}$) and other areas ($q_{infl,\max}$) is estimated according to the following two potential situations (Supplementary Fig. 7):

$$\begin{aligned} q_{infl,\max}^{h2osfc} &= (1-f_{sat})\Theta_{ice}k_{sat} \\ q_{infl,\max} &= \Theta_{ice}k_{sat} \end{aligned}\qquad, f_{h2osfc} > f_{sat}\qquad(3)$$

$$\begin{aligned} q_{infl,\max}^{h2osfc} &= 0 \\ q_{infl,\max} &= \left(1-\frac{f_{sat}-f_{h2osfc}}{1-f_{h2osfc}}\right)\Theta_{ice}k_{sat} \end{aligned}\qquad, f_{h2osfc} \le f_{sat}\qquad(4)$$

where $f_{h2osfc}$ is the ELM-simulated inundation. If the infiltration rate is larger than the available capacity in the soil, the excess infiltrated water will be discharged to surface water storage and becomes standing surface water. We note $f_{h2osfc}$ can capture both periodic (i.e., non-permanent) and permanent inundation, as the sink terms of the surface water storage (e.g., evaporation, infiltration, and outflow) may not always be larger than the source terms (e.g., rainfall, surface runoff, and subsurface discharge).

Model for Scale Adaptive River Transport (MOSART)[79] is the river component of E3SM. It routes freshwater from the land to the ocean through river networks. Specifically, MOSART has a subgrid structure for routing flows over hillslopes, tributaries, and main channels. It uses kinematic or diffusive wave equations. Ref. 35 implemented a macro floodplain inundation scheme with MOSART to simulate inundation dynamics on floodplains when streamflow exceeds the channel capacity. The macroscale floodplain inundation scheme uses the relationship between the flood water volume and inundated area to simulate the riverine inundation dynamics. This volume-area relationship is described by the surface elevation distribution (e.g., at spatial resolution of 90 m) within the computational unit, assuming that riverine inundation propagates from lower elevations to higher elevations[34]. For example, one can estimate the floodplain inundation fraction ($f_{fp}$) given the excess volume ($V_{excess}$, the total river channel volume minus river channel capacity) in the main channel:

$$f_{fp} = F(V_{excess}),\qquad(5)$$

where $F()$ represents the volume-area relationship of the floodplain (e.g., elevation profile), derived from the Cumulative Density Function (CDF) of finer resolution sub-grid elevations.

## Model configuration

We ran simulations using E3SM version 2 with active ELM and MOSART modules over the North American Land Data Assimilation System (NLDAS) domain at a spatial resolution of 0.125° × 0.125° (i.e., ~12.5 km × 12.5 km), including continental United States (CONUS), southern Canada, and northern Mexico (i.e., 25°–53° North). The hourly meteorological forcing of NLDAS phase 2 (NLDAS-2) is used to drive ELM from 1979 to 2020 to obtain model outputs for calibration (see below a description of the calibration procedure). The time steps for ELM and MOSART are 30 min and 60 min, respectively, with a coupling frequency of 180 min between the two models. The default 0.125° × 0.125° ELM surface parameters[80] for the NLDAS domain was

used. The topographic parameters (i.e., flow direction, river length, slope, etc.) of MOSART were generated by the Dominant River Tracing algorithm;[81] and the spatiotemporally varying Manning's roughness coefficients of hillslope, subnetwork, and main channel are estimated online based on land cover and water depth at each time step[37]. Further, the relationship of Eq. (4) was derived from the 90 m-resolution Digital Elevation Model (DEM) from Hydrological Data and Maps Based on Shuttle Elevation Derivatives at Multiple Scales (HydroSHEDS)[82].

Future ELM-MOSART simulations were driven by bias-corrected and downscaled CMIP6 climate forcings[70] (i.e., precipitation, temperature, humidity, radiation, etc.) that archived in the Inter-Sectoral Impact Model Intercomparison Project phase 3b (ISIMIP3b) at 0.5° × 0.5° (i.e., ~50km × 50km) and daily scale. As ELM requires sub-daily inputs, we followed the procedure of https://vic.readthedocs.io/en/vic.4.2.d/Documentation/ForcingData/ to disaggregate the daily forcing to a sub-daily scale, and used the method from Ref. 83 to determine the timing of daily maximum and minimum temperatures. Atmospheric forcings from five climate models for two greenhouse gas emissions scenarios, SSP126, and SSP585, were used for drive ELM-MOSART coupled simulation to project the future wetland changes: GFDL-ESM4, IPSL-CM6A-LR, MPI-ESM1-2-HR, MRI-ESM2-0, and UKESM1-0-LL. We prescribed the land use land cover changes in the future simulation using the projections are from the Land-Use Harmonization LUH2[84].

## Dataset of surface water dynamics

We used the global surface water dynamics dataset from GLAD[15] as observational benchmark dataset in this study. GLAD provides global monthly, annual, and seasonal (i.e., monthly averaged) surface water and permanent water data derived from Landsat images taken during 1999–2020. As there are many data gaps in observations at the monthly scale, seasonal surface water data were used for calibration and annual surface water data were used to validate the model. The original spatial resolution of 30 m × 30 m was upscaled to a model resolution of ~12.5 km × 12.5 km by averaging the values of the finer resolution grid cells within the coarse model resolution grid cell for comparison with the model simulations. The coarse grid cell was assigned to "No data" if over 20% of the finer grid cells within the coarse grid cell contained gaps. Although rivers, lakes, and reservoirs are permanent surface water bodies, they represent different ecosystems than wetlands. We removed permanent water from the upscaled seasonal and annual GLAD surface water to exclude rivers, lakes, and reservoirs. However, permanent wetlands (e.g., surface water ponds and swamp, etc.) can be unintentionally removed as well, which may result in underestimation in the upscaled GLAD surface water dynamics. In addition, GLAD dataset cannot capture the wetlands formed due to soil saturation, which are abundant for some regions (e.g., Supplementary Fig. 17).

## Calibration procedure

Fluvial inundation (simulated in MOSART) process is relatively well represented in ESM since 90 m resolution DEM is used to capture the floodplain storage effects[34,35]. However, the pluvial inundation (simulated in ELM) process adopts constant values for some parameters without justification[12,77]. Calibration is necessary to constrain the parametric uncertainty since sub-grid parameterization are needed to compensate for the typical coarse resolution of ESMs. However, the satellite observations used to validate wetland dynamics cannot differentiate between fluvial and pluvial inundation. Therefore, model simulations that estimate the sum of the two inundations are used to compare and/or calibrate the model against observations.

The following two ELM parameters that control the pluvial inundation process were selected for model calibration: (1) $f_c$ determines the potential maximum inundated area in ELM and (2) $f_{over}$ affects $f_{sat}$ and constrains the infiltration rate under the inundated area. We ran

coupled ELM-MOSART simulations with 100 parameter values randomly sampled from a uniform distribution of $f_c$-$U$[0.001 0.4] and $f_{over}$-$U$[0.1 5], while the default values of other parameters were used. At each grid cell, the parameter set that maximized NSE[85] was identified as the best parameter. The NSE is given as

$$NSE = 1 - \frac{\sum_{i=1}^{12} \left(y_i^{GLAD} - \left(y_i^{Pluvial} + y_i^{Fluvial}\right)\right)^2}{\sum_{i=1}^{12} \left(y_i^{GLAD} - \overline{y^{GLAD}}\right)^2}, \tag{6}$$

where $y_i^{GLAD}$ represents the seasonal surface water fraction from the GLAD dataset, $y_i^{Pluvial}$ is the ELM-simulated surface water inundation, $y_i^{Fluvial}$ is the MOSART-simulated floodplain inundation, and $i$ is the month index. Both the GLAD benchmark and model simulated inundation in the above equation are monthly averaged values from 1999 to 2020. Other satellite datasets[14] can be used as benchmark for calibration as well, though only GLAD was used in this study.

The calibrated parameters can be found in Supplementary Fig. 18. The annual GLAD surface water dynamics are further used to evaluate the calibrated simulation in terms of annual variability and changing trends.

## Attribution analysis

Surface water ($Sw$), temperature ($T$), and precipitation ($P$) are aggregated to the watershed scale with the Hydrologic Unit Codes 8 (HUC 8)[86] within the contiguous US and the Canadian National Hydrographic Network Index in Canada (https://www.nrcan.gc.ca/science-and-data/science-and-research/earth-sciences/geography/topographic-information/geobase-surface-water-program-geeau/watershed-boundaries/20973). The dominant driver for the changes in the simulated surface water ($\Delta Sw$) in the future is determined to be either the change of temperature ($\Delta T$), change of precipitation ($\Delta P$), or both factors. Due to the strong correlation between $\Delta T$ and $\Delta P$, we implemented the following procedure to identify the dominant factor for a given location:

1. For each simulation driven by the atmospheric forcing of a climate model, the annual time series of $\Delta Sw$, $\Delta T$, and $\Delta P$ were derived by subtracting the averaged $Sw$, $T$, and $P$ during the control period (1971–2000) from the future time series (2015-2100).
2. For each simulation of $\Delta Sw$, $\Delta T$, and $\Delta P$, their 10-year moving average series were calculated and the multi-model means of $\Delta Sw$, $\Delta T$, and $\Delta P$ with equal weights for the climate models were then computed.
3. Two linear least-squares regression models using the multi-model ensemble means of $\Delta Sw$, $\Delta T$, and $\Delta P$ were developed: $\Delta Sw = \beta_0 + \beta_1 \times \Delta T$, $\Delta Sw = \beta_0' + \beta_1' \times \Delta P$. The corresponding correlation coefficients: $\rho$ ($\Delta Sw$ $vs.$ $\Delta T$), and $\rho'$ ($\Delta Sw$ $vs.$ $\Delta P$) were determined.
4. If $\rho > \rho_{threshold}$, $\rho' > \rho_{threshold}$, $\beta_1 < 0$, and $\beta_1' > 0$, then $\Delta Sw$ is controlled by both $\Delta T$ and $\Delta P$.
5. If step 4 is false, the location with $\rho > \rho_{threshold}$ and $\beta_1 < 0$ is temperature controlled. If $\rho' > \rho_{threshold}$ and $\beta_1' > 0$, then the location is precipitation controlled.
6. If both conditions in step 5 are false, a multilinear regression model was developed: $\Delta Sw = \beta_0'' + \beta_1'' \times \Delta T + \beta_2'' \times \Delta P + \beta_4'' \times \Delta T \times \Delta P$, and the correlation coefficient, $\rho''$ ($\Delta Sw$ $vs.$ $\Delta T \times \Delta P$), is determined. If $\rho'' > \rho_{threshold}$, then $\Delta Sw$ is controlled by the interactions between $\Delta T$ and $\Delta P$. Otherwise, $\Delta Sw$ cannot be explained by $\Delta T$ or $\Delta P$.

In this study, we select $\rho_{threshold} = 0.5$. We note the correlation coefficients were calculated for each basin with annual time series of $\Delta Sw$, $\Delta T$, and $\Delta P$. Such spatial (basin) and temporal (annual) averaging

significantly reduces the spatial and temporal autocorrelation in the samples being analyzed.

## Data availability

The surface water dynamics dataset of GLAD was downloaded from https://glad.umd.edu/dataset/global-surface-water-dynamics. The five atmospheric forcings used to drive E3SM were retrieved from ISIMP3b (https://data.isimip.org/). NLDAS historical forcing is available at https://disc.gsfc.nasa.gov/datasets?keywords=NLDAS. The National Land Cover Dataset can be downloaded from https://www.usgs.gov/centers/eros/science/national-land-cover-database#data. The simulation results used for plotting the figures in this study have been deposited in Zenodo at https://doi.org/10.5281/zenodo.10099224.

## Code availability

The E3SM code with the improved pluvial inundation process is deposited in Zenodo at https://doi.org/10.5281/zenodo.6982264. Instructions of running E3SM can be found at: https://e3sm.org/model/running-e3sm/e3sm-quick-start/ (last access: Nov 2023). The scripts to process and analyze the simulation results have been deposited in Zenodo at https://doi.org/10.5281/zenodo.10095326.

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

## Acknowledgements

This work was supported by the  U.S. Department of Energy (DOE), Office of Science, Biological and Environmental Research (BER) through the Earth System Model Development (ESMD) and Regional and Global Modeling Analysis (RGMA) program area as part of the multi-program, collaborative integrated Coastal Modeling (ICoM) project (grant no. KP1703110/75415) and Energy Exascale Earth System Model (E3SM) project. D.X., G.B., Z.T. and T.Z. acknowledge support by the ICOM-ESMD. L.R.L. acknowledges the support by the ICoM-RGMA. E.S. and A.D. acknowledge the support by the E3SM. The Pacific Northwest National Laboratory is operated by Battelle for the U.S. Department of Energy under Contract DE-AC05-76RLO1830. V.Y.I. acknowledges the support of NSF CMMI grant 2053429. We would like to acknowledge Dr. Patrick Megonigal for his valuable feedback on the preliminary results and discussion on wetland habitats.

## Author contributions

D.X., G.B. and Z.T. designed the study. D.X. performed the simulations and analysis and wrote the initial manuscript under the mentorship of G.B. A.D. processed the future land use land cover projections. E.S., A.D., T.Z., V.I., and R.L. investigated the results. D.X., G.B., Z.T., E.S., A.D., V.I., and R.L. contributed to the results discussion, review, and manuscript writing.

## Competing interests

The authors declare no competing interests.
