## [Peer Review File · Nature Communications]

Climate change will reduce North American inland wetland areas and disrupt their seasonal regimesREVIEWER COMMENTS

Reviewer #1 (Remarks to the Author):

Thank you for the opportunity to review the "Climate change will reduce wetland areas and disrupt their seasonal regimes in North America " manuscript. The manuscript looks at factors both fluvial and pluvial that drive wetland hydrology and use physically based earth system models to understand inland wetland trajectory and response to climate change under two emissions scenarios. The paper is well written and the topic is certainly important. However, several methodological questions would need to be addressed and the discussion section substantially improved before this deserves publication.

Abstract better to end w a recommendation

Para at 56-58 needs a REF at the end

The case for statistical models based on empirical data and the argument that Physically based Earth system models (ESMs) are the only way to go seems very unbalanced as an argument. Please add a paragraph on the pros and cons of both. For example the authors say that complex relationships between ppt and temp and wetlands cannot be modeled with empirical data, but never acknowledge that using machine learning techniques we can certainly model those complex relationships with empirical data, in cases better than with ESMs.

How were the correlations between wetland area change and ppt and temp calculated at line 215? Across the us over time? If so this is not correct, the spatial and temporal autocorrelation should be accounted for.

The discussion section, the way it's written is mainly a summary of the same results presented in the Results section. You need to integrate and compare and contrast with other studies in this section.

The GLAD dataset was used as an observational benchmark but the authors do not seem to mention anything about the accuracy of this dataset (and thus error propagation) and the fact that GLAD only looks at open water rather than wetlands. How do the authors define wetland for the purpose of this work?

Reviewer #2 (Remarks to the Author):

This study is based on the integration of a new modeling scheme of wetlands in a hydrological model to simulate surface water extent of wetlands. It is used to predict wetland areas until the end of the 21st century in North America.

In my opinion, a thorough validation of the new model parametrization of the flood model is necessary. The validation part of this new parameterization is currently achieved using a single dataset and only performed on a normalized time series. I would have suggested to add other commonly used flood extent datasets such Global Water Surface or Global Inundation Extent from Multi-Satellite and to present real wetland extents to be sure that not only the spatial patterns and, more or less, the temporal variations are consistent, but also the real surface water extent are well simulated. Is this first step is not well validated (I think it should be an article in itself to help the reader to be confident in the results presented in this manuscript), all the climate simulations cannot be trusted.

Reviewer #3 (Remarks to the Author):

The authors use an Earth system model to project inland wetland conditions of North America under

the impact of climate change. The study finds that the annual wetland area will decrease by 10% at a continental scale by the end of the century, while the regional wetlands will vary substantially. Moreover, wetland areas will shrink most in summer due to peak biotic processes in the higher emission scenario.

The manuscript is well written. However, I find a lot of uncertainties in wetland definition, modeling, and the related results. Results of wetland change prediction based on future predicted factors, e.g. precipitation, have too many uncertainties.

1. GLAD appears first in line 95 but the full name appears first in line 471.
2. The definition of wetland seems highly related with open surface water. However, wetlands include forested wetlands, emergent wetlands where no open water can be seen from the satellite images.
3. Lines 139-141: "Specifically, the median annual wetland area in 2071-2100 decreases by 5.2% and 10.6% under SSP126 and SSP585, respectively, relative to the historical period (1971-2000)". The 5.2% and 10.6% were calculated based on historical data during 1971-2000. 5.2% and 10.6% are not very large percentages. Why do you select 1971-2000 as the period for comparison? What are the percentages if you select 1991-2020? What are the uncertainty numbers/percentages?
4. Fig. 2, why the winter wetland areas are more than historical wetland areas? Why winter is different from Spring, Summer, and Fall? If winter of 2071-2100 has more wetlands, can we say the wetlands in 2071-2100 area shrank?
5. Lines 216-218, "Notably, wetland area is only sensitive to one climatic factor in certain regions, such as precipitation changes in the southeastern US under SSP126 (Fig.S8c and e)". Future temperature is likely to increase because of global warming, co2 emission. However, the precipitation change is harder to predict, which will introduce uncertainty to the prediction of future wetland change, which is a big problem because precipitation is a dominant factor for most of the regions under SSP126 (Fig. 5a).
6. Lines 356-358, "Consequently, the wetlands of Florida are projected to expand under SSP126 but shrink under SSP585. Notably, wetlands from cold regions are only sensitive to temperature changes under both SSP126 and SSP585 (e.g., southeastern Canada)." In fact, Florida has the most wetland loss among all U.S. states over the past decades due to large population gain and urbanization. The trend is likely to continue as the population keeps growing.

Reviewer #1 (Remarks to the Author):

Thank you for the opportunity to review the “Climate change will reduce wetland areas and disrupt their seasonal regimes in North America ” manuscript. The manuscript looks at factors both fluvial and pluvial that drive wetland hydrology and use physically based earth system models to understand inland wetland trajectory and response to climate change under two emissions scenarios. The paper is well written and the topic is certainly important. However, several methodological questions would need to be addressed and the discussion section substantially improved before this deserves publication.

Response: We appreciate the reviewer for recognizing the merits of our work. Your comments are important and very helpful for improving the manuscript, and we have carefully addressed them in the revised manuscript.

Abstract better to end w a recommendation

Response: Yes, we added some statements in the abstract to emphasize the vulnerability of wetland to climate change under the high emission scenario. At the end of the abstract, we recommend lowering emissions to a moderate level is helpful to protect wetland ecosystem.

Para at 56-58 needs a REF at the end

Response: We rewrote this sentence and added relevant references at line 57 – line 67 in the revised manuscript as the following:

“Given the complexity and nonlinearity of the climate system and related hydrological responses (Beven, 2001; Rial et al., 2004), machine learning could be useful for deriving the statistical relationships between wetlands dynamics and their hydroclimatic drivers (Gaines et al., 2022). However, the statistics derived from machine learning or other data-driven methods may not be extended to the warmer climate in the future as they are based on historical observations but wetlands are sensitive to nonstationary climate (Milly et al., 2008)(Shen et al., 2021). Besides, the accuracy of machine learning derived statistics depends on the quantity and quality of training data, which are still limited for surface water dynamics. Although numerous studies derived surface water dynamics from satellite dataset at regional or national scales (Jones, 2019; Mueller et al., 2016; Tulbure and Broich, 2013; Tulbure et al., 2016; Zou et al., 2018), such data are limited at continental and global scales, and the spatial coverage at sub-annual scale is very poor (Fig.S1).”

The case for statistical models based on empirical data and the argument that Physically based Earth system models (ESMs) are the only way to go seems very unbalanced as an argument. Please add a paragraph on the pros and cons of both. For example the authors say that complex relationships between ppt and temp and wetlands cannot be modeled with empirical data, but never acknowledge that using machine learning techniques we can certainly model those complex relationships with empirical data, in cases better than with ESMs.

Response: We thank the reviewer for the constructive comment and insightful perspective. We agree that Machine Learning (ML) is a useful tool that can capture the complex relationship between wetland dynamics and the hydrometeorological drivers, and may outperform ESMs in capturing those complex relationships under present climate provided enough data is used for

training (Gaines et al., 2022). But ML approaches or data-driven approaches have the following limitations for climate change studies:

1. Although ML approaches can capture complex, hidden relationship well, they can be problematic when extrapolating outside of the training dataset (Shen et al., 2021). Under climate change with changes in both temperature and precipitation, some ML approaches may fail to capture the wetland changes under an evolving climate. As shown in our attribution analysis, the major driver of wetland trends changes from precipitation dominant under the historical period and SSP126 scenario to temperature dominant under the SSP585 scenario (Fig. 5 in main text). Such driver change may not be learned by ML approaches trained using historical observations, thus potentially yielding unreliable wetland projections.
2. Training and validating an ML model requires a lot of data. Unfortunately, the satellite derived surface water observation has very limited spatial coverage, also resolved at monthly intervals, as shown by Figure R1. In addition, such satellite observation is only available over the past two decades, which is a relatively short period to span the wetland variations and changes due to climate variability and change.

Figure R1. Monthly GLAD surface water fraction [-] of 2010. The white area represents grid cell that does not have a value.

In contrast, physically based ESMs are more suitable for future projection when multiple drivers of wetland dynamics change concurrently. However, physics-based ESMs can be subject to biases resulting from process simplification and parametric uncertainty, which will lower our confidence on their projections. To address these limitations, we improved the model performance by modifying the infiltration scheme and calibrating the relevant parameters. And we further improved the projection confidence by running our model with an ensemble of atmospheric forcing corresponding to future projections produced by different climate models and socioeconomic scenarios. However model calibration and ensemble simulations are computationally intensive, requiring substantial computational resource that are not readily available to systematically explore different simulation settings.

We added the pros and cons for both Machine learning/data-driven method and physics-based ESMs in the introduction of the revised manuscript (line 57-line 93).

How were the correlations between wetland area change and ppt and temp calculated at line 215? Across the us over time? If so this is not correct, the spatial and temporal autocorrelation should be accounted for.

Response: We did account for the spatial and temporal autocorrelation when computing the correlations between wetland area change and their drivers change. First, the temporal correlations between the wetland area change and the two drivers (precipitation and temperature) were calculated at the basin scale using the annual time series. Then, the correlation values were averaged across all basins and reported as 0.76 and 0.86. Such spatial (basin) and temporal (annual) averaging significantly reduces the spatial and temporal autocorrelation in the samples being analyzed.

We have updated the Methods and Material section (line 608-612) to clarify the computation of correlation values.

The discussion section, the way it's written is mainly a summary of the same results presented in the Results section. You need to integrate and compare and contrast with other studies in this section.

Response: We thank the reviewer for the comments. To the best of our knowledge, most studies focus on the wetland changes in the past, with only a few focusing on projecting the wetland changes in the future at large scales, but based on simplified wetland scheme. In the revised manuscript, we added additional discussions to compare our conclusions to previous studies. Specifically, our results are consistent with previous studies that wetland ecosystems will be vulnerable to climate change. However, we found a different response of wetland to global warming over the cold regions compared to previous studies that ignored the generation process of wetlands. Please find our modifications in the Discussion section of the revised manuscript.

The GLAD dataset was used as an observational benchmark but the authors do not seem to mention anything about the accuracy of this dataset (and thus error propagation) and the fact that GLAD only looks at open water rather than wetlands. How do the authors define wetland for the purpose of this work?

Response: Thanks for pointing this out. We acknowledge that GLAD has been carefully validated against other commonly used satellite-based surface water dynamics products in previous study (Pickens et al., 2020). We clarify this in the introduction at line 101 – line 102. However, while the uncertainty of GLAD is inevitable, it is one of the best and most comprehensive global surface water dynamics products to the best knowledge of the authors. The uncertainty of GLAD can propagate to the model projections through the calibrated parameter values. We list this as one of the uncertainty sources in the revised manuscript at line 437 – line 439.

As suggested by Reviewer#2, we further validated our model against the Global Inundation Extent from Multi-Satellite (GIEMS). Figure R2 shows our model can capture the spatial pattern, monthly temporal variation, and importantly, the annual trend well. Please note, the data period of GIEMS is 1993-2007, including multiple years that are not in GLAD (i.e., GLAD is available for 1999-2020).

The reviewer is right that GLAD only looks at open water rather than wetlands that do not always need to be covered with water. Therefore, we defined **wetland area** as the sum of areas subjected to pluvial inundation and fluvial inundation, which reflects the temporal surface water dynamics. So, wetland area as we defined is consistent with GLAD, supporting its use in calibrating the simulated **wetland area**. We also acknowledge that a site can be recognized as wetland even without being covered with water. After consulting with a wetland scientist with rich field experience (Dr. Patrick Megonigal), a site can develop wetland characteristics if inundated persistently over one month during the growing seasons in a year. Therefore, we further define **wetland habitat** as areas that have been inundated for at least one month during the growing seasons in a given year. We provided the definitions in the introduction in the original submission, and we further highlight and clarify our definitions in the revised manuscript at line 451 – line 463.

Our definition of wetlands represents the wetland formed due to inundation process. However, wetland may also be formed by soil saturation rather than inundation process and our model cannot simulate this kind of wetland. We added this as one of the uncertainties at line 439 – line 441.

Reviewer #2 (Remarks to the Author):

This study is based on the integration of a new modeling scheme of wetlands in a hydrological model to simulate surface water extent of wetlands. It is used to predict wetland areas until the end of the 21st century in North America.

In my opinion, a thorough validation of the new model parametrization of the flood model is necessary. The validation part of this new parameterization is currently achieved using a single dataset and only performed on a normalized time series. I would have suggested to add other commonly used flood extent datasets such Global Water Surface or Global Inundation Extent from Multi-Satellite and to present real wetland extents to be sure that not only the spatial patterns and, more or less, the temporal variations are consistent, but also the real surface water extent are well simulated. If this first step is not well validated (I think it should be an article in itself to help the reader to be confident in the results presented in this manuscript), all the climate simulations cannot be trusted.

Response: We thank the reviewer for the comments that help us improve our study. We agree that a thorough validation of our model is important to improve the projection confidence. The Energy Exascale Earth System Model (E3SM) is a state-of-the-science Earth System Model (ESM) that participated in CMIP6 and compared against other CMIP-class models. Its water cycle simulation has been validated in several previous studies (Golaz et al., 2019; Xu et al., 2021; Zhang et al., 2022).

In the revised manuscript, we further validated our model against the Global Inundation Extent from Multi-Satellite (GIEMS) dataset for 1993-2007, which includes data that is outside the model calibration period (1999-2020). Specifically, our model is able to capture the spatial variation (Fig. R2a-b) and temporal dynamics at both annual (Fig. R2c) and monthly scale (Fig. R2d). Our model captures the positive trend of wetland area during 1999-2020 (Fig. 1c in the main text), as well as the negative trend of wetland area during 1993-2007 (Fig R2c). We note that GIEMS represents a better product to validate temporal variation at monthly scale as it has a better spatial coverage than GLAD. However, GIEMS includes permanent water bodies, cannot detect small water bodies (due to its relatively coarse spatial resolution), and cannot separate saturated soil and surface water (Prigent et al., 2007). Therefore, we normalized both the GIEMS data and the model simulation for validation of the monthly and annual time series. We have added the new results on model validation against GIEMS at line 121 – line 124 of the revised manuscript and added Fig R2 in the supplementary materials.

Surface water observations at high spatial resolution (e.g., <1km) are relatively limited, and the spatial coverage of GLAD is low at the sub-annual scale (Figure R1, responses to Reviewer#1), which can result in bias in magnitudes. Therefore, we didn't evaluate our simulated wetland areas against GLAD for the monthly temporal variation. The annual GLAD surface water has better spatial coverage, but its magnitude can also be affected by the limited spatial coverage of monthly GLAD surface water. Therefore, we normalized the dataset at annual scale to focus on the comparison of trend between the GLAD and simulated surface water, which is important to show if our model can capture the sensitivity of wetland to climate forcings.

As the reviewer mentioned, Global Surface Water (GSW) is another comprehensive global surface water dataset at high spatial resolution. However, we didn't validate our model with this

dataset because: (1) The GLAD dataset has been carefully benchmarked against GSW (Pickens et al., 2020) and both GLAD and GSW were derived from the same set of observations (Landsat satellite); and (2) Due to the high spatial resolution, the spatial coverage of GSW is also limited to monthly scale (Figure R3).

Figure R2. (a) Averaged GIEMS inundated fraction from 1993 to 2007; (b) Simulated MOSART and ELM inundated fraction from 1993 to 2007; (c) and (d) illustrates the continent-averaged normalized wetland size (subtracting the mean and divided by the standard deviation) comparison between the simulation and GIEMS at annual and monthly scales, respectively. ρ in subplot (b) and (d) represents correlation coefficient.

Figure R3. Monthly Global Surface Water (GSW) surface water fraction [-] of 2010. The white area represents grid cells that do not have a value.

Reviewer #3 (Remarks to the Author):

The authors use an Earth system model to project inland wetland conditions of North America under the impact of climate change. The study finds that the annual wetland area will decrease by 10% at a continental scale by the end of the century, while the regional wetlands will vary substantially. Moreover, wetland areas will shrink most in summer due to peak biotic processes in the higher emission scenario.

The manuscript is well written. However, I find a lot of uncertainties in wetland definition, modeling, and the related results. Results of wetland change prediction based on future predicted factors, e.g. precipitation, have too many uncertainties.

Response: We appreciate the reviewer for the detailed comments that help us improve our manuscript. In the revised manuscript, we highlight our definition of wetland and add more discussions on the uncertainty due to precipitation. Please find our point-to-point responses in the following.

1. GLAD appears first in line 95 but the full name appears first in line 471.

Response: Thank you for catching this. In the revised manuscript, we specify the full name of GLAD and provide a brief description of this dataset at line 100 – line 102.

2. The definition of wetland seems highly related with open surface water. However, wetlands include forested wetlands, emergent wetlands where no open water can be seen from the satellite images.

Response: The reviewer has brought up an important point about the more general wetland definition and our narrower definition that **wetland area** is the sum of pluvial inundation and fluvial inundation. We clarified at line 50 that surface water is used interchangeably with wetland in this study. Since GLAD observes surface water, our definition of wetland area in the model allows consistent use of GLAD for model calibration. We also define **wetland habitat** to be an area that is inundated at least one month during the growing season in a year because we acknowledge that a site does not need to be covered with water all the time to derive wetland characteristics. This definition of wetland habitat includes emergent wetlands, which is defined as “a transitional area between permanently wet and dry environments” (i.e., the definition given by National Park Service at <https://www.nps.gov/ocmu/learn/historyculture/upload/Accessible-Emergent-Wetlands.pdf>). Overall, our model aims to simulate wetlands that formed due to inundation process (caused by either surface water or subsurface water) but cannot capture the wetlands that are formed due to soil saturation (e.g., no inundation - standing surface water occurs). We added this as a limitation at line 439 – line 441.

The definitions of wetland area and wetland habitat are clarified in the revised manuscript at line 450 – line 463.

Indeed, satellite cannot detect surface water under dense forest, as argued in Pickens et al. (2020): “It is unknown how much surface water is left undetected due to being under forest cover or other vegetation obscuring the surface of the water from above.”. However, our projections over North America are less affected by the uncertainty of forested wetlands because most forested wetlands are located in tropical rainforests (Lehner and Döll, 2004). Although there can

exist forested wetlands in the lower Mississippi, this type of wetland is not very common in continental U.S. according to Global Lakes and Wetlands Database (GLWD, Figure R4). For example, GLWD identified no forested wetlands in continental U.S. Further, we would like to note that our model can simulate forested wetlands that are formed due to inundation processes, even if they are not well identified by satellites, which will impact the parameter during calibration process. We included this uncertainty in the discussion of limitations at line 441 – line 443.

Figure R4. Global Lakes and Wetlands Database over Continental US.

3. Lines 139-141: “Specifically, the median annual wetland area in 2071-2100 decreases by 5.2% and 10.6% under SSP126 and SSP585, respectively, relative to the historical period (1971-2000)”. The 5.2% and 10.6% were calculated based on historical data during 1971-2000. 5.2% and 10.6% are not very large percentages. Why do you select 1971-2000 as the period for comparison? What are the percentages if you select 1991-2020? What are the uncertainty numbers/percentages?

Response: In CMIP6, historical simulations driven by observed forcing end in 2014, and future projections driven by forcing derived from socioeconomic scenarios (i.e., SSP126 and SSP585) start in 2015. To test the sensitivity of the percentage future change to the definition of the historical period while maintaining consistency with the CMIP6 simulations, we selected 1985-2014 as the most recent 30 years to represent the historical period. Specifically, relative to this new historical period, the median annual wetland area in 2071-2100 decreases by 4.3% and 8.0% under SSP126 and SSP585, respectively. Figure R5 shows that the change of future wetland relative to the historical period of 1985-2014, which is consistent with Fig. 2 in the main text.

The relative change of 5.2% and 10.6% represent the changes between the 30-year average of each period, thus the detailed changes in higher spatial and temporal scales are cancelled out to some extent. For example, the continental averaged wetland area during the summer is projected to decrease by about 30% (Figure 2d in the Main text). In additional, the wetland changes at basin scale show divergent directions, with changes that are up to $\pm 50\%$.

We also provided the multi-model ensemble bounds for the changes of continental averaged wetland at line 164 and in the abstract to show the projection uncertainty.

Figure R5. Probability distribution of continent-averaged wetland area (including both pluvial and fluvial inundation) for (a) annual, (b) winter, (c) spring, (d) summer, and (e) fall. Historical, MID, and END denote the historical period (1985-2014), mid-century (2041-2070), and end-of-century (2071-2100), respectively. Each probability distribution function is constructed from the multi-model ensemble. The inset plots show the relative change of wetland size for wetland size at the 25th, 50th, and 75th percentile between each future scenario (FUT) and the historical period (HIS): $\frac{FUT-HIS}{HIS} \times 100\%$. Wetland size is represented as a fraction of total simulation domain area on the x-axis. In (d) and (e), the 5th percentile wetland fraction in the historical period and the 50th percentile wetland fraction at the end of the century under SSP585 are indicated by black square and red circle for comparison.

4. Fig. 2, why the winter wetland areas are more than historical wetland areas? Why winter is different from Spring, Summer, and Fall? If winter of 2071-2100 has more wetlands, can we say the wetlands in 2071-2100 area shrank?

Response: Future winter wetland areas are projected to increase compared to the historical period because of the increased liquid precipitation and snowmelts due to warming effects (Figure R6a). Winter is different from other seasons because the increases in rainfall and snowmelts in Winter are more significant than that in Spring, Summer, and Fall (Figure R6b-d). During the winter, the positive impacts of increased liquid precipitation and snowmelt are larger than the negative impacts of temperature that increases evapotranspiration and infiltration, thus resulting in an overall increase in the wetland areas. However, the positive impacts of precipitation and snowmelt changes during winter are not large enough to compensate for the

negative impacts of temperature increases during other seasons. We added Figure R6 in the supplementary materials and added relevant discussions at line 170 – line 172 in the revised manuscript.

The projected increase in wetland area in winter is less than the projected decrease in the wetland area in Spring, Summer, and Fall. Overall, the annual wetland areas are projected to decrease due to the dominant impacts from Spring, Summer, and Fall months (Figure 2 and line 173 - line 178 in the main text).

Figure R6. Probability distribution function (PDF) of liquid precipitation + snowmelts during (a) Winter, (b) Spring, (c) Summer, and (d) Fall under historical and the two future scenarios. The square scatters in each subplot represent the corresponding median of the PDF.

5. Lines 216-218, “Notably, wetland area is only sensitive to one climatic factor in certain regions, such as precipitation changes in the southeastern US under SSP126 (Fig.S8c and e)”. Future temperature is likely to increase because of global warming, co2 emission. However, the precipitation change is harder to predict, which will introduce uncertainty to the prediction of future wetland change, which is a big problem because precipitation is a dominant factor for most of the regions under SSP126 (Fig. 5a).

Response: We agree with the reviewer that the precipitation projections in climate models are more uncertain than temperature projections, which we noted in the manuscript to provide a fair assessment of the current state of projection uncertainty. Model uncertainty is partly addressed by using a multi-model ensemble to provide a range of future climate projections to infer wetland changes in the future. While recognizing uncertainties in the future projections particularly related to precipitation projections, this study provides useful insights on the relative uncertainty between scenarios by highlighting the more dominant role of temperature (precipitation) in driving wetland changes under the SSP585 (SSP126) scenarios and the physical basis for such differences. This knowledge allows us to assign more confidence in the projections under the SSP585 scenario than the SSP126 scenario due to the differential uncertainty in their dominant driver.

To provide more information about precipitation projection uncertainty, Figure R7 shows the precipitation projections from ISIMIP for the five climate models that are used in our study. Despite differences in magnitude and regional details, the projections are consistent in terms of the broad spatial pattern of precipitation and the corresponding increasing trend in the future under SSP245 scenario.

As the reviewer mentioned, we highlight at line 428 – line 430 that the wetland projection for the SSP126 scenario can be more impacted by the precipitation uncertainty because precipitation represents the dominant driver. We also added more discussion on the precipitation uncertainty at line 426 – line 437.

Figure R7. Spatial pattern of precipitation projection during 2017-2100 from (a). GFDL-ESM4, (b). IPSL-CM6A-LR, (c). MPI-ESM1-2-HR, (d). MRI-ESM2-0, (e). UKESM1-0-LL, and (f) multi-model ensemble mean. Subplot (g) illustrates the annual time series for the continental averaged precipitation. 10-year moving average mean is applied for each time series.

6. Lines 356-358, “Consequently, the wetlands of Florida are projected to expand under SSP126 but shrink under SSP585. Notably, wetlands from cold regions are only sensitive to temperature changes under both SSP126 and SSP585 (e.g., southeastern Canada).” In fact, Florida has the most wetland loss among all U.S. states over the past decades due to large population gain and urbanization. The trend is likely to continue as the population keeps growing.

Response: We thank the reviewer for pointing out the role of human activities on wetland changes. Based on the literature review (line 372 – line 373 in the revised manuscript), we found that population gain and urbanization represent the most dominant factors for wetland loss in the past. In this study, we focused on the natural drivers for the wetland changes associated with global warming. Therefore, all the wetland projections of this study are due to the responses of land surface processes to the change of atmospheric forcings in the future. We further note that there is no wetland unit in the transient land use land cover (LULC) data (Popp et al., 2017) that is used as input to the land model in our simulations. So, it is beyond the scope of this study to evaluate the direct wetland loss due to urbanization or agriculture expansion, but we acknowledge the significant impact of human activities on wetland.

In the revised manuscript, we emphasize that we focus only on the impacts of natural factors on wetlands and mention the direct impacts of human activities to wetland as one of the uncertainties of our wetland projection (line 443 – line 447 in the revised manuscript).

References

- Beven, K. 2001. How far can we go in distributed hydrological modelling? *Hydrol Earth Syst Sc* 5(1), 1-12.
- Gaines, M.D., Tulbure, M.G. and Perin, V. 2022. Effects of Climate and Anthropogenic Drivers on Surface Water Area in the Southeastern United States. *Water Resour Res* 58(3), e2021WR031484.
- Golaz, J.-C., Caldwell, P.M., Van Roekel, L.P., Petersen, M.R., Tang, Q., Wolfe, J.D., Abeshu, G., Anantharaj, V., Asay-Davis, X.S., Bader, D.C., Baldwin, S.A., Bisht, G., Bogenschutz, P.A., Branstetter, M., Brunke, M.A., Brus, S.R., Burrows, S.M., Cameron-Smith, P.J., Donahue, A.S., Deakin, M., Easter, R.C., Evans, K.J., Feng, Y., Flanner, M., Foucar, J.G., Fyke, J.G., Griffin, B.M., Hannay, C., Harrop, B.E., Hoffman, M.J., Hunke, E.C., Jacob, R.L., Jacobsen, D.W., Jeffery, N., Jones, P.W., Keen, N.D., Klein, S.A., Larson, V.E., Leung, L.R., Li, H.-Y., Lin, W., Lipscomb, W.H., Ma, P.-L., Mahajan, S., Maltrud, M.E., Marnett, A., Matusov, A., McClean, J.L., McCoy, R.B., Neale, R.B., Price, S.F., Qian, Y., Rasch, P.J., Reeves Eyre, J.E.J., Riley, W.J., Ringer, T.D., Roberts, A.F., Roesler, E.L., Salinger, A.G., Shaheen, Z., Shi, X., Singh, B., Tang, J., Taylor, M.A., Thornton, P.E., Turner, A.K., Veneziani, M., Wan, H., Wang, H., Wang, S., Williams, D.N., Wolfram, P.J., Worley, P.H., Xie, S., Yang, Y., Yoon, J.-H., Zelinka, M.D., Zender, C.S., Zeng, X., Zhang, C., Zhang, K., Zhang, Y., Zheng, X., Zhou, T. and Zhu, Q. 2019. The DOE E3SM Coupled Model Version 1: Overview and Evaluation at Standard Resolution. *J Adv Model Earth Sy* 11(7), 2089-2129.
- Jones, J.W. 2019. Improved Automated Detection of Subpixel-Scale Inundation—Revised Dynamic Surface Water Extent (DSWE) Partial Surface Water Tests. *Remote Sensing* 11(4), 374.
- Knutti, R., Furrer, R., Tebaldi, C., Cermak, J. and Meehl, G.A. 2010. Challenges in Combining Projections from Multiple Climate Models. *J Climate* 23(10), 2739-2758.
- Knutti, R. and Sedláček, J. 2012. Robustness and uncertainties in the new CMIP5 climate model projections. *Nat Clim Change* 3(4), 369-373.
- Lehner, B. and Döll, P. 2004. Development and validation of a global database of lakes, reservoirs and wetlands. *J Hydrol* 296(1), 1-22.
- Milly, P.C.D., Betancourt, J., Falkenmark, M., Hirsch, R.M., Kundzewicz, Z.W., Lettenmaier, D.P. and Stouffer, R.J. 2008. Stationarity Is Dead: Whither Water Management? *Science* 319(5863), 573-574.
- Mueller, N., Lewis, A., Roberts, D., Ring, S., Melrose, R., Sixsmith, J., Lymburner, L., McIntyre, A., Tan, P., Curnow, S. and Ip, A. 2016. Water observations from space: Mapping surface water from 25 years of Landsat imagery across Australia. *Remote Sens Environ* 174, 341-352.
- Pickens, A.H., Hansen, M.C., Hancher, M., Stehman, S.V., Tyukavina, A., Potapov, P., Marroquin, B. and Sherani, Z. 2020. Mapping and sampling to characterize global inland water dynamics from 1999 to 2018 with full Landsat time-series. *Remote Sens Environ* 243, 111792.
- Popp, A., Calvin, K., Fujimori, S., Havlik, P., Humpenöder, F., Stehfest, E., Bodirsky, B.L., Dietrich, J.P., Doelmann, J.C., Gusti, M., Hasegawa, T., Kyle, P., Obersteiner, M., Tabeau, A., Takahashi, K., Valin, H., Waldhoff, S., Weindl, I., Wise, M., Kriegler, E., Lotze-Campen, H.,

- Fricko, O., Riahi, K. and Vuuren, D.P.v. 2017. Land-use futures in the shared socio-economic pathways. *Global Environmental Change* 42, 331-345.
- Prigent, C., Papa, F., Aires, F., Rossow, W.B. and Matthews, E. 2007. Global inundation dynamics inferred from multiple satellite observations, 1993–2000. *Journal of Geophysical Research: Atmospheres* 112(D12).
- Rial, J.A., Pielke, R.A., Beniston, M., Claussen, M., Canadell, J., Cox, P., Held, H., de Noblet-Ducoudré, N., Prinn, R. and Reynolds, J.F. 2004. Nonlinearities, feedbacks and critical thresholds within the Earth's climate system. *Climatic Change* 65(1), 11-38.
- Shen, C., Chen, X. and Laloy, E. 2021. Editorial: Broadening the Use of Machine Learning in Hydrology. *Frontiers in Water* 3.
- Tebaldi, C. and Knutti, R. 2007. The use of the multi-model ensemble in probabilistic climate projections. *Philos Trans A Math Phys Eng Sci* 365(1857), 2053-2075.
- Tulbure, M.G. and Broich, M. 2013. Spatiotemporal dynamic of surface water bodies using Landsat time-series data from 1999 to 2011. *ISPRS Journal of Photogrammetry and Remote Sensing* 79, 44-52.
- Tulbure, M.G., Broich, M., Stehman, S.V. and Kommareddy, A. 2016. Surface water extent dynamics from three decades of seasonally continuous Landsat time series at subcontinental scale in a semi-arid region. *Remote Sens Environ* 178, 142-157.
- Xu, D., Bisht, G., Sargsyan, K., Liao, C. and Leung, L.R. 2021. Using an Uncertainty Quantification Framework to Calibrate the Runoff Generation Scheme in E3SM Land Model V1. *Geosci. Model Dev. Discuss.* 2021, 1-34.
- Zhang, C., Golaz, J.C., Forsyth, R., Vo, T., Xie, S., Shaheen, Z., Potter, G.L., Asay-Davis, X.S., Zender, C.S., Lin, W., Chen, C.C., Terai, C.R., Mahajan, S., Zhou, T., Balaguru, K., Tang, Q., Tao, C., Zhang, Y., Emmenegger, T., Burrows, S. and Ullrich, P.A. 2022. The E3SM Diagnostics Package (E3SM Diags v2.7): a Python-based diagnostics package for Earth system model evaluation. *Geosci. Model Dev.* 15(24), 9031-9056.
- Zou, Z., Xiao, X., Dong, J., Qin, Y., Doughty, R.B., Menarguez, M.A., Zhang, G. and Wang, J. 2018. Divergent trends of open-surface water body area in the contiguous United States from 1984 to 2016. *Proceedings of the National Academy of Sciences* 115(15), 3810.

REVIEWER COMMENTS

Reviewer #2 (Remarks to the Author):

This study analyzes the impact of projected climate scenarios on wetlands areas in North America during the 21st century. Depending on the magnitude of the change in each scenario, the shrinking of wetland areas will range between 5 and 25% of their current areas. The main driver (temperature or rainfall) is identified for each scenario depending on the location of the wetland. As the drying is expected to occur during summer, consequences on wetland habitats were also estimated. In my previous review, I mentioned the need for the model used here to simulate wetland extent to be validated. The authors successfully achieved this validation. I think this work is important to better understand the consequences of climate change on wetlands that provide many ecosystem services.

Reviewer #4 (Remarks to the Author):

Xu et al.
Climate change and inland wetlands

Xu and coauthors explore inland wetland hydrological dynamics using an Earth systems model and an ensemble approach to climate change through 2100. There's a lot in this paper, and useful information for subsequent analyses. At this juncture, I found myself asking more questions than I found answered by the paper. There are not intractable problems, it is more that the clarity I found necessary for Nature Coms was often missing, leading to some conclusions that might not be completely supported (without caveats). For instance, inland wetlands in this study are limited to inundated systems – but not permanently inundated systems. That's fine for an operational CONUS-scale study, but I'd argue that the repeated use of the word "wetland" or even "inland wetland" is not appropriate. The study is on non-permanently flooded inland surface waters, including wetlands. Further, with that condition on wetlands I wasn't able to discern later if their model runs addressed those currently permanently flooded waters that might become semi-permanently flooded (e.g., dry at least one month but flooded at least one month during the growing season). The use of the word wetland and inland wetland has many implications yet I feel the authors need to provide additional clarity to the readers as to what's meant throughout the entirety of the paper (e.g., did they contrast the GIEMS data with their wetlands after removing the permanently inundated waters?). Further, the study does not address "North America" but rather the conterminous US plus a smidge of Mexico and an area of southeastern Canada that – to me – appears to misbehave. That is, the authors take pains to explain how 'soil ice' affects infiltration in that area of Canada...yet the Canadian government explicitly excludes that area from modern maps (2022) of permafrost (which, I can only assume, is what's meant by soil ice). Another major issue that I found was in describing for Nature Com readers the granularity of the study. Data from sources that are 30 m (GLAD) as well as 90 m were used, then interposed and coarse-grained to 0.125 x 0.125 arc degrees – which my hamfisted analyses suggest is around 120 km². The point: wetlands, especially those that are pluvial in nature are generally pretty small. And models that are coarse, like this one, will miss a lot of the dynamics of those systems. Sure, explorations at CONUS scale require coarse-graining. But explorations of surface-water inundation dynamics in seasonally flooded wetlands require finer grained analyses – or at least readily digestible information for the reader to understand some of the grain-related limitations of the study. (I note that zooming into demonstrate the utility of their approach, at scales reasonable for readers to discern individual wetlands and/or complexes, would be useful and possible in the SI.) In closing, there's much the authors should be proud of in their analyses. So that others can replicate their approaches – including their conclusions – assumptions and limitations of the study to me should be much more clearly articulated and incorporated into the analyses so that the conclusions are contextualized in a more nuanced way (as befits a model analysis of this scale, scope, and

conclusion).

L39 Would be remiss to not include Nahlik and Fennessy here.

Nahlik, A. M. and M. S. Fennessy (2016). "Carbon storage in US wetlands." *Nature Communications* 7: 13835.

L46 Good place to reference Royal Gardner & Nature piece (relevant to US, anyway)

Gardner, R. C. (2023). "What the US Supreme Court decision means for wetlands." *Nature* 618: 215.

L48 periodically to permanently inundated...

L50 ...though we note that not all wetlands express surface water (e.g., saturated soils are sufficient to create wetland ecosystems).

L59 maybe drop the intrinsic, hidden for the simplicity of "deriving the relationships".

L60 should wetlands be plural? Should it be possessive if so?

L67 Consider adding Park et al. 2022. Also, the GLAD data (Figure S1) show that most of CONUS is covered between April and November 2010 (i.e., it has data, though those values may be low). If the point is to show that surface water fractionation is low during that period, then GLAD may not be the correct source. For instance, the Global Surface Water product (Pekel et al. 2016, already in references) shows the availability of water in a given Landsat pixel over the period of record (~30 years now). Or, the DSWE does much the same (Jones 2019; <https://www.usgs.gov/landsat-missions/landsat-collection-1-level-3-dynamic-surface-water-extent-science-product>). I strongly recommend zooming in to an area of interest w the data (be it GLAD or other) to allow readers to see what 30 m resolution data is providing. Figure S2 is otherwise not particularly useful.

And regarding the seasonality, Borja et al. (2020) used the Pekel et al. GSW to identify the global extent of ephemeral waters...which is exactly what this study is analyzing as well. Why not use these data? At the least, the presence of these data (Borja et al. 2020) should likely have come to the attention of the authors and needs to be referenced.

Borja, S., et al. (2020). "Global Wetting by Seasonal Surface Water Over the Last Decades." *Earth's Future* 8(3): e2019EF001449.

Jones, J.W. (2019). Improved Automated Detection of Subpixel-Scale Inundation—Revised Dynamic Surface Water Extent (DSWE) Partial Surface Water Tests. *Remote Sens.*, 11, 374
<https://doi.org/10.3390/rs11040374>.

Jones, J. W. (2015). Efficient wetland surface water detection and monitoring via Landsat: Comparison with in situ data from the Everglades Depth Estimation Network. *Remote Sensing*, 7(9), 12503-12538.
<http://dx.doi.org/10.3390/rs70912503>.

Park, J. et al. (2022). Seasonality of inundation in geographically isolated wetlands across the United States. *Environmental Research Letters* 17: 054005

L68 Maybe add a line to answer this question, "What are Earth system models?" other than tools. Are they process-based models? Statistical models? GIS-based prediction tools? Etc.

L83 Fan et al. (2013) developed a global database of depth to groundwater that allows for applications such as this to explore how groundwater supports the presence of wetlands.

Fan, Y., H. Li and G. Miguez-Macho (2013). "Global Patterns of Groundwater Table Depth." *Science* 339(6122): 940-943.

L86 is awkwardly phrased. Maybe missing a word ("Accurately represent...")

L86 I've not seen "diagnostic scheme" used to describe what is essentially a model or approach to mapping wetlands.

L88 What are high latitudes? Clarify for the reader.

L99 I'd argue that the calibration was done using a 30-m resolution dataset, not that the ESM data were calibrated against the high-resolution data. Two problems with that statement are that the ESM data are relatively coarse-grained, much coarser than the GLAD data and the calibration occurred after the GLAD data were modified to match the grain of the ESM. Secondly, unless I'm mistaken, the GLAD data are 30 m. That's medium resolution at best. In fact, it's probably best to simply state the resolution (in meters) and don't add the modifier regarding low, medium, or high resolution.

L101 Readers might want to know what other datasets are available. For instance, why not use the Pekel et al. (2016) dataset? Or some of the land cover products from ESA? Not saying that using GLAD is wrong by any stretch, but the readers will want to know (and should be told) that you're aware of other data sets and chose to use GLAD because...why? (GLAD is comprehensive – more comprehensive than other dataset how? And GLAD was 'carefully validated' – what does that mean, and though I appreciate careful validation, I'm also interested in highly accurate validation and a robust end product. Please include something along those lines that describe why GLAD was chosen over other data.)

L116 Though discussed further in the methods, the resolution of the model needs to be in here. GLAD is identified as a "high resolution" dataset with the implication then that the ESM is similarly of high resolution. But my understanding is that the ESM is at roughly 120 km². Readers need to know that (and I'm acknowledging that many readers won't go to the full methods section to discern that, and I point out further that the mixing of arc degrees and SI units further confuses the reader and should be avoided; use SI.)

L118 Figure S4 needs clarification. The scale has 0 for GLAD surface water fraction, but the caption states that "white area represents no data." It's currently both 0 and No Data, then. I recommend changing no data to gray. (And, as before, zooming in allows readers to visually discern the model, which can be reassuring.)

L120 The positive trend in wetland evolution should be simplified, as the authors mean, I think, "The creation or expansion of more non-permanently flooded waters". Two issues with this phrase include the use of evolution (what is evolving here?) and the lack of support for the conclusion. Where are the data that support the claim that more wetlands have been created between 2000 and 2020? For instance, the Fish and Wildlife Service publishes the Status and Trends series on wetlands, with the last one analyzing trends through 2009. This document (p. 16, here) states that "Overall, freshwater wetlands realized a slight increase in area between 2004 and 2009." There are other analyses, such as the varied works by the National Land Cover Database that could be cited as well. Zou et al. (2018) is another one to consider citing.

Zou, Z., X. Xiao, J. Dong, Y. Qin, R. B. Doughty, M. A. Menarguez, G. Zhang and J. Wang (2018). "Divergent trends of open-surface water body area in the contiguous United States from 1984 to 2016." *Proceedings of the National Academy of Sciences* 115(15): 3810.

L122 GIEMS requires a reference. Perhaps one of these...

Fluet-Chouinard, E., Lehner, B., Rebelo, L.-M., Papa, F., and Hamilton, S. K.: Development of a global

inundation map at high spatial resolution from topographic downscaling of coarse-scale remote sensing data, *Remote Sens. Environ.*, 158, 348–361, <https://doi.org/10.1016/j.rse.2014.10.015>, 2015.

Prigent, C., C. Jimenez and P. Bousquet (2020). "Satellite-Derived Global Surface Water Extent and Dynamics Over the Last 25 Years (GIEMS-2)." *Journal of Geophysical Research: Atmospheres* 125(3): e2019JD030711.

L124 Is the relationship with GIEMS associated w the wetland area, or the inundation? That is, the authors take pains to explain that their analyses exclude permanently inundated waters. But did they exclude permanently inundated pixels (waters) from the GIEMS data? There needs to be clarity throughout the manuscript on the specifics of the system analyzed and contrasted.

L134 Were the permanently flooded areas removed from (b), the simulated surface waters? And at this scale it is very difficult to discern how well the model performed at the meaningful local or large scale. I'd recommend adding (in the SI, as this is crowded already) a zoomed-in, high scale image that demonstrates the performance of the model such that individual pixels can be visually discerned.

L150 How were the basins selected (or delineated)?

L151 I only see the CONUS analyses here, nothing about the entirety (or even majority) of Canada or Mexico. Thus, the "continental scale" statement cannot be supported.

L154 I see more than just the MS valley. What about the California Central Valley? Colorado River? Okefenokee and Okeechobee swamp and lake, respectively? The Basin and Range area of Nevada? The Atlantic Coastal plain watersheds?

L154-156 Why are these wetland areas removed from pervious studies? Were the wetland areas removed or the surface water pixels removed?

L188 Historical is misspelled in the figure 2a.

L253 I'm assuming that 'soil ice' could also be called permafrost. The map from the Canadian government shows zero permafrost in the southern (southeastern) section of the country. Please clarify the statement and conclusion.
https://ftp.maps.canada.ca/pub/nrcan_rncan/raster/atlas_5_ed/eng/environment/land/mcr4177.jpg

L259 Why not spell out Temperature and Precipitation in the colorful line-bar instead of using Ta and Pr, which are not as clear and require reading the caption?

L264 Why was this particular example watershed selected?

L276 "...warming amplified water sinks in the wetland hydrological cycle..." could this be simplified to express how higher evap is increasingly negatively affecting wetland area? The current phrasing is overly complicated.

L280 see L253

L318 The conclusion that wetlands hydraulic conductivity will increase because of decreased soil ice (read: permafrost?) seems spurious. The authors have not provided literature that supports their notion that Canadian wetlands (or Prairie Pothole wetlands, if they want to analyze that more central region shared by the US and Canada) are wetlands because of "soil ice" that is assumed to be frozen and hence impermeable. Sure, the water in wetlands and ponds freezes during the winter..but the creation of a perched wetland due to soil ice that's maintained year round (or not, according to their

models) is not supported.

L324 I am unsure of what is a cold region and a hot region. Perhaps a map that defines this would be useful. Or, perhaps using the Köppen-Geiger map to interpret their results would be helpful. Source:

Beck, H. E., N. E. Zimmermann, T. R. McVicar, N. Vergopolan, A. Berg and E. F. Wood (2018). "Present and future Köppen-Geiger climate classification maps at 1-km resolution." *Scientific Data* 5(1): 180214.

L331 Rather than call this the Upper Mississippi River, it is probably more informative to call it the Prairie Pothole Region. Further, the results should be couched relative to other studies – do the authors' results hold up? Here are some PPR studies, at the least, to consider. And I would expect the authors to further couch their results in other analyses of the Everglades, the Mississippi Alluvial Valley, and the Great Salt Lake – all of which have been analyzed by others for climate change effects (search, and you will find...).

Ganming, L. and S. F. W. (2012). "Climate-driven variability in lake and wetland distribution across the Prairie Pothole Region: From modern observations to long-term reconstructions with space-for-time substitution." *Water Resources Research* 48(8).

McKenna, O. P., D. M. Mushet, D. O. Rosenberry and J. W. LaBaugh (2017). "Evidence for a climate-induced ecohydrological state shift in wetland ecosystems of the southern Prairie Pothole Region." *Climatic Change* 145(3): 273-287.

Millett, B., W. C. Johnson and G. Guntenspergen (2009). "Climate trends of the North American prairie pothole region 1906–2000." *Climatic Change* 93(1-2): 243-267.

Renton, D. A., D.M. Mushet, E.S. DeKeyser (2015). *Climate Change and Prairie Pothole Wetlands—Mitigating Water-Level and Hydroperiod Effects Through Upland Management*, Scientific Investigations Report 2015-5004. U.S. Department of the Interior, U.S. Geological Survey.

L373 One of the most confusing parts of this study is the use of non-permanent wetlands by design, but then the use of the term "wetlands" throughout (which, again, includes permanent waters to most...). For instance, the previous pages discussed how some areas would have an increase in wetlands, and some a decrease. Were the increases de novo wetlands, pixels that were previously 'dry' that had a >1 month (see 28-31 day comment, below) modeled "water present" during the growing season? What about the wetlands that were 'wet' for just a month but under the modeled climate changes ceased to be wet for that month? I assume they were losses. What about wetlands that were wet for, hypothetically, 10 months but then decreased to just one month (during the growing season)? Was that captured by the model and reported? And what about permanent waters (wetlands, to most) that ceased to be permanently (surface water) inundated during the growing season? Where they at all included – were they considered "new wetlands" since, but this study's definition, waters that are dry for a month or more are considered wetlands (though permanently flooded waters are not)?

L373 Climate change is most certainly not a natural driver but caused by human alterations of the climate (e.g., IPCC for a quick-n-dirty reference). Remove this parenthetical.

L378 How was wetland size discerned? Were the neighboring pixels region-grouped? And how were these statistical analyses done to explore how sizes increase or decrease over time? This should, IMO, be done via a multiple comparison test and reported...

L378 When "continental scale" is used, I see "CONUS" instead. I don't see any Mexican nor substantive Canadian analyses to support "continental scale" here or anywhere in the paper.

L382 Wetland hydrology seasonality? Or seasonality of non-permanent wetland hydrology?

L397 I think conclusions addressing the entirety of the “Northern Latitude high regions” based on the relatively temperate zones to the east of the Great Lakes are a bit overblown. Especially in light of the lack of supporting data associated with “soil ice” in this analysis.

L400 Studies, perhaps?

L400 Consider additional supporting docs here, FWIW...

<https://agupubs.onlinelibrary.wiley.com/doi/full/10.1029/2020EF001858>

<https://www.nature.com/articles/ngeo1160>

L416 sequent?

L420 constraining?

L442 Bottomland hardwoods of the Mississippi Alluvial Valley are flooded forested areas...and they are often able to be discerned through satellite mixed-pixel analyses. Furthermore, the NLCD identifies forested wetlands as one of their classes, and the NWI has palustrine forested and palustrine shrub scrub as woody wetlands. It is incorrect to state that most forested wetlands are tropical. (Not to mention mangrove systems, but that’s another story...)

L447 – I did not see any discussion material on how the decision to limit wetlands in this analysis to those systems with a) inundation (versus saturation) discerned by satellites thereby missing those with emergent vegetation, such as all the NLCD herbaceous wetlands or NWI emergent palustrine wetlands, or b) that these inundated wetlands were further limited to those that were not permanently flooded, thereby missing a great deal of waters (see, e.g., Figure 4 here)

Vanderhoof, M. K., et al. (2023). "High-frequency time series comparison of Sentinel-1 and Sentinel-2 satellites for mapping open and vegetated water across the United States (2017–2021)." *Remote Sensing of Environment* 288: 113498.

Materials and Methods

L454 Regarding permanently flooded waters: It is an important distinction that permanently flooded wetlands are NOT included in this study. One might argue this is sufficient to retitle this paper as it focuses on “seasonally inundated inland surface waters” rather than on what most would articulate as inland wetlands (which include the gradient from permanently flooded to ephemeral systems). For instance, Lane and D’Amico (2016, Figure 4), identify high abundances of “permanently flooded” National Wetlands Inventory water regimes from non-floodplain-based palustrine wetlands across CONUS, in particular in the southeast and Atlantic coastal areas. These abundant wetlands would be missed by this study...which ultimately is fine, as long as the caveat is noted more explicitly than it is currently in this paper.

L457 Regarding the use of the word “inundated” and its meaning regarding standing water as a required definition of a wetland. The origin of this expert understanding should be cited.

“The definition of ‘wetland habitat’ is based on expert understanding that although wetland environments (e.g., emergent wetlands) do not need to be covered by water permanently, they have to be inundated for at least one month during the growing season to develop suitable biotic habitats.”

It’s okay to have an operational definition of wetlands – but that should be contextualized and

acknowledged more explicitly. For instance, this operational definition of wetlands excludes all wetlands that are saturated but not inundated. Inundation is 'nice' because satellites can relatively easily capture the signature of water. But that is not the case with saturated soils. Yet saturated soils are critical to wetlands (see, e.g., Fan et al. 2013, in particular their Figure S17, wherein the water table depth of within 25 cm of the surface is plotted with wetland area, with an R2 of 0.89). Tootchi et al. (2018) utilized the Fan et al. approach and derived a global wetland map with (relatively) high accuracy to comparable datasets, further supporting the use of saturated soils as maintaining wetlands globally.

The conceptual definition of wetlands used here by the authors, wetlands are supported by fluvial and pluvial processes that support wetland (hydrologic) characteristics, is operationalized to this: "Surface water present for [at least 30 days] during the growing season, but not permanently inundated." This should be more clearly stated in the main text as well as in the methods section.

That's because most experts (in North America, or at least the US) would most likely refer to both the Cowardin classification system (Cowardin et al. 1979) or the 1987 Army Corps of Engineers manual (ref) to define wetlands – and both of these documents do not require standing water to define wetlands. Cowardin et al., for instance, classify wetlands thusly (with my highlights for emphasis, Status and Trends doc):

"Wetlands are lands transitional between terrestrial and aquatic systems where the water table is usually at or near the surface or the land is covered by shallow water. For purposes of this classification wetlands must have one or more of the following three attributes: (1) at least periodically, the land supports predominantly hydrophytes³¹, (2) the substrate is predominantly undrained hydric soil, and (3) the substrate is non-soil and is saturated with water or covered by shallow water at some time during the growing season of each year."

The Corps defines wetlands for jurisdictional purposes as areas that "...are inundated or saturated by surface or ground water at a frequency and duration to support, and that under normal circumstances do support, a prevalence of vegetation typically adapted for life in saturated soil conditions" (my emphasis). Further, the Corps uses approximately 12 inches to define 'near the soil surface'. So if an area has saturated soils near the surface and other wetland-defining characteristics, the Corps would consider it a wetland – despite not having water above the soil surface.

I'll be looking to see if the authors acknowledge the limitations of their approach. Though an example of emergent marsh is given, I would argue their approach likely misses many emergent marsh because of not only the lack of saturated soils as a defining characteristic, but also because emergent vegetation can block the spectral signature of water.

Side note: I'd argue for changing the "...inundated for one month..." to days, since months can be 28, 29, 30, or 31 days in length. And, for what it is worth, the 5 C value is typically reserved for below the soil surface (as that affects microbial activity). The short-hand for at the surface is frost-free days. Here's some relevant info from the 1987 Corps Manualv:

Growing season. The portion of the year when soil temperatures at 19.7 in. below the soil surface are higher than biologic zero (5o C) (U.S. Department of Agriculture & Soil Conservation Service 1985). For ease of determination this period can be approximated by the number of frost-free days (U.S. Department of the Interior 1970).

L457-458 I believe this should be changed to "suitable biotic characteristics" at the end, not "habitats".

L460 does the journal require a period after the Celsius abbreviation?

L481 Where is the data to show where ice in the soil would be expected to affect infiltration?

L484 Where is the data to show the expected depth to groundwater? The reference provided by Fan et al. (2016) would be a useful, citable map for the authors to contrast their assumptions.

L493 A reference supporting this supposition is needed.

L503 What is the input stream and river map used to be flooded? NHD? Something else? This is important to know how far up the stream network is mapped and modeled. (As well as replication.)

L508 90 m is generally not considered high resolution esp in the days of sub-meter resolution satellite data. 30 m is generally considered 'medium resolution'; 90 m is fairly coarse.

L519 This is roughly 120 km², correct? The readers of Nature Com should be provided that approximate cell size (at the equator, etc.) rather than having to do the math. I strongly recommend using SI throughout the manuscript for clarity (and definitely not jumping back and forth w arc degrees and meters). The 'size matters' wrt wetlands, so providing easy to read understanding of the granularity of this model is important.

L527 This implies that Eq 4 solves at finer resolutions than 90 m, when it seems to be in fact a 90-m product that's used. (This also goes to the point of mixing arc degrees and metric units, L519 comment; I recommend sticking with metric throughout.)

L535 I'm pretty sure that journal doesn't reference websites in this many; please refine.

L551 highly recommend keeping the data to meter-scale pixels instead of degrees. It's intuitive to the reader to go from 30 m to 120 km² by doing this or that, but not so much to go from 0.00025°×0.00025° to 0.125°×0.125°.

L554 Why not use other products that might be higher resolution (and independent?) to remove rivers, lakes, and reservoirs? Examples include the Global Lakes and Wetlands Database (1km, Lerner and Doll 2004), Globeland30 product (Chen et al., 2015), European Space Agency (ESA) WorldCover 2020 (Zananga et al., 2021), and the Dynamic World dataset (Brown et al., 2022); the Pekel data could work here, too at 30-m resolution...

Chen, J., J. Chen, A. Liao, X. Cao, L. Chen, X. Chen, C. He, G. Han, S. Peng, M. Lu, W. Zhang, X. Tong and J. Mills (2015). "Global land cover mapping at 30m resolution: A POK-based operational approach." ISPRS Journal of Photogrammetry and Remote Sensing 103: 7-27.

Lehner, B. and P. Doll (2004). "Development and validation of a global database of lakes, reservoirs and wetlands." Journal of Hydrology 296(1-4): 1-22.

Pekel, J.-F., A. Cottam, N. Gorelick and A. S. Belward (2016). "High-resolution mapping of global surface water and its long-term changes." Nature 540(7633): 418-422.

Zananga, D., Van De Kerchove, R., De Keersmaecker, W., Souverijns, N., Brockmann, C., Quast, R., Wevers, J., Grosu, A., Paccini, A., Vergnaud, S., Cartus, O., Santoro, M., Fritz, S., Georgieva, I., Lesiv, M., Carter, S., Herold, M., Li, Linlin, Tsendbazar, N.E., Ramoino, F., Arino, O. (2021). ESA WorldCover 10 m 2020 v100.

L549 SI is a good place to show that there were data gaps. And if there are many data gaps in monthly GLAD data, why wouldn't we expect there to be many data gaps in seasonal (i.e., 3-month periods)? And if the model was calibrated seasonally but validated annually, why should we trust any of the results presented in Figure 2 b-e (the seasonal analyses)?

L555 also removed permanently flooded ponds and wetlands, it should be noted

L559 What is a finer-resolution DEM? 10-m? 90-m? 250 m? Clarify.

L582 HUC08, noted.

Supplemental Comments

Eq 5: What supports the use of the default configurations for F_c and u in this equation? How would the model perform if a different suite of parameters were selected? Are there parameters appropriate for a CONUS-scale model? Can that be supported by literature?

Figure S1: How did the authors handle the apparent errors in GLAD that affected the distribution of water across the CONUS? For instance, it's apparent that March, May, October, and November panels have generally vertical areas of No Data (perhaps due to the scan-line correct failure; I'm unsure). How did those errors affect your analyses? (And why not cross-walk the GLAD data with NLCD data to see how well it captured the wetlands and open waters?)

Figure S3 What are the demarcations of the CONUS (here and throughout)? Were the permanent waters removed from the comparison?

Figure S4 and throughout: Were the Bahamas included in these analyses? If not, I recommend removing them from the mapped study area. Same with the islands off the southern coastal of CA (and perhaps the area around Cape Cod, too). Note that the legend includes white as zero values whereas the caption says white is no data. Recommend using gray for no data here and throughout.

Figure S5 Clarify in the caption that these were analyzed at what pixel size? Was this the 120 km² pixel contrast, after the GLAD data had been coarse-scaled to 120 km²? And were the surface waters here not including those permanently inundated systems?

Figure S6c Why is there such a great discrepancy between the GIEMS and modeled surface water in 2005? Add to caption or text. And, here and elsewhere, does the modeled surface water include those permanent bodies? Does GIEMS include 'permanently inundated' data?

Figure S7 I wonder if the scale should be red to blue? The white to blue makes it hard to discern if there are any data in the low inundation maps. To wit, should readers assume that the white is not inundated? (I'm not colorblind that I know of but very light blue and white are hard to discern...)

Figure F8 square scatters? There's probably a better term. Doublecheck – PDFs can have y-axis values greater than 1.0?

Figure S9 The striking difference between southeastern Canada and the northeastern United States is eye-catching. I recognize that there are different data available depending on the country. Is that striking difference because of underlying geological and climatological forcings, or a function of the data source? That's important to articulate in the manuscript...it certainly looks to be an artifact...

Figure S12 Why are there pretty substantive areas of Mexico in this plot, but not in any of the above HUC plots? Where were the HUCs derived for Mexico?

FS13 Add linear regression equation and R^2 to the plot. Note that the linear nature is fairly evident w/ SSP585 but the SSP126 relationship does not appear to be linear. Why should you force a linear relationship on it?

Reviewer #4 (Remarks to the Author):

Xu et al.

Climate change and inland wetlands

Xu and coauthors explore inland wetland hydrological dynamics using an Earth systems model and an ensemble approach to climate change through 2100. There's a lot in this paper, and useful information for subsequent analyses. At this juncture, I found myself asking more questions than I found answered by the paper. There are not intractable problems, it is more that the clarity I found necessary for Nature Coms was often missing, leading to some conclusions that might not be completely supported (without caveats). For instance, inland wetlands in this study are limited to inundated systems – but not permanently inundated systems. That's fine for an operational CONUS-scale study, but I'd argue that the repeated use of the word "wetland" or even "inland wetland" is not appropriate. The study is on non-permanently flooded inland surface waters, including wetlands. Further, with that condition on wetlands I wasn't able to discern later if their model runs addressed those currently permanently flooded waters that might become semi-permanently flooded (e.g., dry at least one month but flooded at least one month during the growing season).

Response: We appreciate the thoughtful comments from the reviewer. This comment shows that the definitions used in the previous manuscript version required additional work, which is what has been done in the revised version (Line 627 – Line 649). For completeness of the response, we provide unambiguous definitions below.

First, within the range of permanently inundated systems, we consider rivers, lakes, and reservoirs as water bodies representing different types of ecosystems from those present in permanently inundated wetlands. Thus, **we excluded rivers, lakes, and reservoirs** from the definition of wetlands. This is explicitly described in the main text (Line 167, Line 599, Line 629) and was also stated in the previous manuscript version.

Second, by design, our model simulates **both permanent and non-permanent** inundated waters. As described in the manuscript (Line 658 - 707), the model simulates fluvial and pluvial inundation process in the river floodplain and land components, respectively. For fluvial inundation, as correctly understood by the reviewer, the simulated inundated waters are not permanent, since the river flow does not always exceed the channel capacity (e.g., during drier seasons). However, for pluvial inundation, whether the inundation waters persist year-round or does not depend on the simulated water storage. The surface water storage in the model is determined by the balance between precipitation, evaporation, surface runoff, infiltration, and outflow (e.g., the inundated water in the wetland overflows when the water volume is larger than the wetland effective storage capacity). For some grid cells, due to very low evaporation and infiltration, the surface water storage can be above zero the whole year. As a result, **permanently inundated areas can persist in some grid cells due to pluvial inundation**. This can be confirmed by Figure R1a which shows the seasonal variation of the total simulated inundated area in an exemplary watershed (USGS 08020401 Lower Arkansas watershed). The

inundated area of the watershed is always larger than 0 in both historical and future periods. In other words, the model simulates both permanently inundated waters and a seasonal expansion of wetland areas.

Third, the model **does simulate possible transitions of permanently inundated areas into seasonally inundated regions** (as well as transitions in the opposite direction). Using the same example in Fig. R1, one can appreciate that the minimum inundation area simulated in the future period is mostly smaller than that in the historical period (Figure R1b). This is testament to the above statement that a fraction of the permanent inundation in this watershed is projected to transition to seasonal inundation in the future.

To expand on that point at the scale of the entire study domain, we analyze the change of annual minimum inundation between the future and historical periods for the study domain. Figure R2a and b show that many watersheds have annual minimum inundation larger than zero in both the historical and future periods. In addition, Figure R2c demonstrates that our **model can simulate the transitions of both permanent inundation to seasonal inundation** (i.e., minimum inundation decreases from historical to future periods), **and vice versa**.

To be consistent with how the model is applied (i.e., we excluded rivers, lakes, and reservoirs in our definition of wetland, see above), we removed **permanent surface water bodies** identified by Global Land Analysis & Discovery (GLAD; Pickens et al., 2020) dataset from GLAD seasonal surface water dynamics before this dataset was used for model calibration. Although the permanently flooded ponds and other permanent wetlands should be included in the benchmark dataset, they could be unintentionally excluded from the upscaled GLAD in above procedure. Additionally, it is still challenging to separate permanent wetlands from other permanent water bodies from GLAD. Therefore, the use of the upscaled GLAD dataset for calibration may result in an underestimation of the total inundation areas (i.e., seasonal + permanent inundation), and this bias can propagate to our model simulation through the calibration process.

Although a different dataset, Global Lakes and Wetlands Database (GLWD; Lehner and Döll, 2004), provides a map for lakes, rivers, and reservoirs separated from wetlands, the summed areas of lakes, rivers, and reservoirs (e.g., $\sim 510,000 \text{ km}^2$) are much larger than the permanent surface water captured in GLAD (e.g., $\sim 280,000 \text{ km}^2$). Therefore, removing lakes, rivers, and reservoirs based on the GLWD map from the original GLAD dataset can result in smaller total inundation areas, than removing the permanent surface water bodies (i.e., rivers, lakes, reservoirs, ponds, etc.) identified in the GLAD product. It is also a matter of consistency that we “prepare” the model calibration data using the same dataset and not a hybrid one (i.e., a mix of GLAD and GLWD), which will introduce additional significant uncertainties.

As a result of the apparent confusion with model capabilities and setup of the experimental design, in the revised manuscript, we stressed that our model can capture both seasonal and permanent inundated waters at Line 687 – Line 690 and highlighted the relevant uncertainties of removing permanent water bodies at Line 595– Line 611.

Figure R1. (a) Simulated total inundation areas of a watershed at monthly scales from historical period (1971-2000 on the bottom X-Axis) and future period under SSP585 scenario (2071-2100 on the top X-Axis). Subplot (b) shows the annual minimum total inundation areas of an exemplary watershed for both historical period (1971-2000 on the bottom X-Axis) and future periods (under the SSP585 scenario, 2071-2100 on the top X-Axis). The selected watershed is USGS 08020401 Lower Arkansas watershed located at Lon: -91.6587°, Lat: 34.2740°.

Figure R2. (a) Averaged annual minimum total inundation fraction during (a) the historical period (1971 – 2000) and (b) the future period (2071-2100) under the SSP585 scenario. Subplot (c) shows the change of the total inundation fraction between the future and historical periods.

The use of the word wetland and inland wetland has many implications yet I feel the authors need to provide additional clarity to the readers as to what's meant throughout the entirety of the paper (e.g., did they contrast the GIEMS data with their wetlands after removing the permanently inundated waters?).

Response: We removed rivers, lakes, and reservoirs (which we do not define as wetlands, per above) from the GLAD dataset to obtain a calibration set. As explained above, our simulated inundation does include permanently inundated waters. It is difficult to compare our results with the GIEMS dataset directly because the latter includes permanent surface water bodies (e.g., rivers, lakes, reservoirs, etc.) that we have removed from GLAD for calibration. Thus, we normalized our simulation and GIEMS dataset (e.g., remove the mean and divide by the standard deviation) to focus on the validation of monthly temporal variation and interannual variation.

Further, the study does not address "North America" but rather the conterminous US plus a smidge of Mexico and an area of southeastern Canada that – to me – appears to misbehave.

Response: We used exactly the same domain as used by the North American Land Data Assimilation System (NLDAS), which provides parameters and data inputs for modeling at the model spatial resolution (i.e., $\sim 12.5\text{km} \times 12.5\text{km}$). To further clarify the extent of our study area, we specified our domain covers the continental United States, southern Canada, and northern Mexico ($25^\circ - 53^\circ$ North) in the abstract and at Line 129 and Line 712 – Line 713 in the main text.

That is, the authors take pains to explain how 'soil ice' affects infiltration in that area of Canada...yet the Canadian government explicitly excludes that area from modern maps (2022) of permafrost (which, I can only assume, is what's meant by soil ice).

Response: We thank the reviewer for bringing the new information about permafrost. First of all, we want to clarify that the model also doesn't simulate permafrost. 'Soil ice' mentioned in the manuscript is not permafrost. Instead, 'soil ice' refers to seasonal freeze-up of near-surface soil moisture due to temperature falling below the freezing point. Please let us know if there is a better term should be used. In the manuscript, our conclusion is that the reduced content of soil ice due to global warming increases the infiltration rate.

Another major issue that I found was in describing for Nature Com readers the granularity of the study. Data from sources that are 30 m (GLAD) as well as 90 m were used, then interposed and coarse-grained to 0.125×0.125 arc degrees – which my hamfisted analyses suggest is around 120 km². The point: wetlands, especially those that are pluvial in nature are generally pretty small. And models that are coarse, like this one, will miss a lot of the dynamics of those systems. Sure, explorations at CONUS scale require coarse-graining. But explorations of surface-water inundation dynamics in seasonally flooded wetlands require finer grained analyses – or at least readily digestible information for the reader to understand some of the grain-related limitations of the study. (I note that zooming into demonstrate the utility of their approach, at scales reasonable for readers to discern individual wetlands and/or complexes, would be useful and possible in the SI.)

Response: We agree with the reviewer that high resolutions (e.g., less than 30m) and possibly a different type of physics formulation (e.g., 2D overland flow model coupled with non-isothermal models of energy exchange above and belowground) can potentially yield additional details on the space-time variation of pluvial inundation process. Whether this would add anything substantive to **the broad patterns of process interplay in the projected wetland changes in the 21st century elucidated by this study** is unclear. We do not infer a clear reason from on why such an approach would be critically important or, rather, why the “zooming-in” exercise on individual wetlands and/or complexes would bring anything of substantial value to the novelty of results our research already demonstrates. We contend that it is out of scope for this research.

Beyond the philosophical point of questionable added value to the broad patterns of change we already discover, we note that it is not currently computationally feasible to implement a high-resolution model at the continental (or even regional) scales, especially for the long-term climate-scale simulations carried out in this research. For example, it took us about 1.3 million CPU hours for the calibration and future ensemble simulations at our model resolution (e.g., $\sim 12.5\text{km} \times 12.5\text{km}$). We note that many impactful papers assessing climate change impacts on land-surface dynamics rarely used resolutions finer than 0.5° : clearly, there are persisting limits to what can actually be done given the state-of-science tools and computational facilities (e.g., Avis et al., 2011; Jung et al., 2010; Zhang et al., 2022; Zhang et al., 2023; Zhou et al., 2023).

To our best knowledge, current large-scale wetland projections relied on over-simplified algorithms (e.g., deriving wetland based on ground water table) and even coarser spatial resolutions (e.g., $0.25^\circ \sim 1^\circ$) (Avis et al., 2011; Ekici et al., 2019; Xi et al., 2021), as compared to our study. The clear distinction of our work from these studies is that it aims to improve the representation of wetland dynamics in an Earth system model by (1) running the model at $\sim 12.5\text{km} \times 12.5\text{km}$, a relatively high spatial resolution (in the context of Earth system modeling); (2) using a physically-based inundation scheme (i.e., the simulated inundation can interact with other land-surface processes); and (3) calibrating uncertain parameters against satellite observations. We highlight our advancement in the introduction section (Line 138 – Line 164).

Since we implement our model at $\sim 12.5\text{km} \times 12.5\text{km}$ resolution, we cannot show the simulated inundation dynamics below this resolution or at the resolution of satellite pixels. We only simulate the averaged inundation area for each $\sim 12.5\text{km} \times 12.5\text{km}$ grid cell. At a broader level in our attempt to capture the observed wetland dynamics, our model performance is **very good - to - excellent**, as demonstrated in Figure 1 (correlation = 0.84 and NSE = 0.7) and Figure S8 (validation against an independent dataset that was not involved in the calibration). **This gives us confidence that the discovery of process interplay in the projected wetland changes is correct and the broad patterns of change are valid**, even though the projection details at higher resolutions remain obscure.

In closing, there’s much the authors should be proud of in their analyses. So that others can replicate their approaches – including their conclusions – assumptions and limitations of the

study to me should be much more clearly articulated and incorporated into the analyses so that the conclusions are contextualized in a more nuanced way (as befits a model analysis of this scale, scope, and conclusion).

Response: We appreciate the reviewer's detailed comments, which are useful to improve our manuscript. Please find our point-by-point responses in the following.

L39 Would be remiss to not include Nahlik and Fennessy here.

Nahlik, A. M. and M. S. Fennessy (2016). "Carbon storage in US wetlands." Nature Communications 7: 13835.

Response: Added. Thanks for sharing the reference.

L46 Good place to reference Royal Gardner & Nature piece (relevant to US, anyway)

Gardner, R. C. (2023). "What the US Supreme Court decision means for wetlands." Nature 618: 215.

Response: Added.

L48 periodically to permanently inundated...

Response: Thank you for the suggestion which we have adopted.

L50 ...though we note that not all wetlands express surface water (e.g., saturated soils are sufficient to create wetland ecosystems).

Response: We added this clarification in the revised manuscript.

L59 maybe drop the intrinsic, hidden for the simplicity of "deriving the relationships".

Response: Modified.

L60 should wetlands be plural? Should it be possessive if so?

Response: We changed to "wetland dynamics".

L67 Consider adding Park et al. 2022. Also, the GLAD data (Figure S1) show that most of CONUS is covered between April and November 2010 (i.e., it has data, though those values may be low). If the point is to show that surface water fractionation is low during that period, then GLAD may not be the correct source. For instance, the Global Surface Water product (Pekel et al. 2016, already in references) shows the availability of water in a given Landsat pixel over the period of record (~30 years now). Or, the DSWE does much the same (Jones 2019; <https://www.usgs.gov/landsat-missions/landsat-collection-1-level-3-dynamic-surface-water-extent-science-product>). I strongly recommend zooming in to an area of interest w the data (be it GLAD or other) to allow readers to see what 30 m resolution data is providing.

Response: We used Fig.S1 to demonstrate that the GLAD dataset has poor spatial coverage at sub-annual scale, thus the monthly GLAD dataset cannot be used as benchmark for calibration. Specifically, the poor spatial coverage applies to any other satellite datasets due to long satellite revisit time as well as impacts of cloud, shade, and snow to the observation. For example, Pekel et al. 2016 suffers from the same issue and has worse spatial coverage in 2010 than GLAD as shown in Figure R3. The GLAD seasonal surface water dynamics (i.e., monthly

surface water dynamics averaged between 1999-2020) has much better spatial coverage, which was used in the calibration. Please see the details of calibration procedure in the Method section (Line 557 – L579 in previous version). Therefore, we don't think Park et al. 2022 is an appropriate reference at Line 67.

The focus of this study is to project wetland changes using an Earth system model (ESM) at spatial resolution of $0.125^\circ \times 0.125^\circ$ (i.e., $\sim 12.5\text{km} \times 12.5\text{km}$). Because ESM cannot resolve sub-grid variability of wetland and only simulates the inundated area as a fraction of the grid cell, we cannot show the simulation at 30m resolution to compare with the GLAD dataset at its original resolution. The authors of the GLAD dataset did a great job in visualizing their dataset at 30m resolution, thus the readers should refer to Pickens et al. (2020) for the visualization.

Figure R3. Monthly Global Surface Water (GSW) surface water fraction [-] of 2010. The white area represents grid cells that do not have a value.

Figure S2 is otherwise not particularly useful.

Response: We cannot agree with the reviewer. The default model cannot simulate the pluvial inundation process accurately and the modified infiltration scheme, which is developed in this work, significantly improves the pluvial inundation process. The modified infiltration scheme

has not been published earlier and we believe the schematic describing the modified infiltration scheme in Figure S2 will be helpful to other model developers to gain a detailed understanding of the new scheme. The necessity of using the modified scheme in Figure S2 can be found at Line 199 – Line 203 and Line 675 – Line 684 in the revised manuscript.

And regarding the seasonality, Borja et al. (2020) used the Pekel et al. GSW to identify the global extent of ephemeral waters...which is exactly what this study is analyzing as well. Why not use these data? At the least, the presence of these data (Borja et al. 2020) should likely have come to the attention of the authors and needs to be referenced.

Borja, S., et al. (2020). "Global Wetting by Seasonal Surface Water Over the Last Decades." *Earth's Future* 8(3): e2019EF001449.

Jones, J.W. (2019). Improved Automated Detection of Subpixel-Scale Inundation—Revised Dynamic Surface Water Extent (DSWE) Partial Surface Water Tests. *Remote Sens.*, 11, 374 <https://doi.org/10.3390/rs11040374>.

Jones, J. W. (2015). Efficient wetland surface water detection and monitoring via Landsat: Comparison with in situ data from the Everglades Depth Estimation Network. *Remote Sensing*, 7(9), 12503-12538. <http://dx.doi.org/10.3390/rs70912503>.

Park, J. et al. (2022). Seasonality of inundation in geographically isolated wetlands across the United States. *Environmental Research Letters* 17: 054005

Response: We agree Borja et al. (2020) and Pekel et al. (2016) can be potentially used for calibration. Jones, J.W. (2019) has been cited in a previous version. The other suggested references are also very good and relevant. Unfortunately, we cannot cite all the references because we already exceed the limit of references we can include for the journal. In addition, we didn't cite Borja et al. (2020) in the revised manuscript as we found Borja et al. (2020) was derived from Pekel et al. (2016). We cited Pekel et al. (2016) at Line 782 – Line 784 to make the readers aware of other satellite datasets.

L68 Maybe add a line to answer this question, "What are Earth system models?" other than tools. Are they process-based models? Statistical models? GIS-based prediction tools? Etc.

Response: We added the following explanation "ESMs are physically based models that couple atmosphere, land, ocean, land ice, sea ice, and river processes at large scales. Typically applied at spatial resolutions of 100 km or coarser, ESMs parameterize smaller-scale processes that are not explicitly resolved by the models." at Line 81 – Line 84 in the revised manuscript.

L83 Fan et al. (2013) developed a global database of depth to groundwater that allows for applications such as this to explore how groundwater supports the presence of wetlands.

Fan, Y., H. Li and G. Miguez-Macho (2013). "Global Patterns of Groundwater Table Depth." *Science* 339(6122): 940-943.

Response: We acknowledge Fan et al. (2013) is a good reference, but the discussion of relationship between wetland and groundwater table is not the focus of Fan et al. (2013). However, the other cited references focus on deriving wetland from groundwater table. Since we are already beyond the limit of how many references we can cite, we decide to not add this reference.

L86 is awkwardly phrased. Maybe missing a word (“Accurately represent...”)

Response: We rewrote this sentence as “Accurately representing the soil freeze-thaw process is crucial for understanding the interactions between wetland dynamics and groundwater under future climate, as over half of the global wetlands are located in the northern high latitudes (e.g., above 50° N)”.

L86 I’ve not seen “diagnostic scheme” used to describe what is essentially a model or approach to mapping wetlands.

Response: We refer “diagnostic scheme” as models that don’t explicitly simulate the wetland dynamics. A commonly used “diagnostic scheme” in Earth system models or Land surface models for wetland estimates wetland area as the fraction of the grid cell that is below the groundwater table. Groundwater table is dynamically simulated in the model, but wetland area is estimated based on an empirical function relating wetland area with the simulated groundwater table. This is a “diagnostic scheme” because wetland area is not estimated based on its value in a previous time step (e.g., through a differential equation) and the estimated wetland area has no impact on the hydrological cycle or other land processes. The reviewer can refer to Xi et al. (2021), which used the “**TOPMODEL-based diagnostic model**” to project the wetland changes in the future.

We added a brief discussion of a diagnostic scheme at Line 98 – Line 118.

In contrast, the simulated wetland areas in our physically based wetland scheme interact with other land and hydrological processes, such as runoff, evaporation, infiltration, and energy balance.

L88 What are high latitudes? Clarify for the reader.

Response: We specified it as north of 50° N.

L99 I’d argue that the calibration was done using a 30-m resolution dataset, not that the ESM data were calibrated against the high-resolution data. Two problems with that statement are that the ESM data are relatively coarse-grained, much coarser than the GLAD data and the calibration occurred after the GLAD data were modified to match the grain of the ESM. Secondly, unless I’m mistaken, the GLAD data are 30 m. That’s medium resolution at best. In fact, it’s probably best to simply state the resolution (in meters) and don’t add the modifier regarding low, medium, or high resolution.

Response: We accepted the suggestion. Please see the modification at Line 133 in the revised manuscript.

L101 Readers might want to know what other datasets are available. For instance, why not use the Pekel et al. (2016) dataset? Or some of the land cover products from ESA? Not saying that using GLAD is wrong by any stretch, but the readers will want to know (and should be told) that you're aware of other data sets and chose to use GLAD because...why? (GLAD is comprehensive – more comprehensive than other dataset how? And GLAD was 'carefully validated' – what does that mean, and though I appreciate careful validation, I'm also interested in highly accurate validation and a robust end product. Please include something along those lines that describe why GLAD was chosen over other data.)

Response: We acknowledge that satellite datasets are preferred for use in model calibration over time-invariant land cover products. This is because wetland dynamics has strong seasonal and interannual variability, which is ignored in time-invariant land cover products.

To our best knowledge, both the GLAD and Pekel et al. (2016) datasets are comprehensive surface water datasets, as they detect different types of surface water, and are available at global scales and different temporal scales.

We used GLAD for calibration in this study because we found GLAD was easier for us to download and process when we designed the study. We figured out how to download and process the dataset of Pekel et al. (2016) during the last revision. And we found Pekel et al. (2016) has a worse spatial coverage compared to GLAD in year 2010 (Figure R3). Overall, we don't have any suggestions on which dataset is better for calibration because we think both GLAD and Pekel et al. (2016) represent good benchmarks for calibrating/validating Earth system model. And it is beyond the scope of this study to evaluate existing satellite datasets. In the revised manuscript, we mentioned other satellite datasets can be used for calibration as well at Line 135 – Line 136.

L116 Though discussed further in the methods, the resolution of the model needs to be in here. GLAD is identified as a "high resolution" dataset with the implication then that the ESM is similarly of high resolution. But my understanding is that the ESM is at roughly 120 km². Readers need to know that (and I'm acknowledging that many readers won't go to the full methods section to discern that, and I point out further that the mixing of arc degrees and SI units further confuses the reader and should be avoided; use SI.)

Response: We added the model resolution at Line 176. And we clarified that the satellite dataset is upscaled to the model resolution.

L118 Figure S4 needs clarification. The scale has 0 for GLAD surface water fraction, but the caption states that "white area represents no data." It's currently both 0 and No Data, then. I recommend changing no data to gray. (And, as before, zooming in allows readers to visually discern the model, which can be reassuring.)

Response: We appreciate your comment. In the revised manuscript, we used gray to represent no data in the plots of surface water. We also added zoomed in regions as suggested by the

reviewer to show the detailed comparison. Please find the description in the main text at Line 183 – Line 197 and the comparison in Figure S6.

L120 The positive trend in wetland evolution should be simplified, as the authors mean, I think, “The creation or expansion of more non-permanently flooded waters”. Two issues with this phrase include the use of evolution (what is evolving here?) and the lack of support for the conclusion. Where are the data that support the claim that more wetlands have been created between 2000 and 2020? For instance, the Fish and Wildlife Service publishes the Status and Trends series on wetlands, with the last one analyzing trends through 2009. This document (p. 16, here) states that “Overall, freshwater wetlands realized a slight increase in area between 2004 and 2009.” There are other analyses, such as the varied works by the National Land Cover Database that could be cited as well. Zou et al. (2018) is another one to consider citing.

Zou, Z., X. Xiao, J. Dong, Y. Qin, R. B. Doughty, M. A. Menarguez, G. Zhang and J. Wang (2018). "Divergent trends of open-surface water body area in the contiguous United States from 1984 to 2016." *Proceedings of the National Academy of Sciences* 115(15): 3810.

Response: We have changed “wetland evolution” to “annual wetland changes”. And we added the reference (Zou et al. 2018) to support the positive trend.

L122 GIEMS requires a reference. Perhaps one of these...

Fluet-Chouinard, E., Lehner, B., Rebelo, L.-M., Papa, F., and Hamilton, S. K.: Development of a global inundation map at high spatial resolution from topographic downscaling of coarse-scale remote sensing data, *Remote Sens. Environ.*, 158, 348–361, <https://doi.org/10.1016/j.rse.2014.10.015>, 2015.

Prigent, C., C. Jimenez and P. Bousquet (2020). "Satellite-Derived Global Surface Water Extent and Dynamics Over the Last 25 Years (GIEMS-2)." *Journal of Geophysical Research: Atmospheres* 125(3): e2019JD030711.

Response: We added the corresponding references in the revised manuscript.

L124 Is the relationship with GIEMS associated with the wetland area, or the inundation? That is, the authors take pains to explain that their analyses exclude permanently inundated waters. But did they exclude permanently inundated pixels (waters) from the GIEMS data? There needs to be clarity throughout the manuscript on the specifics of the system analyzed and contrasted.

Response: We removed rivers, lakes and reservoirs which we do not define as wetlands from GLAD during calibration, and as explained above our simulated inundation does include permanently inundated waters. It is difficult to compare our results with the GIEMS dataset directly because the latter includes permanent water bodies (e.g., rivers, lakes, reservoirs) that we have removed from GLAD for calibration, and saturated soil that is not captured by our model. Additionally, there are other uncertainties in GIEMS, such as its inability to detect small water bodies and it includes saturated soil. Therefore, in the validation against GIEMS, we normalized our simulation and GIEMS to focus on the monthly temporal variations and interannual variations, which are less influenced by time-invariant permanent water bodies and

uncertainties. We add this clarification in the caption of Figure S8.

L134 Were the permanently flooded areas removed from (b), the simulated surface waters? And at this scale it is very difficult to discern how well the model performed at the meaningful local or large scale. I'd recommend adding (in the SI, as this is crowded already) a zoomed-in, high scale image that demonstrates the performance of the model such that individual pixels can be visually discerned.

Response: We didn't remove permanently flooded areas from the simulated surface waters. In the revised manuscript, we stressed that our model can simulate both permanent and seasonal inundation at Line 687 – Line 690.

We would like to note that Earth system models (ESMs) are not designed to resolve processes at local scales. Or more precisely for wetland, ESMs cannot simulate the change of a specific wetland as requested by the reviewer but rather they simulate the dynamics of wetland area within a grid cell. As the model was run at $\sim 12.5\text{km} \times 12.5\text{km}$, our simulation cannot provide details below this resolution. $12.5\text{km} \times 12.5\text{km}$ is relatively high resolution in the context of ESMs, and we acknowledge our model performance (Figure 1b) is excellent compared to existing ESMs. We added Figure S6 to show the comparison for the zoomed-in regions, but downscale technique is needed to downscale ESM's simulations to satellite original resolution (e.g., 30m). However, downscaling ESM's simulations is beyond the scope of this study.

L150 How were the basins selected (or delineated)?

Response: We clarified that basins were selected based on the dominance of fluvial inundation, defined as the fluvial process accounting for more than 70% of the annual surface water area. Please find the clarification at Line 230 – Line 231 in the revised manuscript.

L151 I only see the CONUS analyses here, nothing about the entirety (or even majority) of Canada or Mexico. Thus, the "continental scale" statement cannot be supported.

Response: Although our simulation didn't cover the whole North America continent, our analysis can be considered continental scale since our domain covers a significant fraction of the North America continent. In studies such as Maxwell et al. (2015); O'Neill et al. (2021) using CONUS as their study domains, their simulations were regarded as continental scale simulation.

L154 I see more than just the MS valley. What about the California Central Valley? Colorado River? Okefenokee and Okeechobee swamp and lake, respectively? The Basin and Range area of Nevada? The Atlantic Coastal plain watersheds?

Response: In the revised manuscript, we changed the sentence to "is mostly significant along the major rivers (e.g., Mississippi River, Colorado River, etc.)" at Line 235.

L154-156 Why are these wetland areas removed from pervious studies? Were the wetland areas removed or the surface water pixels removed?

Response: In the satellite-based surface water dynamics dataset, the pluvial and fluvial inundation processes are not differentiated. Therefore, previous global fluvial inundation studies (e.g., Decharme et al., 2012; Mao et al., 2019) derived the satellite-based floodplain inundation dynamics by removing wetlands based on a wetland map from the satellite surface

water dynamics, assuming wetland is time-invariant. However, we found the pluvial process contributes more to the surface water dynamics than the fluvial inundation process in both magnitude and variations. We aimed to stress the importance of pluvial inundation process in simulating surface water dynamics, which is ignored in current Earth system models and Land surface models.

L188 Historical is misspelled in the figure 2a.

Response: Thank you for catching this typo. It is fixed in the revised manuscript.

L253 I'm assuming that 'soil ice' could also be called permafrost. The map from the Canadian government shows zero permafrost in the southern (southeastern) section of the country. Please clarify the statement and

conclusion. https://ftp.maps.canada.ca/pub/nrcan_rncan/raster/atlas_5_ed/eng/environment/land/mcr4177.jpg

Response: We don't agree that 'soil ice' could also be called permafrost. By definition of <https://www.sciencedirect.com/topics/agricultural-and-biological-sciences/permafrost>, permafrost refers to ground that remains at or below 0°C for two or more years. However, our model doesn't simulate permafrost in the southeastern Canada so the 'soil ice' mentioned here is definitely not permafrost. Instead, 'soil ice' refers to seasonal frozen soil moisture in the subsurface due to temperature falling below the freezing point during winter or cold months. Our model explicitly simulates the phase change in the soil water, please refer to Section 6.2.1 of Oleson et al. (2013) for a more detailed description. Therefore, we think it is more appropriate to refer 'soil ice' to frozen soil (<https://www.sciencedirect.com/topics/agricultural-and-biological-sciences/frozen-soils>). Please see our added explanation at Line 354 – Line 356.

L259 Why not spell out Temperature and Precipitation in the colorful line-bar instead of using Ta and Pr, which are not as clear and require reading the caption?

Response: Thank you for the suggestion. Temperature and Precipitation are used in the revised manuscript.

L264 Why was this particular example watershed selected?

Response: This region is arbitrarily picked from the regions that have a changed driver from precipitation in SSP126 to temperature in SSP585.

L276 "...warming amplified water sinks in the wetland hydrological cycle..." could this be simplified to express how higher evap is increasingly negatively affecting wetland area? The current phrasing is overly complicated.

Response: We modified this statement as suggested at Line 391 – Line 392.

L280 see L253

Response: Please see our response to L253.

L318 The conclusion that wetlands hydraulic conductivity will increase because of decreased soil ice (read: permafrost?) seems spurious. The authors have not provided literature that

supports their notion that Canadian wetlands (or Prairie Pothole wetlands, if they want to analyze that more central region shared by the US and Canada) are wetlands because of “soil ice” that is assumed to be frozen and hence impermeable. Sure, the water in wetlands and ponds freezes during the winter...but the creation of a perched wetland due to soil ice that’s maintained year round (or not, according to their models) is not supported.

Response: Please refer to our previous response regarding the meaning of soil ice, which is not related to permafrost. The impact of the soil ice (frozen soil moisture) is to reduce the permeability of the surface and it is one of the wetland generation mechanisms during cold months for the southeastern Canada region. Since southeastern Canada is not a permafrost region, there will not be ice in the soil during the warm months (i.e., summer and fall). Similar to the other regions, wetland or pluvial inundation can still form in such Canadian region due to the soil saturation and the surface water cannot infiltrate into the subsurface. Therefore, it is possible for the wetland to be maintained year-round in southeastern Canada, but due to the soil saturation instead of soil ice.

In the revised manuscript, we added this explanation at Line 441 – Line 443.

L324 I am unsure of what is a cold region and a hot region. Perhaps a map that defines this would be useful. Or, perhaps using the Köppen-Geiger map to interpret their results would be helpful. Source:

Beck, H. E., N. E. Zimmermann, T. R. McVicar, N. Vergopolan, A. Berg and E. F. Wood (2018). "Present and future Köppen-Geiger climate classification maps at 1-km resolution." *Scientific Data* 5(1): 180214.

Response: We added Fig.S16 in supplementary materials for the annual temperature. We specified in the main text that cold regions refer to regions with an annual mean temperature around 0°C at Line 447 – Line 448.

L331 Rather than call this the Upper Mississippi River, it is probably more informative to call it the Prairie Pothole Region. Further, the results should be couched relative to other studies – do the authors’ results hold up? Here are some PPR studies, at the least, to consider. And I would expect the authors to further couch their results in other analyses of the Everglades, the Mississippi Alluvial Valley, and the Great Salt Lake – all of which have been analyzed by others for climate change effects (search, and you will find...).

Ganming, L. and S. F. W. (2012). "Climate-driven variability in lake and wetland distribution across the Prairie Pothole Region: From modern observations to long-term reconstructions with space-for-time substitution." *Water Resources Research* 48(8).

McKenna, O. P., D. M. Mushet, D. O. Rosenberry and J. W. LaBaugh (2017). "Evidence for a climate-induced ecohydrological state shift in wetland ecosystems of the southern Prairie Pothole Region." *Climatic Change* 145(3): 273-287.

Milllett, B., W. C. Johnson and G. Guntenspergen (2009). "Climate trends of the North American

prairie pothole region 1906–2000." Climatic Change 93(1-2): 243-267.

Renton, D. A., D.M. Mushet, E.S. DeKeyser (2015). Climate Change and Prairie Pothole Wetlands—Mitigating Water-Level and Hydroperiod Effects Through Upland Management, Scientific Investigations Report 2015-5004. U.S. Department of the Interior, U.S. Geological Survey.

Response: Thank you for bringing the relevant references to our attention. We prefer to use Upper Mississippi as we didn't use the exact boundary for Prairie Pothole Region and other wetland regions. This is because our model doesn't simulate each individual wetland habitat (e.g., Prairie Pothole wetland), but simulate the dynamics of wetland area in each computational grid cell. We aggregated the gridded simulation to different subregions, which were circled in Fig.7.

In the revised manuscript, we added more references on the impacts of climate change on the selected wetland habitat regions and compared to our conclusion. Please find the modifications at Line 459 – Line 470.

L373 One of the most confusing parts of this study is the use of non-permanent wetlands by design, but then the use of the term "wetlands" throughout (which, again, includes permanent waters to most...). For instance, the previous pages discussed how some areas would have an increase in wetlands, and some a decrease. Were the increases de novo wetlands, pixels that were previously 'dry' that had a >1 month (see 28-31 day comment, below) modeled "water present" during the growing season? What about the wetlands that were 'wet' for just a month but under the modeled climate changes ceased to be wet for that month? I assume they were losses. What about wetlands that were wet for, hypothetically, 10 months but then decreased to just one month (during the growing season)? Was that captured by the model and reported? And what about permanent waters (wetlands, to most) that ceased to be permanently (surface water) inundated during the growing season? Where they at all included – were they considered "new wetlands" since, but this study's definition, waters that are dry for a month or more are considered wetlands (though permanently flooded waters are not)?

Response: Please refer to our previous response regarding the discussion of permanent wetlands. In brief, our model is able to capture permanent inundated waters and the transition from permanent inundated water to seasonal inundated water under climate change conditions, and vice versa (Figure R1 and Figure R2). Thus, the situations mentioned by the reviewer are considered in our analysis.

What about the wetlands that were 'wet' for just a month but under the modeled climate changes ceased to be wet for that month? I assume they were losses.

Response: Yes. This wetland habitat will be lost under the modelled climate change scenario based on our definition of wetland habitat. However, if the "wet" shifts to another month during the growing season, that site is still considered as wetland under the modeled climate changes.

What about wetlands that were wet for, hypothetically, 10 months but then decreased to just one month (during the growing season)?

Response: In this case, the wetland habitat extent remains unchanged based on our definition. Once a site is inundated by more than a month (the threshold used in this study), it will develop wetland characteristics so it doesn't matter how long it is inundated.

We admit that the selection of one month as the threshold may introduce uncertainty in the wetland habitat definition, though the criteria used to define wetland habitat vary among existing studies. This limitation was discussed in the revised manuscript at Line 647 – Line 649.

And what about permanent waters (wetlands, to most) that ceased to be permanently (surface water) inundated during the growing season?

Response: If this site is still inundated for one month (e.g., change from permanently inundated), it still provides wetland functions. Our definition of wetland habitat only requires the site to be inundated for one month during the growing seasons, and it doesn't matter if the site is seasonally or permanently inundated.

L373 Climate change is most certainly not a natural driver but caused by human alterations of the climate (e.g., IPCC for a quick-n-dirty reference). Remove this parenthetical.

Response: We changed the term to climate driver.

L378 How was wetland size discerned? Were the neighboring pixels region-grouped? And how were these statistical analyses done to explore how sizes increase or decrease over time? This should, IMO, be done via a multiple comparison test and reported...

Response: The wetland size here means the sum of total inundated areas from the whole domain. And the change is the difference of the total inundated areas between future period and historical period. We changed "wetland size" to "wetland area" to avoid confusion. We also rewrote that sentence for clarity (see Line 514 – Line 518).

L378 When "continental scale" is used, I see "CONUS" instead. I don't see any Mexican nor substantive Canadian analyses to support "continental scale" here or anywhere in the paper.

Response: Although our simulation didn't cover the whole North America continent, our analysis can be considered continental scale since our domain covers a significant fraction of the North America continent. Please refer to our reply to a similar comment earlier.

L382 Wetland hydrology seasonality? Or seasonality of non-permanent wetland hydrology?

Response: We changed the term to "Seasonality of wetland dynamics". For example, in some regions, the maximum wetland area is projected to occur earlier in the future than in the historical.

L397 I think conclusions addressing the entirety of the "Northern Latitude high regions" based on the relatively temperate zones to the east of the Great Lakes are a bit overblown. Especially in light of the lack of supporting data associated with "soil ice" in this analysis.

Response: Here we discuss an implication based on our analysis and literature review. Our simulation over the southeastern Canada is representative of cold regions with no permafrost. While previous studies suggested global warming may thaw the permafrost in the high latitude and result in wetland loss (Avis et al., 2011), our study shows that even in cold regions with no permafrost, warming can greatly reduce the wetland area. We added references Avis et al. 2011 and Kåresdotter et al. (2021) to support our statement (please see Line 547).

L400 Studies, perhaps?

L400 Consider additional supporting docs here, FWIW...

<https://agupubs.onlinelibrary.wiley.com/doi/full/10.1029/2020EF001858>

<https://www.nature.com/articles/ngeo1160>

Response: Thank you for the references. We think those two references are more appropriate to support our previous statement. In the revised manuscript, we cited them at Line 547.

L416 sequent?

Response: This is changed to “later”.

L420 constraining?

Response: Accepted.

L442 Bottomland hardwoods of the Mississippi Alluvial Valley are flooded forested areas...and they are often able to be discerned through satellite mixed-pixel analyses. Furthermore, the NLCD identifies forested wetlands as one of their classes, and the NWI has palustrine forested and palustrine shrub scrub as woody wetlands. It is incorrect to state that most forested wetlands are tropical. (Not to mention mangrove systems, but that’s another story...)

Response: Thank you for pointing this out. We removed the argument that “such wetlands are mainly located in tropical rainforest” in the revised manuscript.

L447 – I did not see any discussion material on how the decision to limit wetlands in this analysis to those systems with a) inundation (versus saturation) discerned by satellites thereby missing those with emergent vegetation, such as all the NLCD herbaceous wetlands or NWI emergent palustrine wetlands, or b) that these inundated wetlands were further limited to those that were not permanently flooded, thereby missing a great deal of waters (see, e.g., Figure 4 here)

Vanderhoof, M. K., et al. (2023). "High-frequency time series comparison of Sentinel-1 and Sentinel-2 satellites for mapping open and vegetated water across the United States (2017–2021)." *Remote Sensing of Environment* 288: 113498.

Response: As noted in our response to your comments regarding permanent wetlands, our model can capture the permanent inundated wetlands. However, the calibrated model may underestimate the total wetland areas because permanent wetlands were unintentionally

removed from GLAD. We added this as a limitation at Line 595 – Line 611 in the revised manuscript.

Materials and Methods

L454 Regarding permanently flooded waters: It is an important distinction that permanently flooded wetlands are NOT included in this study. One might argue this is sufficient to retitile this paper as it focuses on “seasonally inundated inland surface waters” rather than on what most would articulate as inland wetlands (which include the gradient from permanently flooded to ephemeral systems). For instance, Lane and D’Amico (2016, Figure 4), identify high abundances of “permanently flooded” National Wetlands Inventory water regimes from non-floodplain-based palustrine wetlands across CONUS, in particular in the southeast and Atlantic coastal areas. These abundant wetlands would be missed by this study...which ultimately is fine, as long as the caveat is noted more explicitly than it is currently in this paper.

Response: We would like to emphasize that the definition of wetland is not consistent among different studies, which results in significant uncertainty in the estimation of wetland extent (Tootchi et al., 2019). In the revised manuscript, we further clarified the definition of the wetland area to be flooded waters, excluding rivers, lakes, and reservoirs. Our definition of wetland includes both seasonally and permanently flooded waters, which are captured by our model. We removed permanent water bodies from the GLAD dataset to exclude river, lakes, and reservoirs. In this way, the permanent inundated wetland could be unintentionally removed from GLAD as well because satellite dataset cannot differentiate permanent inundated wetland from other permanent water bodies. Therefore, the calibrated model may result in underestimation of total wetlands (i.e., sum of seasonally and permanently wetlands). We explicitly include this limitation in the main text at Line 595 – Line 611.

In addition, we exclude coastal regions from our study domain because we focus on the inland wetland. Our model cannot capture the coastal wetlands, which are generated due to tidal dynamics.

L457 Regarding the use of the word “inundated” and its meaning regarding standing water as a required definition of a wetland. The origin of this expert understanding should be cited.

“The definition of ‘wetland habitat’ is based on expert understanding that although wetland environments (e.g., emergent wetlands) do not need to be covered by water permanently, they have to be inundated for at least one month during the growing season to develop suitable biotic habitats.”

Response: The definition of ‘wetland habitat’ is based on our consultation with a wetland expert from his field work experience. We acknowledged him in the acknowledgments. And we don’t have a reference to support this definition. In the revised manuscript, we added this definition as another source of uncertainty in our analysis at Line 645 – Line 649.

It’s okay to have an operational definition of wetlands – but that should be contextualized and acknowledged more explicitly. For instance, this operational definition of wetlands excludes all

wetlands that are saturated but not inundated. Inundation is ‘nice’ because satellites can relatively easily capture the signature of water. But that is not the case with saturated soils. Yet saturated soils are critical to wetlands (see, e.g., Fan et al. 2013, in particular their Figure S17, wherein the water table depth of within 25 cm of the surface is plotted with wetland area, with an R2 of 0.89). Tootchi et al. (2018) utilized the Fan et al. approach and derived a global wetland map with (relatively) high accuracy to comparable datasets, further supporting the use of saturated soils as maintaining wetlands globally.

Response: We mentioned in the previous version at Line 441 that our model cannot capture wetlands that are solely formed due to soil saturation. This explanation can also be found at Line 614 – Line 616 in the revised manuscript. It is mainly because ESMs are still too coarse and also do not resolve the sub-grid spatial heterogeneity of groundwater table within a grid cell. Comparing this grid-cell averaged water table depth with an arbitrary groundwater table criterion (such as within 25 cm of the surface used by Fan et al. 2013 and others) is not appropriate. We acknowledge the caveat in our wetland definition that can exclude wetland solely formed due to soil saturation, but we also argue that this practical definition includes most of the inland wetland categories in the study domain as defined by Coward et al. (1979) and demonstrated by our validation. Besides, it is not surprising that Fan et al. 2013 found a good relationship between shallow groundwater table and wetland extent because shallower groundwater tables usually occur in relatively low-lying regions, which are easily inundated due to precipitation or surface runoff. But we note that Fan et al. 2013 did not include inundation processes in their model, thus their simulations couldn’t well define inundated wetland that consist of most of the wetland categories according to Coward et al. (1979). In the revised manuscript, we explicitly mentioned that our defined wetland habitat could miss the wetlands formed solely due to soil saturation (see Line 645 – Line 646).

The conceptual definition of wetlands used here by the authors, wetlands are supported by fluvial and pluvial processes that support wetland (hydrologic) characteristics, is operationalized to this: Surface water present for [at least 30 days] during the growing season, but not permanently inundated.” This should be more clearly stated in the main text as well as in the methods section.

Response: Again, we note our simulated inundation includes both seasonal and permanent inundation. We explicitly described the definition of wetlands at Line 165 – Line 166.

That’s because most experts (in North America, or at least the US) would most likely refer to both the Cowardin classification system (Coward et al. 1979) or the 1987 Army Corps of Engineers manual (ref) to define wetlands – and both of these documents do not require standing water to define wetlands. Coward et al., for instance, classify wetlands thusly (with my highlights for emphasis, Status and Trends doc):

“Wetlands are lands transitional between terrestrial and aquatic systems where the water table is usually at or near the surface or the land is covered by shallow water. For purposes of this classification wetlands must have one or more of the following three attributes: (1) at least periodically, the land supports predominantly hydrophytes³¹, (2) the substrate is predominantly undrained hydric soil, and (3) the substrate is non-soil and is saturated with

water or covered by shallow water at some time during the growing season of each year.”

The Corps defines wetlands for jurisdictional purposes as areas that "...are inundated or saturated by surface or ground water at a frequency and duration to support, and that under normal circumstances do support, a prevalence of vegetation typically adapted for life in saturated soil conditions" (my emphasis). Further, the Corps uses approximately 12 inches to define 'near the soil surface'. So if an area has saturated soils near the surface and other wetland-defining characteristics, the Corps would consider it a wetland – despite not having water above the soil surface.

Response: We agree with the definition that wetland does not require standing water. This is also the reason that we introduce the wetland habitats in this study. Specifically, our definition of wetland habitats requires a site to be inundated for at least one month, though there can be no standing water in the other months. So, our definition of wetland habitats is consistent with the classification of Cowardin et al. (1979) and Army Corps. An exception is when a site becomes wetland when the soil is only saturated during the growing season, but no inundation occurs during the year. Our model cannot capture such wetland types as shown in Figure R4. In addition, our model can simulate saturated soil, but it is not clear when the saturated soil will become wetlands, and such wetlands cannot be observed from satellite. However, we think the soil saturation wetlands should be closely connected to inundated wetlands (Figure R4). Although the soil saturation wetlands cannot be explicitly simulated in our model, its sensitivity to climate change may be consistent with that of inundated wetlands. This is because surface water has been found to be closely related to groundwater dynamics (Brunner et al., 2009; Zou et al., 2018).

Figure R4. Conceptual diagram for wetland in reality and wetland in simulation.

I'll be looking to see if the authors acknowledge the limitations of their approach. Though an example of emergent marsh is given, I would argue their approach likely misses many emergent marsh because of not only the lack of saturated soils as a defining characteristic, but also because emergent vegetation can block the spectral signature of water.

Response: We added an additional explanation for the model limitation in capturing soil saturation wetlands at Line 612– Line 615.

Side note: I'd argue for changing the "...inundated for one month..." to days, since months can

be 28, 29, 30, or 31 days in length. And, for what it is worth, the 5 C value is typically reserved for below the soil surface (as that affects microbial activity). The short-hand for at the surface is frost-free days. Here's some relevant info from the 1987 Corps Manual:

Growing season. The portion of the year when soil temperatures at 19.7 in. below the soil surface are higher than biologic zero (5o C) (U.S. Department of Agriculture & Soil Conservation Service 1985). For ease of determination this period can be approximated by the number of frost-free days (U.S Department of the Interior 1970).

Response: It should be more accurate to use 30 days to define wetland habitats. But we saved the simulation outputs at monthly scale as archiving daily data requires too much space. So, we cannot analyze the data using 30 days as the threshold for wetland habitats. In addition, we only saved selected simulated variables to further reduce data storage, so unfortunately, we didn't save soil temperature data. We don't have the computational allocations to rerun all the simulations, which requires tens of thousands of node hours on our high-performance computing facility. We explicitly describe these limitations in the revised manuscript at Line 647 – Line 649.

L457-458 I believe this should be changed to “suitable biotic characteristics” at the end, not “habitats”.

Response: Changed.

L460 does the journal require a period after the Celsius abbreviation?

Response: It was a typo. We removed the period in the revised manuscript.

L481 Where is the data to show where ice in the soil would be expected to affect infiltration?

Response: In the model, Θ_{ice} is estimated during the simulation based on the fraction of ice in the soil. Please refer to Eq (7.171) in Oleson et al. (2013). The presence of ice in the soil lowers the infiltration capacity for water because the ice “decreases the total pore space and increase the tortuosity of the remaining pore space” as explained by Appels et al. (2018).

L484 Where is the data to show the expected depth to groundwater? The reference provided by Fan et al. (2016) would be a useful, citable map for the authors to contrast their assumptions.

Response: Z_{wt} is simulated in our model. Simulated groundwater table is commonly highly biased in Earth system model, but it is critical for modeling wetland dynamics. Therefore, we calibrated f_{over} that links the groundwater table to pluvial inundation process to reduce the impacts of the bias in simulated groundwater table (Eq (2)).

L493 A reference supporting this supposition is needed.

Response: We cited Jackson et al. (2014) to support our assumption.

L503 What is the input stream and river map used to be flooded? NHD? Something else? This is important to know how far up the stream network is mapped and modeled. (As well as replication.)

Response: The river network and channel geometry were provided by the Dominant River Tracing algorithm (DRT; Wu et al., 2012). We included this reference at Line 523 – Line 527 in our previous submission. And the reviewer can also find it at Line 718 – Line 722 in the revised manuscript.

L508 90 m is generally not considered high resolution esp in the days of sub-meter resolution satellite data. 30 m is generally considered ‘medium resolution’; 90 m is fairly coarse.

Response: In the context of Earth system modeling with typical spatial resolution of ~100km, 90m is considered very high resolution.

To avoid confusing readers from other fields, we changed this sentence to: “This volume-area relationship is described by surface elevation distribution (e.g., at spatial resolution of 90m) within the computational unit, assuming that riverine inundation propagates from lower elevations to higher elevations”.

L519 This is roughly 120 km², correct? The readers of Nature Com should be provided that approximate cell size (at the equator, etc.) rather than having to do the math. I strongly recommend using SI throughout the manuscript for clarity (and definitely not jumping back and forth w arc degrees and meters). The ‘size matters’ wrt wetlands, so providing easy to read understanding of the granularity of this model is important.

Response: We provide the resolution in SI unit, which is ~12.5km × 12.5km.

L527 This implies that Eq 4 solves at finer resolutions than 90 m, when it seems to be in fact a 90-m product that’s used. (This also goes to the point of mixing arc degrees and metric units, L519 comment; I recommend sticking with metric throughout.)

Response: We changed this sentence to “Further, the relationship of Eq. (4) was derived from the 90 m-resolution Digital Elevation Model (DEM) from the Hydrological Data and Maps Based on Shuttle Elevation Derivatives at Multiple Scales (HydroSHEDS)”.

L535 I’m pretty sure that journal doesn’t reference websites in this many; please refine.

Response: We checked the website, and unfortunately, they don’t provide a doi for the document.

L551 highly recommend keeping the data to meter-scale pixels instead of degrees. It’s intuitive to the reader to go from 30 m to 120 km² by doing this or that, but not so much to go from 0.00025°×0.00025° to 0.125°×0.125°.

Response: Changed to 30m × 30m and ~12.5km × 12.5km, respectively.

L554 Why not use other products that might be higher resolution (and independent?) to remove rivers, lakes, and reservoirs? Examples include the Global Lakes and Wetlands Database (1km, Lerner and Doll 2004), GlobeLand30 product (Chen et al., 2015), European Space Agency (ESA) WorldCover 2020 (Zananga et al., 2021), and the Dynamic World dataset (Brown et al., 2022); the Pekel data could work here, too at 30-m resolution...

Chen, J., J. Chen, A. Liao, X. Cao, L. Chen, X. Chen, C. He, G. Han, S. Peng, M. Lu, W. Zhang, X. Tong and J. Mills (2015). "Global land cover mapping at 30m resolution: A POK-based

operational approach." ISPRS Journal of Photogrammetry and Remote Sensing 103: 7-27.
Lehner, B. and P. Doll (2004). "Development and validation of a global database of lakes, reservoirs and wetlands." Journal of Hydrology 296(1-4): 1-22.
Pekel, J.-F., A. Cottam, N. Gorelick and A. S. Belward (2016). "High-resolution mapping of global surface water and its long-term changes." Nature 540(7633): 418-422.
Zanaga, D., Van De Kerchove, R., De Keersmaecker, W., Souverijns, N., Brockmann, C., Quast, R., Wevers, J., Grosu, A., Paccini, A., Vergnaud, S., Cartus, O., Santoro, M., Fritz, S., Georgieva, I., Lesiv, M., Carter, S., Herold, M., Li, Linlin, Tsendbazar, N.E., Ramoino, F., Arino, O. (2021). ESA WorldCover 10 m 2020 v100.

Response: Each dataset has its own algorithms and definitions for identifying rivers, lakes, and reservoirs. It is possible that the seasonal flooded water identified in GLAD is identified instead as lakes or rivers in another dataset. The resolution mismatch may also introduce uncertainties. For example, we analyzed the total area of rivers, lakes, and reservoirs in our study domain using Global Lakes and Wetlands Database, which is about 500,000 [km^2]. However, the total permanent surface water detected by GLAD is only about 280,000 [m^2].

L549 SI is a good place to show that there were data gaps. And if there are many data gaps in monthly GLAD data, why wouldn't we expect there to be many data gaps in seasonal (i.e., 3-month periods)? And if the model was calibrated seasonally but validated annually, why should we trust any of the results presented in Figure 2 b-e (the seasonal analyses)?

Response: The GLAD seasonal surface water dynamics represents monthly averaged surface water between 1999-2020. As we are not the authors of this dataset, the reviewer and reader should refer to their reference for how the seasonal surface water was generated.

The evaluation at seasonal scale against GLAD can be found in Fig.S3. We also validated the monthly variation against GIEMS, which is an independent dataset compared to GLAD and contains an independent period different from the calibration period. Please refer to Line 177 – Line 179 in the revised manuscript for the validation.

L555 also removed permanently flooded ponds and wetlands, it should be noted

Response: Yes, thanks for pointing this out. We added additional discussion about the permanent flooded ponds and wetlands at Line 755 – Line 757.

L559 What is a finer-resolution DEM? 10-m? 90-m? 250 m? Clarify.

Response: In the revised manuscript, we changed this sentence to "Fluvial inundation (simulated in MOSART) process is relatively well represented in ESM since 90m resolution DEM is used to capture the floodplain storage effects".

L582 HUC08, noted.

Response: We checked the USGS website, and it should be HUC 8.

Supplemental Comments

Eq 5: What supports the use of the default configurations for F_c and u in this equation? How

would the model perform if a different suite of parameters were selected? Are there parameters appropriate for a CONUS-scale model? Can that be supported by literature?

Response: As described in the model description tech note of Oleson et al. (2013), $f_c = 0.4$ and $\mu = 0.14$ were assumed to be globally uniform due to the lack of microtopography dataset. During the design of this study, we found the pluvial inundation process in our model is more sensitive to f_c and f_{over} . Therefore, we calibrated those two parameters in this study to improve the model performance of simulating surface water dynamics. Our study is the first to improve the representation of surface water dynamics in our model by calibrating those two parameters. Previous studies used the default global uniform parameter values, such as Ekici et al. (2019). As shown in the supplementary materials Figure S9, using default parameter values fail to capture the spatial pattern of surface water.

Figure S1: How did the authors handle the apparent errors in GLAD that affected the distribution of water across the CONUS? For instance, it's apparent that March, May, October, and November panels have generally vertical areas of No Data (perhaps due to the scan-line correct failure; I'm unsure). How did those errors affect your analyses? (And why not cross-walk the GLAD data with NLCD data to see how well it captured the wetlands and open waters?)

Response: Fig.S1 is included to demonstrate the poor spatial coverage of satellite dataset at monthly scale, so the seasonal map of GLAD was used for calibration. The GLAD seasonal dataset has much better spatial coverage. We note the GLAD seasonal dataset is the monthly averaged surface water dynamics derived from 1999-2020.

Figure S3 What are the demarcations of the CONUS (here and throughout)? Were the permanent waters removed from the comparison?

Response: The demarcations denote the subregions of CONUS, Northeast, Midwest, West, and South. In the revised manuscript, we added Fig.S7 to explain the demarcations. The permanent water bodies are removed from GLAD in this figure and other similar comparison figures. We clarified this in the caption.

Figure S4 and throughout: Were the Bahamas included in these analyses? If not, I recommend removing them from the mapped study area. Same with the islands off the southern coastal of CA (and perhaps the area around Cape Cod, too). Note that the legend includes white as zero values whereas the caption says white is no data. Recommend using gray for no data here and throughout.

Response: Thank you for the suggestion. We only plotted the border of the United States continent in the revised manuscript. We also changed the no data to gray in all the relevant figures in the revised manuscript.

Figure S5 Clarify in the caption that these were analyzed at what pixel size? Was this the 120 km² pixel contrast, after the GLAD data had been coarse-scaled to 120 km²? And were the surface waters here not including those permanently inundated systems?

Response: Thank you for this comment. We confirmed Fig.S5 shows the validation with all the grid cells in the domain. And it is validated against GLAD with permanent water bodies removed.

Figure S6c Why is there such a great discrepancy between the GIEMS and modeled surface water in 2005? Add to caption or text. And, here and elsewhere, does the modeled surface water include those permanent bodies? Does GIEMS include 'permanently inundated' data?

Response: There could be several reasons for the significant discrepancy between GIEMS and the simulated surface water in 2005. One is uncertainties in the NLDAS atmosphere forcings used to drive the model, and it is very common for large scale reanalysis dataset to have large bias. Another reason can be the uncertainty from the GIEMS dataset. As we show in Fig.S1, the spatial coverage of satellite dataset can be very limited at sub-annual scale, which can result in significant bias when upscale to annual scale.

We would like to clarify that permanent water bodies are not equal to permanently inundated wetlands. Permanent water bodies include rivers, lakes, reservoirs, most of which are also excluded in the wetland definition by Coward et al. (1979) and instead defined as deep water habitats. We removed permanent water bodies from the GLAD dataset for model calibration. But because the permanently inundated wetland is not differentiable from other permanent water bodies in GLAD, permanently inundated wetland can be unintentionally removed from GLAD to derive wetland benchmark for calibration. However, as we discussed in a previous response, our model is able to capture permanent wetland. Due to the permanent wetland being unintentionally removed from the benchmark, the calibrated model may underestimate the total wetland area.

We added in the caption of Fig.S6 that GIEMS includes permanent water bodies, including rivers, lakes, reservoirs, and permanent wetlands.

Figure S7 I wonder if the scale should be red to blue? The white to blue makes it hard to discern if there are any data in the low inundation maps. To wit, should readers assume that the white is not inundated? (I'm not colorblind that I know of but very light blue and white are hard to discern...)

Response: We think a sequential colormap is better than a divergent colormap to visualize a quantity like inundation. In addition, using a red to blue still suffers the same issue around the transition from light red to white to light blue. And such red to blue colormap will result in confusion around the median inundation values.

Figure F8 square scatters? There's probably a better term. Doublecheck – PDFs can have y-axis values greater than 1.0?

Response: We changed to "squares". Yes, the y-axis of probability density functions (PDFs) represent density, and the area under each PDF is equal to 1. So, the y-axis can be larger than 1. And we made a mistake in the y-axis, which should be "density" instead of "frequency".

Figure S9 The striking difference between southeastern Canada and the northeastern United States is eye-catching. I recognize that there are different data available depending on the country. Is that striking difference because of underlying geological and climatological forcings, or a function of the data source? That's important to articulate in the manuscript...it certainly

looks to be an artifact...

Response: We believe the striking difference the reviewer mentioned refers to Fig.S9 (a), the calibrated f_c . The f_c represents the threshold of wetland (due to pluvial inundation) to outflow, which determines the maximum extent of wetland in a grid cell. f_c is much larger in southeastern Canada than northern US because the surface water is much broader in southeastern Canada than that in northern US (Fig.S4).

Figure S12 Why are there pretty substantive areas of Mexico in this plot, but not in any of the above HUC plots? Where were the HUCs derived for Mexico?

Response: This is because we used the domain of the NLDAS which includes both Mexico and partial Canada. However, we only found watershed boundary dataset from the US and Canada, thus we only focus on the US and southeastern Canada in our analysis at watershed scale. Considering wetlands are not very common in Mexico, the exclusion of analysis at watershed scale in Mexico doesn't impact our conclusion.

FS13 Add linear regression equation and R2 to the plot. Note that the linear nature is fairly evident w SSP585 but the SSP126 relationship does not appear to be linear. Why should you force a linear relationship on it?

Response: Thanks for the suggestion. We added the linear regression equation and R2 in Fig.S13 (it is Fig.S15 in the revised manuscript) for SSP585. We also removed the regression line for SSP126 due to the low correlation.

References

- Appels, W.M., Coles, A.E. and McDonnell, J.J. 2018. Infiltration into frozen soil: From core-scale dynamics to hillslope-scale connectivity. *Hydrol Process* 32(1), 66-79.
- Avis, C.A., Weaver, A.J. and Meissner, K.J. 2011. Reduction in areal extent of high-latitude wetlands in response to permafrost thaw. *Nat Geosci* 4(7), 444-448.
- Brunner, P., Cook, P.G. and Simmons, C.T. 2009. Hydrogeologic controls on disconnection between surface water and groundwater. *Water Resour Res* 45(1).
- Decharme, B., Alkama, R., Papa, F., Faroux, S., Douville, H. and Prigent, C. 2012. Global off-line evaluation of the ISBA-TRIP flood model. *Clim Dynam* 38(7), 1389-1412.
- Ekici, A., Lee, H., Lawrence, D.M., Swenson, S.C. and Prigent, C. 2019. Ground subsidence effects on simulating dynamic high-latitude surface inundation under permafrost thaw using CLM5. *Geosci. Model Dev.* 12(12), 5291-5300.
- Jackson, C.R., Thompson, J.A. and Kolka, R.K. 2014. Wetland soils, hydrology and geomorphology. In: Batzer, D.; Sharitz, R., eds. *Ecology of freshwater and estuarine wetlands*. Berkeley, CA: University of California Press: 23-60. Chapter 2., 23-60.
- Jung, M., Reichstein, M., Ciais, P., Seneviratne, S.I., Sheffield, J., Goulden, M.L., Bonan, G., Cescatti, A., Chen, J., de Jeu, R., Dolman, A.J., Eugster, W., Gerten, D., Gianelle, D., Gobron, N., Heinke, J., Kimball, J., Law, B.E., Montagnani, L., Mu, Q., Mueller, B., Oleson, K., Papale, D., Richardson, A.D., Rouspard, O., Running, S., Tomelleri, E., Viovy, N., Weber, U., Williams, C., Wood, E., Zaehle, S. and Zhang, K. 2010. Recent decline in the global

- land evapotranspiration trend due to limited moisture supply. *Nature* 467(7318), 951-954.
- Kåresdotter, E., Destouni, G., Ghajarnia, N., Hugelius, G. and Kalantari, Z. 2021. Mapping the Vulnerability of Arctic Wetlands to Global Warming. *Earth's Future* 9(5), e2020EF001858.
- Lehner, B. and Döll, P. 2004. Development and validation of a global database of lakes, reservoirs and wetlands. *J Hydrol* 296(1), 1-22.
- Mao, Y., Zhou, T., Leung, L.R., Tesfa, T.K., Li, H.-Y., Wang, K., Tan, Z. and Getirana, A. 2019. Flood Inundation Generation Mechanisms and Their Changes in 1953–2004 in Global Major River Basins. *Journal of Geophysical Research: Atmospheres* 124(22), 11672-11692.
- Maxwell, R.M., Condon, L.E. and Kollet, S.J. 2015. A high-resolution simulation of groundwater and surface water over most of the continental US with the integrated hydrologic model ParFlow v3. *Geosci. Model Dev.* 8(3), 923-937.
- O'Neill, M.M.F., Tijerina, D.T., Condon, L.E. and Maxwell, R.M. 2021. Assessment of the ParFlow–CLM CONUS 1.0 integrated hydrologic model: evaluation of hyper-resolution water balance components across the contiguous United States. *Geosci. Model Dev.* 14(12), 7223-7254.
- Oleson, K., Lawrence, D.M., Bonan, G.B., Drewniak, B., Huang, M., Koven, C.D., Levis, S., Li, F., Riley, W.J., Subin, Z.M., Swenson, S., Thornton, P.E., Bozbiyik, A., Fisher, R., Heald, C.L., Kluzek, E., Lamarque, J.-F., Lawrence, P.J., Leung, L.R., Lipscomb, W., Muszala, S.P., Ricciuto, D.M., Sacks, W.J., Sun, Y., Tang, J. and Yang, Z.-L. 2013 Technical description of version 4.5 of the Community Land Model (CLM).
- Pickens, A.H., Hansen, M.C., Hancher, M., Stehman, S.V., Tyukavina, A., Potapov, P., Marroquin, B. and Sherani, Z. 2020. Mapping and sampling to characterize global inland water dynamics from 1999 to 2018 with full Landsat time-series. *Remote Sens Environ* 243, 111792.
- Tootchi, A., Jost, A. and Ducharne, A. 2019. Multi-source global wetland maps combining surface water imagery and groundwater constraints. *Earth Syst. Sci. Data* 11(1), 189-220.
- Wu, H., Kimball, J.S., Li, H., Huang, M., Leung, L.R. and Adler, R.F. 2012. A new global river network database for macroscale hydrologic modeling. *Water Resour Res* 48(9).
- Xi, Y., Peng, S., Ciais, P. and Chen, Y. 2021. Future impacts of climate change on inland Ramsar wetlands. *Nat Clim Change* 11(1), 45-51.
- Zhang, S., Zhou, L., Zhang, L., Yang, Y., Wei, Z., Zhou, S., Yang, D., Yang, X., Wu, X., Zhang, Y., Li, X. and Dai, Y. 2022. Reconciling disagreement on global river flood changes in a warming climate. *Nat Clim Change* 12(12), 1160-1167.
- Zhang, Y., Li, C., Chiew, F.H.S., Post, D.A., Zhang, X., Ma, N., Tian, J., Kong, D., Leung, L.R., Yu, Q., Shi, J. and Liu, C. 2023. Southern Hemisphere dominates recent decline in global water availability. *Science* 382(6670), 579-584.
- Zhou, S., Yu, B., Lintner, B.R., Findell, K.L. and Zhang, Y. 2023. Projected increase in global runoff dominated by land surface changes. *Nat Clim Change* 13(5), 442-449.
- Zou, Z., Xiao, X., Dong, J., Qin, Y., Doughty, R.B., Menarguez, M.A., Zhang, G. and Wang, J. 2018. Divergent trends of open-surface water body area in the contiguous United States from 1984 to 2016. *Proceedings of the National Academy of Sciences* 115(15), 3810.

REVIEWERS' COMMENTS

Reviewer #4 (Remarks to the Author):

Xu et al. Revision to "Climate change will reduce inland wetland areas and disrupt their seasonal regimes in North America"

The authors have done well in responding to the multiple pages of comments that were made to an earlier version of this dense manuscript. I have tried to reread the paper with my original perspective and not find "new things" on which to comment. Most of the comments, therefore, are stylistic as my original comments were addressed by the authors. I did have a few areas that I think could use some additional touch, however, and note them below.

For instance, throughout the figures I see that the model inundation maximum is 0.1 (e.g., Fig 1b). Is that indeed the case? For instance, the Everglades is an exceptionally wet area, nearly year-round. Are there not wetted areas that are inundated >0.1 of the time within the cell resolution? Or were the cells downscaled for visual display purposes? It would be useful to clarify for readers what the maximum inundation just may be within the area shown.

S2 comes after S3

L131 the average performance (NSE, R2) perhaps should be given in the main text so that readers can understand how well the model is performing without requiring a trip to the Supplemental Material section

We lost S7 in the main text (so going from S6 to S8 will confuse readers); might as well do away w S7 by itself and have it as an inset in Figure S6.

L141 should perhaps add, "... (Fig. S9) when there is likely significant over-estimation of surface water extent." I added that as S9 more-or-less follows expected snow abundance in the mountainous areas of the country (and Pennsylvania, fwiw). Those areas with high wetlands in the winter are areas with a great deal of snow (noted by the authors, but that it's an over-estimation indicator should assuage some readers who will otherwise see it as tracking mountain snow).

L353-354 adding this information into the abstract, if there is room, my increase the interest across readership

L407 change "significantly" to "substantially" as the implication of the stat-loaded word is that the ~5% decrease is not significant whereas the ~10% decrease is (i.e., it begs the question of significant when compared to what, through what statistical test, and with what reported p-value and assumptions)

L430 "Other study also projected" needs to be rephrased

L434 increasing infiltration is a fine rationale, but I caution the authors to include a caveat – what happens to your increased infiltration when the available soil-water holding capacity is exceeded? There's a finite amount of space for water to go, and those areas are typically pretty water-logged to begin with. If that caveat isn't incorporated into this discussion (rationalizing the findings of this study versus others) it will leave your conclusions open to a sharp retort.

L444 Considering that the maximum inundation in the figures presented is 0.1 (e.g., Figure 1(b)), there's not much room for the wetland organisms to operate within (as 0.0 to 0.1 is rather small relative to 0.1 to 1.0). So in essence there's not much of a threshold in the authors' model to 'allow' for wetlands.

L478 This is also due to human issues associated with naming conventions. For instance, the Ramsar classification of wetlands includes lakes (e.g., <https://www.dcceew.gov.au/water/wetlands/ramsar/wetland-type-classification> and Richardson, D. C., M. A. Holgerson, M. J. Farragher, K. K. Hoffman, K. B. S. King, M. B. Alfonso, M. R. Andersen, K. S. Cheruveil, K. A. Coleman, M. J. Farruggia, R. L. Fernandez, K. L. Hondula, G. A. López Moreira Mazacotte, K. Paul, B. L. Peierls, J. S. Rabaey, S. Sadro, M. L. Sánchez, R. L. Smyth and J. N. Sweetman (2022). "A functional definition to distinguish ponds from lakes and wetlands." *Scientific Reports* 12(1): 10472.) It would be remiss to not acknowledge that this stems from definitional issues – likely moreso than by other “science-based” issues.

L520 fix ref⁴³ (same w L558, and L591, elsewhere to ensure citations comport to journal standards

L614 It would be interesting to contrast a small number of focal areas to see how many wetlands were included in the GLAD data versus wetlands in the National Land Cover Dataset (NLCD, classes 90 and 95, I believe). Reporting those data (around L614) would not invalidate any findings of this study but would provide context associated with scale and resolution (and expectations for subsequent researchers). That's to note NLCD can be used as the benchmark by which the GLAD wetland abundance can be contrasted so readers will know whether GLAD wetlands hit or miss the mark - and NLCD at 30m will be a high bar, but useful (and more useful than things like the GLWD, which is coarse and dated).

Reviewer #4 (Remarks to the Author):

Xu et al. Revision to “Climate change will reduce inland wetland areas and disrupt their seasonal regimes in North America”

The authors have done well in responding to the multiple pages of comments that were made to an earlier version of this dense manuscript. I have tried to reread the paper with my original perspective and not find “new things” on which to comment. Most of the comments, therefore, are stylistic as my original comments were addressed by the authors. I did have a few areas that I think could use some additional touch, however, and note them below.

Response: We appreciate the reviewer’s efforts of reviewing our revised manuscript. We have carefully addressed the comments in the revised manuscript. Please find our point-by-point response in the following.

For instance, throughout the figures I see that the model inundation maximum is 0.1 (e.g., Fig 1b). Is that indeed the case? For instance, the Everglades is an exceptionally wet area, nearly year-round. Are there not wetted areas that are inundated >0.1 of the time within the cell resolution? Or were the cells downscaled for visual display purposes? It would be useful to clarify for readers what the maximum inundation just may be within the area shown.

Response: The maximum inundation can be larger than 0.1 for some regions, for example, the Everglades as the reviewer suggested. We set the maximum value in the colorbar to be 0.1 for better visualization because most regions have maximum inundation less than 0.1. In the revised manuscript, we modified the label in the colorbar to be ≥ 0.1 .

S2 comes after S3

Response: We change Fig.S2 to Fig.S7 in the revised manuscript to make sure the figures from the supplementary material are discussed in the correct order in the main text.

L131 the average performance (NSE, R2) perhaps should be given in the main text so that readers can understand how well the model is performing without requiring a trip to the Supplemental Material section

Response: Thank you for the suggestion. In the revised manuscript, we added the performance metric at Line 131.

We lost S7 in the main text (so going from S6 to S8 will confuse readers); might as well do away w S7 by itself and have it as an inset in Figure S6.

Response: Thank you for the suggestion. Adding Figure S7 as an inset in Figure S6 will make the figure very busy. In the revised manuscript, we moved Figure S7 to the end of the supplementary materials as Figure S19. Then the supplementary materials figures in the main text are discussed in the correct order.

L141 should perhaps add, “...(Fig. S9) when there is likely significant over-estimation of surface water extent.” I added that as S9 more-or-less follows expected snow abundance in the mountainous areas of the country (and Pennsylvania, fwiw). Those areas with high wetlands in

the winter are areas with a great deal of snow (noted by the authors, but that it's an over-estimation indicator should assuage some readers who will otherwise see it as tracking mountain snow).

Response: Thank you for the suggestion. We added this at Line 140 – Line 141 in the revised manuscript.

L353-354 adding this information into the abstract, if there is room, my increase the interest across readership

Response: The impacts of climate change on the major wetland habitats were mentioned in the abstract at Line 24 – Line 26. However, due to the word limit in the abstract, we cannot specify the specific values of the relative changes.

L407 change “significantly” to “substantially” as the implication of the stat-loaded word is that the ~5% decrease is not significant whereas the ~10% decrease is (i.e., it begs the question of significant when compared to what, through what statistical test, and with what reported p-value and assumptions)

Response: We changed “significantly” to “substantially” as the reviewer suggested.

L430 “Other study also projected” needs to be rephrased

Response: We modified it to “Other study also suggested a potential loss of wetland...”.

L434 increasing infiltration is a fine rationale, but I caution the authors to include a caveat – what happens to your increased infiltration when the available soil-water holding capacity is exceeded? There's a finite amount of space for water to go, and those areas are typically pretty water-logged to begin with. If that caveat isn't incorporated into this discussion (rationalizing the findings of this study versus others) it will leave your conclusions open to a sharp retort.

Response: In E3SM land model (ELM), the infiltration rate is constrained when soil-water approaches the soil-water holding capacity. For example, the infiltration rate is zero when the soil is saturated (See Eq (3) in the Methods section). In addition, if the vertical groundwater flow is larger than the lower soil layer capacity, the additional water is discharged to the surface water storage and become standing surface water. Therefore, the situation mentioned by the reviewer is well considered in our model process. We clarified this at Line 464 – Line 466 in the revised manuscript.

L444 Considering that the maximum inundation in the figures presented is 0.1 (e.g., Figure 1(b)), there's not much room for the wetland organisms to operate within (as 0.0 to 0.1 is rather small relative to 0.1 to 1.0). So in essence there's not much of a threshold in the authors' model to 'allow' for wetlands.

Response: The maximum inundation fraction of a grid cell can be much larger than 0.1 over some regions. We set the range of the colorbar to be from 0 to 0.1 for better visualization.

Given our simulations are at spatial resolution of $\frac{1}{8}^\circ \times \frac{1}{8}^\circ$ (i.e., $\sim 12.5\text{km} \times 12.5\text{km}$), an inundation fraction of 0.1 represents substantial inundation area, for example, $0.1 \times 12.5\text{km} \times 12.5\text{km} = 15.625 \text{ km}^2$.

L478 This is also due to human issues associated with naming conventions. For instance, the Ramsar classification of wetlands includes lakes (e.g., <https://www.dcceew.gov.au/water/wetlands/ramsar/wetland-type-classification> and Richardson, D. C., M. A. Holgerson, M. J. Farragher, K. K. Hoffman, K. B. S. King, M. B. Alfonso, M. R. Andersen, K. S. Cheruveil, K. A. Coleman, M. J. Farruggia, R. L. Fernandez, K. L. Hondula, G. A. López Moreira Mazacotte, K. Paul, B. L. Peierls, J. S. Rabaey, S. Sadro, M. L. Sánchez, R. L. Smyth and J. N. Sweetman (2022). "A functional definition to distinguish ponds from lakes and wetlands." *Scientific Reports* 12(1): 10472.) It would be remiss to not acknowledge that this stems from definitional issues – likely moreso than by other “science-based” issues.

Response: We added this reference in the revised manuscript at Line 411 to acknowledge lakes are counted as wetland in other classification systems.

L520 fix ref⁴³ (same w L558, and L591, elsewhere to ensure citations comport to journal standards

Response: We changed it to ref. ⁴³ after we checked the published paper from Nature Communications.

L614 It would be interesting to contrast a small number of focal areas to see how many wetlands were included in the GLAD data versus wetlands in the National Land Cover Dataset (NLCD, classes 90 and 95, I believe). Reporting those data (around L614) would not invalidate any findings of this study but would provide context associated with scale and resolution (and expectations for subsequent researchers). That's to note NLCD can be used as the benchmark by which the GLAD wetland abundance can be contrasted so readers will know whether GLAD wetlands hit or miss the mark - and NLCD at 30m will be a high bar, but useful (and more useful than things like the GLWD, which is coarse and dated).

Response: Thank you for your comment. We compared GLAD and NLCD for two 1° × 1° regions from Mississippi river valley and Prairie Pothole Region, respectively, in Figure R1. We included this figure in the supplementary materials Fig. S17 and referred it in the revised manuscript at Line 529 – Line 530. This comparison suggests the GLAD dataset may underestimate woody wetlands defined in NLCD. However, it is not clear to us if NLCD captures the wetlands accurately since NLCD was derived from Landsat images as well and uncertainty is inevitable.

Figure R1. Comparison between National Land Cover Dataset (NLCD) and Global Land Analysis & Discovery (GLAD) surface water dynamics for two $1^\circ \times 1^\circ$ regions. Subplots (a) and (b) are from NLCD, and subplots (c) and (d) are from GLAD for the corresponding regions.